# A model for super El Niños

Saji N. Hameed [1], Dachao Jin[1,2] & Vishnu Thilakan[1]

Super El Niños, the strongest and most powerful of El Niños, impact economies, societies, and ecosystems disproportionately. Despite their importance, we do not fully understand how super El Niños develop their intensity and unique characteristics. Here, combining observational analyses with simple numerical simulations, we suggest that eastern Pacific intensified super El Niños result from the interaction of an El Niño and a positive Indian Ocean Dipole. Further, we identify a self-limiting behavior inherent to El Niño Southern Oscillation (ENSO) dynamics. This behavior—a consequence of the atmospheric Kelvin wave response that develops to the east of ENSO's convective anomalies—dampens sea surface temperature (SST) variations in the eastern Pacific, thereby preventing super El Niños from developing through tropical Pacific dynamics alone. Our model explains the features of the large 1972, 1982, and 1997 El Niños; the large SST anomalies during the 2015 El Niño, however, were likely enhanced by strong decadal variability.

[1] The University of Aizu, Aizuwakamatsu, Fukushima 9658580, Japan. [2] Nanjing University of Information Science & Technology, Nanjing 210044, China. These authors contributed equally: Dachao Jin, Vishnu Thilakan. Correspondence and requests for materials should be addressed to S.N.H. (email: saji@u-aizu.ac.jp)

The year 1972 was marked by extreme climate anomalies worldwide. Catastrophic droughts gripped Central America, Sahel, Australia, Brazil, India, Indonesia, and the Soviet Union[1]. Consequently, global food production declined for the first time since the end of World War II[2]. In the far-eastern equatorial Pacific Ocean, exceedingly warm temperatures[3] led to a total collapse of Peru's fishing industry[4]. These climatic and socioeconomic catastrophes were caused by an extreme El Niño event that developed in that year.

Prior to 1972, El Niño was viewed as a regional phenomenon that interested only a few specialists[5]. The intensity and global impacts of the 1972 event brought El Niños to the forefront of the scientific research agenda[2]. The 1982 El Niño was another game changer—more intense than the 1972 event, its devastating effects[1] brought El Niños to the attention of governments worldwide[5]. Finally, it was the 1997 El Niño, the strongest event of the twentieth century, that made El Niño a term familiar to all people[5].

Not only do these extreme El Niños stand out for their powerful impacts, but they also have significantly different properties from other El Niños. Their extremely strong interannual sea surface temperature (SST) anomalies (Supplementary Fig. 1), exceeding three standard deviations[6,7], are eastern-ocean intensified: their amplitude increases from the central towards the far-eastern Pacific[8,9] with strong variations along the western South American coast. These anomalies are accompanied by an extreme east–west tilt in the thermocline depth anomaly across the Pacific and unusually strong zonal wind stress anomalies in the western and central equatorial Pacific[6]—three key state variables that represent interactions across the ocean–atmosphere interface. Analyzing these variables together, Hong et al.[6] found that the 1972, 1982, and 1997 events formed a distinct super El Niño class, statistically well separated from other El Niños.

Eastern-ocean intensification has important climatic implications. The eastern equatorial Pacific is normally devoid of rainfall[9] due to the presence of the so-called cold tongue[10,11]—a region where SST is below the convective threshold of about 27.5 °C[12]. Eastern-ocean intensification signals a large eastward expansion of the warm pool, which by reducing the intensity and spatial extent of the cold pool[9,13], favors extraordinary rainfall in the normally dry eastern equatorial Pacific—as observed during the 1972, 1982, and 1997 El Niños[2,4,9].

In general, eastern-ocean intensification is not a feature of equatorial Pacific SST anomalies, nor do strong rainfall anomalies occur in the far-eastern equatorial Pacific[9] during El Niño. Figure 1 shows the interannual standard deviation of SST and outgoing longwave radiation (OLR) anomalies—here, OLR is used as a proxy for rainfall; negative OLR anomalies are associated with positive rainfall anomalies and vice versa. SST anomalies are eastern-ocean intensified only during boreal spring. From early summer, the SST anomaly maximum shifts away from the far-eastern Pacific, moving west of 120°W by boreal winter. For OLR, standard deviations exceeding 10 W m$^{-2}$ occur west of 135°W only, except during boreal spring. Eastern-ocean intensification is therefore an unusual and unique feature of the 1972, 1982, and 1997 El Niños, as recent analyses demonstrate[14,15].

Proposed factors for extreme El Niños include oceanic nonlinearity[16], state-dependent stochastic noise[17] such as strong westerly wind events[18–20], Pacific Ocean heat content[21], and nonlinear interaction between convection and SST[22]. However, none of these hypotheses take into account the observation that super El Niños are eastern-ocean intensified. Several of these hypotheses were put to test when nearly perfect conditions for super El Niño development were perceived over the Pacific in early 2014. Subsequently, many climate models and groups around the world predicted that a super El Niño would occur in 2014[5,23], although a minority of forecasts were neutral. The event did not materialize despite the supposedly favorable conditions being present[23].

Numerical experiments with state-dependent westerly wind bursts[19] are also used to explain why some El Niño events[24] are eastern-ocean damped (i.e., not eastern Pacific intensified). However, very strong westerly wind burst amplitudes are used in these experiments: 0.17 N m$^{-2}$ in Eisenman et al.[25] and 0.25 N m$^{-2}$ in Chen et al.[19] (for comparison, monthly zonal stress anomalies during the extremely strong 1997 El Niño are weaker than 0.1 N m$^{-2}$. Supplementary Fig. 2 shows that daily zonal wind stress anomalies greater than 0.17 N m$^{-2}$ occurred only 0.2% of

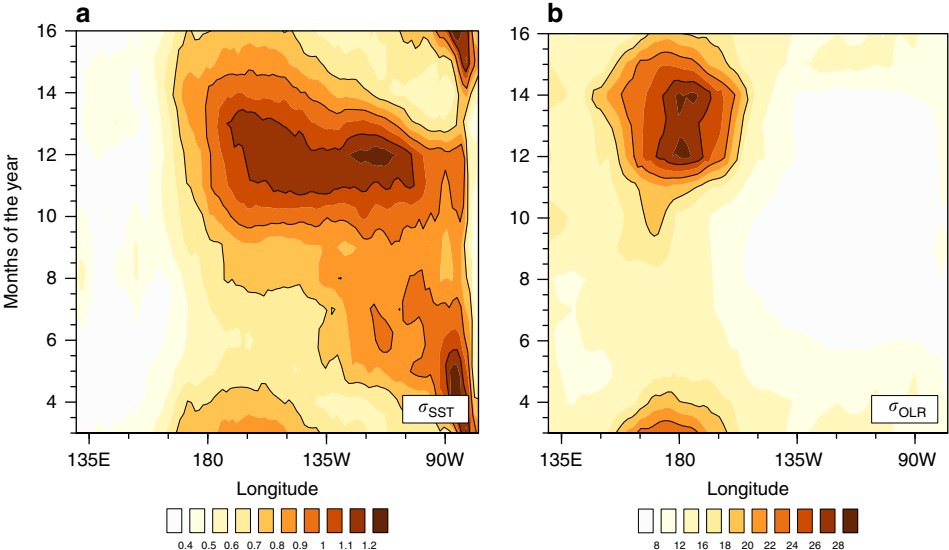

**Fig. 1** Seasonal cycle of the standard deviation of interannual equatorial Pacific SST and outgoing longwave radiation anomalies. The seasonal cycle of standard deviation for **a** SST (unit: K) and **b** outgoing longwave radiation (OLR, unit W m$^{-2}$) is presented from March, instead of from January—this represents the ENSO signature around the peak of its variations better. Monthly SST anomalies[55] and OLR anomalies[56] for the period 1982 to 2015 were used in the calculation. Before we calculated their standard deviations, we averaged SST anomalies and OLR anomalies over 5°S to 5°N. The data for all months of 1982 and 1997 were excluded from the analysis

the time during the event, with a maximum occurrence rate of 1.7% in March 1997. Further, the wind bursts are prescribed at least for 20 days (twice the model's time-step) in these experiments[25] to avoid numerical instabilities—a 20-day westerly wind burst is on the long side of observed events[25]. Finally, the westerly wind burst models are rather ad hoc, and it is unclear what physical mechanisms generate the prescribed relation between wind bursts and SST.

The influence of the Indian Ocean Dipole (IOD)[26], a coupled mode of tropical Indian Ocean climate variability, is neglected in available super El Niño hypotheses, although observational and modeling evidence suggest that IOD may modulate El Niño Southern Oscillation (ENSO) characteristics[27–31]. Note that all the three super El Niños, identified by Hong et al.[6], co-occurr with positive IOD events[26,32,33] (a positive IOD event is characterized[26,32–34] by anomalously cool SST in the equatorial eastern Indian Ocean and warm SST in the equatorial central to western Indian Ocean; during negative IOD events, this anomaly pattern reverses).

We wish to suggest a radically different, but simpler, model to explain why eastern-ocean damped El Niños are the norm rather than the exception, and why super El Niños are strong and eastern-ocean intensified. Our analysis suggests that eastern Pacific SST variations are damped by the atmospheric Kelvin wave-induced circulation found east of ENSO convective anomalies. We refer to this effect as self-limitation, reflecting the idea that it arises from ENSO dynamics itself. Next, we show that IOD-induced western Pacific surface wind anomalies overcome self-limiting ENSO dynamics effectively. Forced by eastern Indian Ocean convective anomalies, these wind anomalies persistently modulate the equatorial Pacific Ocean throughout the IOD life-cycle, and are a key to understanding the evolution of super El Niños in our model. Finally, we highlight the differences of the extreme 2015 El Niño from the super El Niños mentioned, and show that its SST anomalies were strongly influenced by decadal variations. Our model, although simple, brightens the prospect for long-term prediction of super El Niños and their impacts and has profound implications for understanding and simulating El Niño.

## Results

**Self-limiting ENSO dynamics.** The lack of eastern-ocean intensification in Fig. 1 may be explained in terms of the equatorial atmospheric response to Pacific SST anomalies. SST, convection, and surface wind anomalies modify each other continuously in a positive (Bjerknes) feedback loop during a developing El Niño: a warm SST anomaly enhances deep convection; enhanced convection forces wind anomalies that, in turn, amplify the original SST anomaly. However, wind anomalies are oppositely directed in the atmospheric Rossby and Kelvin responses generated by enhanced convection, and therefore affect the SST differently[35]. On the one hand, westerly wind anomalies in the atmospheric Rossby wave favor the growth of warm SST anomalies—not only locally but also in the eastern Pacific—by forcing eastward propagating oceanic Kelvin waves that deepen the thermocline along its path. On the other, easterly wind anomalies in the atmospheric Kelvin wave counter the growth of warm SST anomalies in the eastern Pacific[35]. It is these easterly surface wind anomalies—an inherent component of

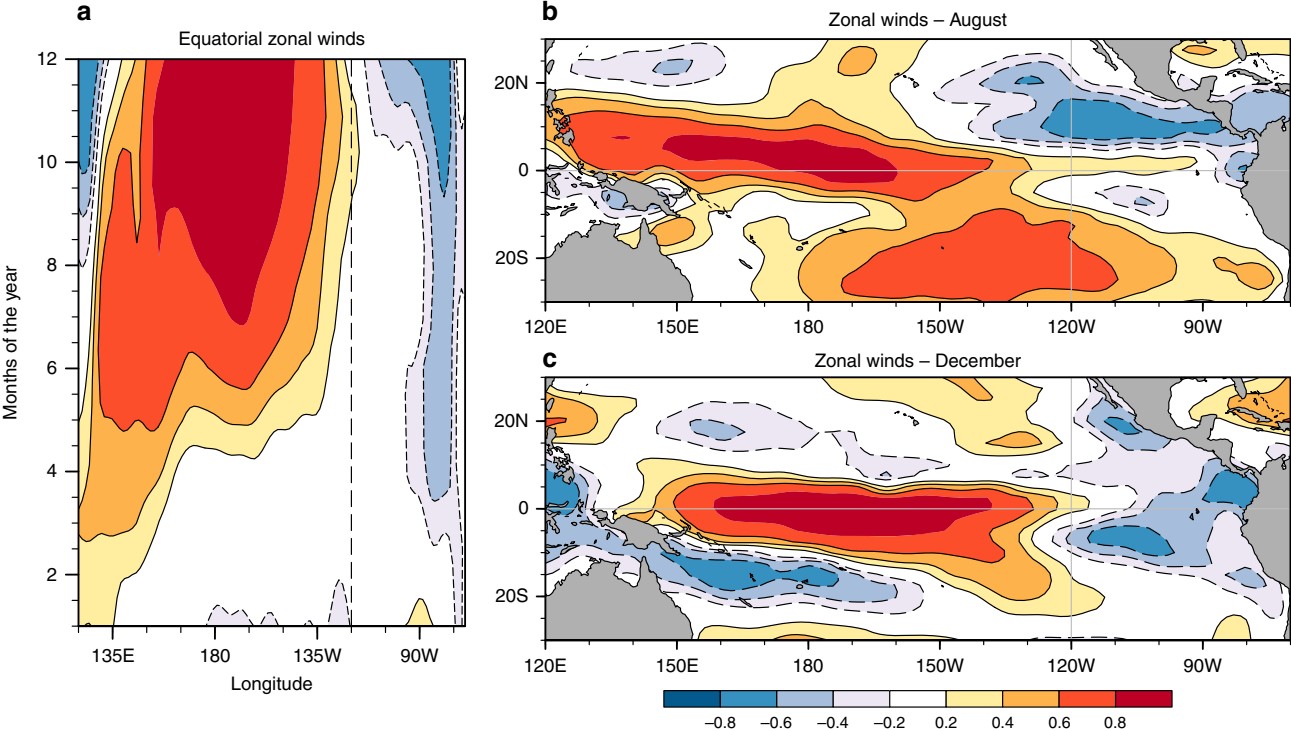

**Fig. 2** The structure of zonal wind anomalies associated with El Niño. **a** The correlation between boreal winter Nino3.4 SST index and monthly equatorial (averaged over 5°S to 5°N) zonal wind anomalies. Panels **b** and **c** are the same, but here the Nino3.4 index was correlated with wind anomalies during August and December, respectively. Monthly data from 1958 to 2015 were used in the calculation. These were pre-processed by removing decadal anomalies with periods longer than 7 years. Then, a 5-month running mean was used to smooth the data of high-frequency variations. Data during all months of 1972, 1982, and 1997 were excluded from the analysis. The maps in the figure were rendered with the NCAR Command Language software (https://doi.org/10.5065/D6WD3XH5) from the Global Self-consistent, Hierarchical, High-resolution Geography Database (GSHHG). The GSHHG is available online at https://www.ngdc.noaa.gov/mgg/shorelines/gshhs.html

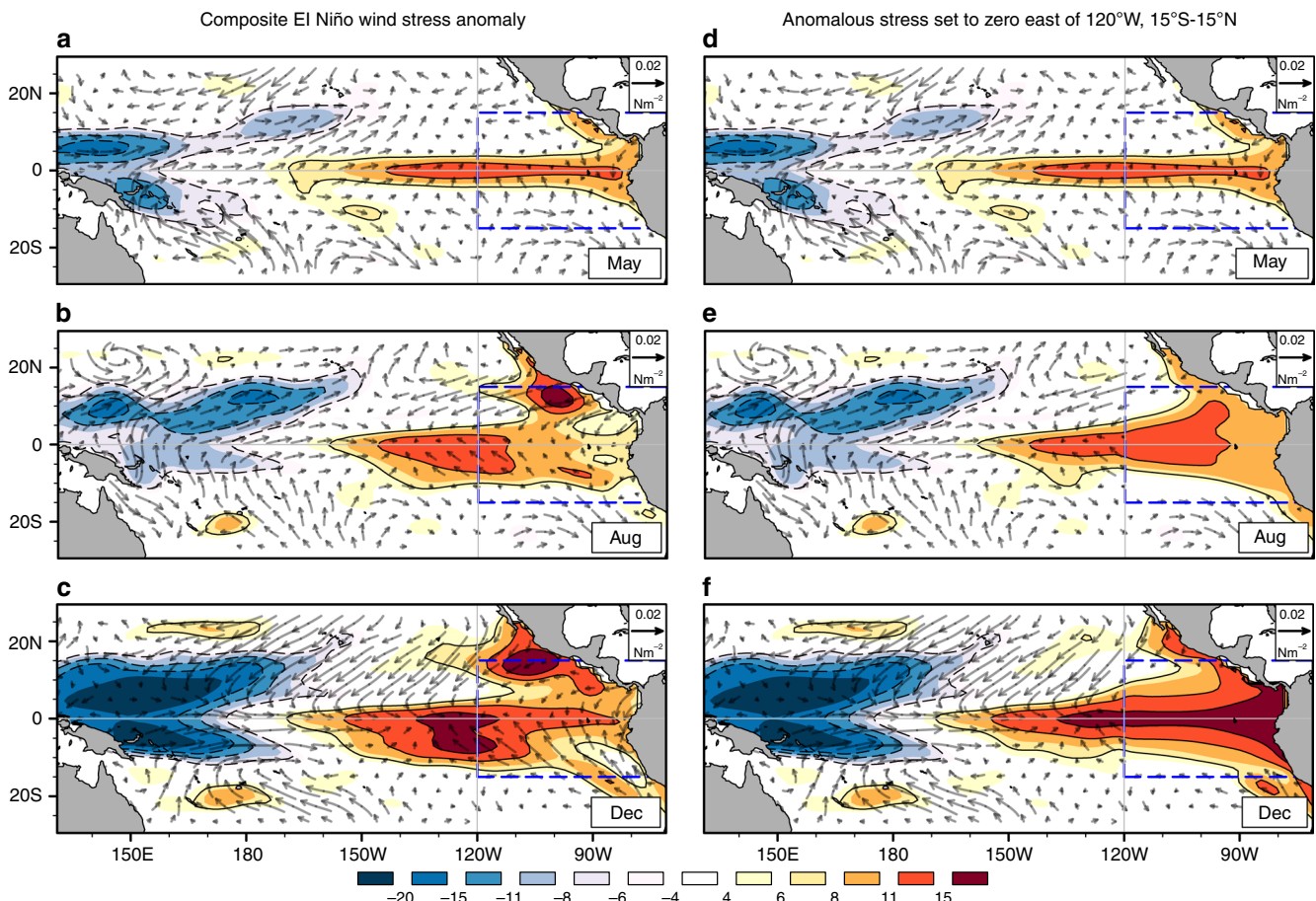

**Fig. 3** The impact of far-eastern Pacific easterly wind stress anomalies on a simulated El Niño. Thermocline depth anomalies (shaded, units: m) from two ocean model simulations are shown for May (**a**, **d**), August (**b**, **e**), and December (**c**, **f**). Left: This simulation was forced with composite wind stress anomalies (vector, units: N m$^{-2}$) during the 1986, 1987, 1991, and 2002 El Niños. Right: In the simulation shown here, the wind forcing was modified by setting the stress anomalies in the far-eastern Pacific (15°S–15°N, 120°W–290°W) to zero, from June to December. This region (bounded by the blue dashed lines in the figures) was chosen, because it experienced easterly wind stress anomalies during the composite El Niño event. The simulated SST anomalies from the two experiments are shown in Supplementary Fig. 3. The maps in the figure were rendered with the NCAR Command Language software (https://doi.org/10.5065/D6WD3XH5) from the Global Self-consistent, Hierarchical, High-resolution Geography Database (GSHHG). The GSHHG is available online at https://www.ngdc.noaa.gov/mgg/shorelines/gshhs.html

El Niño's atmospheric response—that, by favoring local equatorial divergence and upwelling, offsets the downwelling effect of oceanic Kelvin waves arriving from the west, thereby preventing strong eastern Pacific SST variations during El Niño.

In the rest of the section, we first describe the observed structure of these easterly wind anomalies, and then demonstrate their impact on the eastern Pacific Ocean using a simple linear ocean model. Finally, we discuss how seasonal SST cooling in concert with ENSO self-limiting dynamics prevents strong convective anomalies from developing over the eastern equatorial Pacific.

Equatorial easterly wind anomalies are observed (Fig. 2) throughout the year during an El Niño: confined east of 90°W during boreal spring and summer, they abruptly extend west to 120°W from October till the end of the year. Off the equator, between 5° and 15°, there is a zonal easterly wind anomaly jet, which is most prominent in boreal summer; equatorwards of this feature, weak equatorially trapped westerly wind anomalies extend eastward from about 130°W.

To understand how the easterly surface wind anomalies modulate the spatial structure of El Niño, we carried out two experiments with a simple, linear model of the tropical Pacific Ocean[36]. In one experiment (Fig. 3a, c), we forced the model with monthly means of composite El Niño wind stress anomalies from

January to December, starting from a state of rest. We then repeated the experiment, setting the wind stress anomalies to zero over the far-eastern Pacific (15°S–15°N, 120°W–80°W) from June onwards (Fig. 3d, f) (note that the model simulates depth and velocity anomalies for a shallow-water layer overlying a motionless bottom layer, and includes an empirical equation that simulates the impact of thermocline and zonal advection anomalies on SST. The simulated SST anomalies, however, do not feedback to the ocean dynamics. More details are in the Methods section).

The surface wind anomalies in the far-eastern Pacific affect the simulated El Niño in two distinct ways. At the equator, easterly wind anomalies prevent eastern-ocean intensification of the thermocline anomaly (Fig. 3b, c); removing them facilitates an eastern-ocean intensified ocean response (Fig. 3e, f). North of the equator, the meridional variation of flow in the easterly surface wind anomaly jet creates a cyclonic curl. The curl forces north–south dipolar thermocline depth anomalies, generating the zonally elongated positive thermocline anomalies (Fig. 3b, c) that extend westward from south of Baja California.

Atmospheric Kelvin waves forced by tropical Pacific convective anomalies explain the equatorial easterly anomalies, but not the easterly surface wind anomaly jet. However, during El Niño,

convective anomalies are enhanced in the intertropical convergence zone (ITCZ)[9] also. Lying north of the equator, the meridionally narrow, but zonally elongated, ITCZ anomaly generates, through a Rossby wave response[37], weak equatorial westerly anomalies flanked by easterly anomalies to its north and south. At the equator, the Rossby wave response partially cancels the atmospheric Kelvin wave's easterly anomalies, while it enhances the latter's easterly anomalies north and south of the equator. This interaction creates the off-equatorial easterly wind anomaly jet in the far-eastern Pacific.

The fetch of the easterly wind anomalies over the eastern Pacific Ocean decreases as El Nino's convective anomalies extend eastward. The more the fetch, the stronger self-limitation acts to dampen eastern-ocean variations. In observations (Fig. 1b), large-scale anomalous convection rarely extends east of 120°W between boreal summer and early winter. Consequently, during the latter half of the year, self-limiting ENSO dynamics dampen eastern-ocean variations significantly.

During the latter half of the year, the seasonal cycle lowers mean SST over the eastern Pacific cold tongue[10]. Consequently, larger positive SST anomalies are needed to initiate convection in the latter half of the year. Because of self-limiting ENSO dynamics, cold tongue SST anomalies may not rise faster than the rate of seasonal cooling (Supplementary Fig. 4c), thereby preventing the development of anomalous convection east of 120°W during regular El Niños.

In other words, Bjerknes feedback within the tropical Pacific alone cannot raise SST anomaly fast enough to counter seasonal SST cooling in the eastern Pacific and keep convective anomalies expanding towards the eastern Pacific. Therefore, tropical Pacific dynamics alone cannot explain why eastern-ocean intensified super El Niños exist.

**Impact of IOD on western Pacific surface winds.** Why then do super El Niños with strong eastern-ocean intensification exist? Their existence requires the presence of external factors (not originating through Bjerknes feedback within the tropical Pacific) with the following properties: the factor must operate during the months when seasonal SST cooling tendency is significant (July to November[10]), and it should augment the Bjerknes feedback during El Niño to accelerate the growth of warm SST anomalies over the cold tongue. Because IOD develops from early boreal summer and peaks in late fall[26], it satisfies the first property.

Here, we show that IOD generates westerly wind anomalies over the western Pacific through a downstream teleconnection, which is maintained throughout IOD growing phase. In the next section, we show that these moderate—but persistent—westerly wind anomalies can potentially augment Bjerknes feedback during El Niño to generate super El Niños.

The role of IOD in modulating ENSO has been explored before. Saji and Yamagata[27] show, from historical data, that El Niños develop and decay faster and were stronger when they co-occur with positive IODs. Based on these properties, they suggest that IOD may distinctly influence El Niño evolution. Subsequent studies[29–31] add further support: Behera et al.[29] show that ENSO exhibited a shorter recurrence period when an active Indian Ocean was incorporated in their coupled model; Luo et al.[30] show that an active Indian Ocean was necessary to predict the onset of tropical Pacific SST anomalies in their coupled model during 1994, 1997, and 2006—years when positive IOD events also occurred.

While the above studies support a role for IOD in influencing ENSO, the idealized numerical experiments of Annamalai et al.[38] suggest otherwise (this contradictory result will be discussed, after we elucidate the dynamics of IOD impact over the western Pacific). Further, the mechanisms by which IOD impacts ENSO are not demonstrated in the available literature.

Here, supplementing previous work, we clarify the mechanisms through which IOD modulates western Pacific winds. Next, by analyzing Indo-Pacific variations during 2006—a year with a prominent IOD event, but very weak or no El Niño—we argue that IOD-induced western Pacific wind anomalies impact tropical Pacific SST significantly.

As we shall see, IOD affects the Pacific through a downstream circulation forced by Indian Ocean convective anomalies. Therefore, we first highlight the strength of these convective anomalies. Observations suggest that Indian Ocean convective anomalies during IOD compare in magnitude to Pacific convective anomalies during El Niño: for example, at the peak of the 2015 El Niño, basin-averaged Pacific OLR variability (shaded curve in Fig. 4) is about 16 W m$^{-2}$, which is similar to the basin-averaged Indian Ocean OLR variability during a strong IOD event (the metric discussed here is the square root of areal averages of squared grid-point OLR anomalies. Also, note that the Indian Ocean domain, considered here, is about 60% of the area of the Pacific Ocean domain).

The impact of IOD on the Pacific is also controlled by the spatial structure of IOD convective anomaly—a property that we

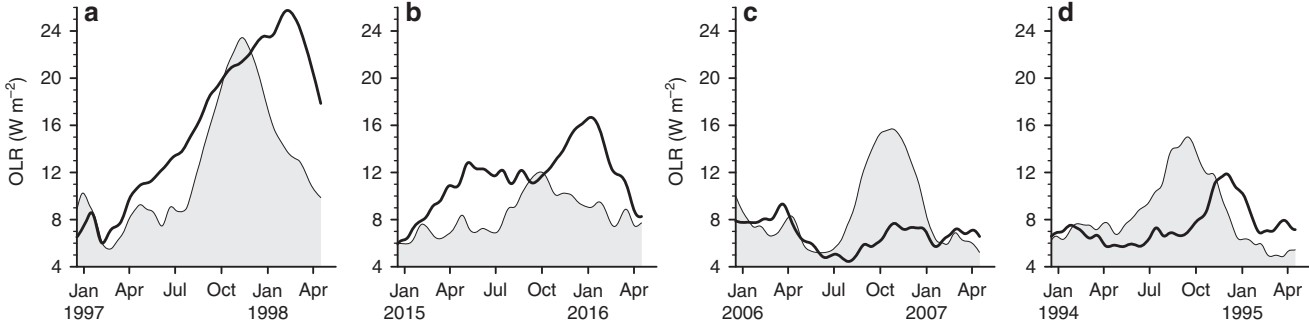

**Fig. 4** Amplitude of convective variations during selected IOD and El Niño events. The shaded curve shows root mean amplitude of squared outgoing longwave radiation (OLR) anomaly over the Indian Ocean for **a** 1997, **b** 2015, **c** 2006, and **d** 1994; the solid lines are the same, but for the Pacific Ocean. We used daily anomalies that were were low-pass filtered to retain only periodicities longer than 90 days. At each grid point, these anomalies were squared. Then, their area averages were computed over the tropical (20°S–20°N) Indian (40°E–120°E) and Pacific (140°E–280°E) Oceans—the square root of these areal averages is shown. During ENSO and IOD events, convective anomalies are not of a single sign, neither over the tropical Pacific Ocean, nor over the tropical Indian Ocean. For example, a positive IOD event is associated with negative OLR anomaly over the central tropical Indian Ocean and positive OLR anomaly over the eastern tropical Indian Ocean. The use of squared OLR anomalies capture the amplitude of both phases of OLR variations during IOD or ENSO events in a single OLR index

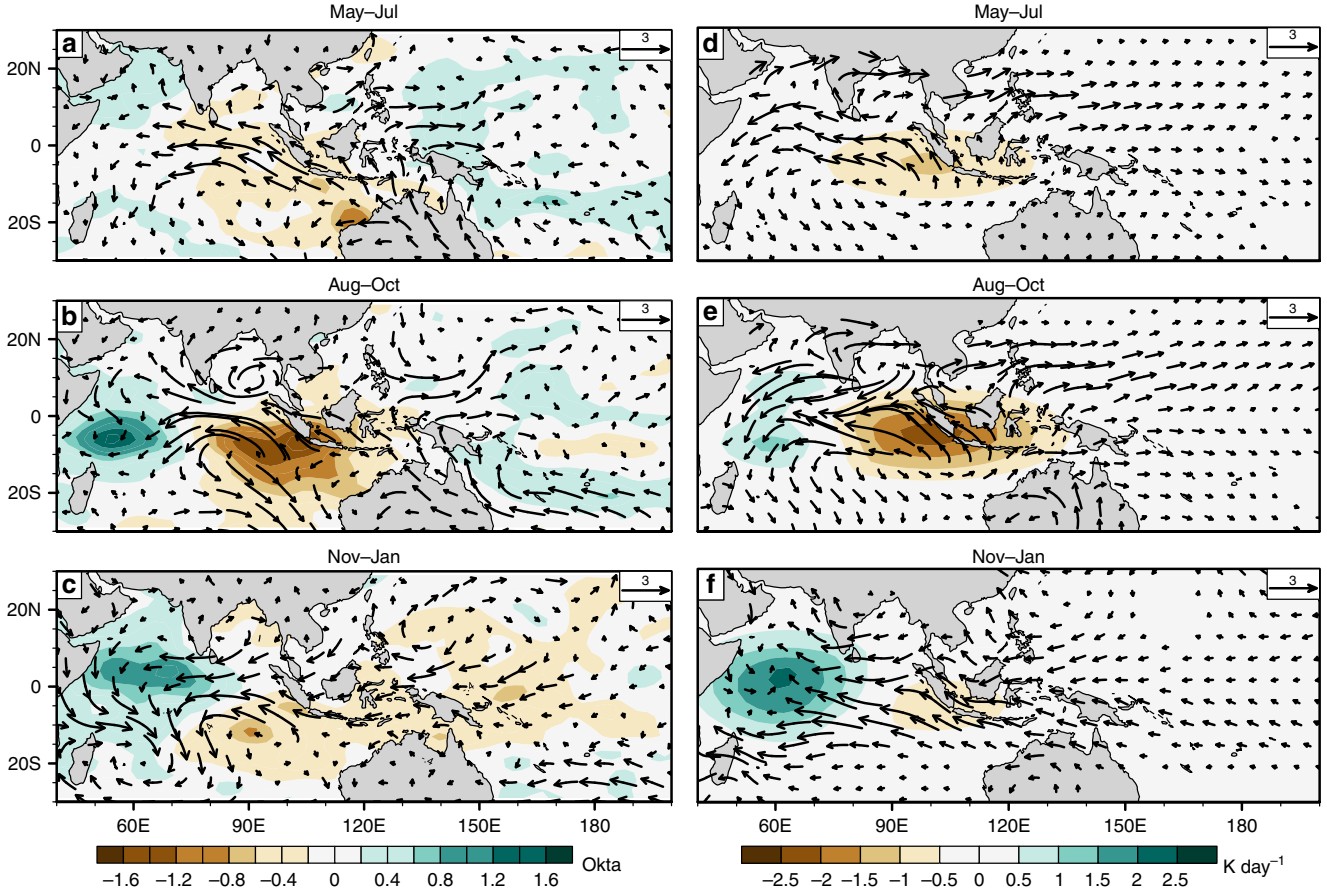

**Fig. 5** Surface wind anomalies during IOD from an observational estimate and simulations. We estimated the impact of IOD on surface wind anomalies (left) by regressing observed wind anomalies from 1958 to 2015 onto the Dipole Mode Index (DMI) and Nino3.4 SST anomaly time series. The vectors show the partial regression coefficients of wind anomalies with DMI, controlling for Nino3.4, for **a** May—July, **b** August—October, and **c** November—January. The shaded contours on the left are for marine cloudiness, which were analyzed in the same way. The brown-colored contours in (**a–c**) mean less cloudiness (and by inference, less rainfall), while green colors indicate more cloudiness (more rainfall). The right panels are for the same seasons as the left, but they show surface wind anomalies from a simulation. The synthetic thermal forcing (shaded contours in **d-f**) that was used to drive the atmospheric model mimics the patterns of marine cloudiness anomalies shown in (**a–c**); anomalously low thermal heating (brown contours in **d–f**) was prescribed in regions where anomalously low rainfall is noted in the observations (**a–c**); enhanced thermal heating (green shading in **d–f**,) in regions with anomalously high rainfall. A partial regression measures the amount by which the dependent variable (e.g., surface wind anomaly) changes when one of the independent variables (here, DMI) is increased by one unit while keeping the other independent variable (here, Nino3.4) constant (or controlled). The maps in the figure were rendered with the NCAR Command Language software (https://doi.org/10.5065/D6WD3XH5) from the Global Self-consistent, Hierarchical, High-resolution Geography Database (GSHHG). The GSHHG is available online at https://www.ngdc.noaa.gov/mgg/shorelines/gshhs.html

demonstrate shortly using idealized experiments. The observed changes in the structure of Indian Ocean convective anomalies during an IOD lifecycle are shown in Fig. 5a–c. Here, in a multiple regression analysis, we regressed observed marine cloudiness anomalies onto the Nino3.4 and Dipole Mode Index (DMI) time series, for the period 1958 to 2015. A similar analysis was also performed for surface wind anomalies (note that marine cloudiness was used instead of OLR, because OLR data are not available throughout the analysis period; reduced cloudiness is interpreted as reduced convective anomaly and vice versa. The Indian Ocean region considered is part of the tropical warm pool and features deep convection throughout the year[39]; over here, marine cloudiness may reasonably represent deep convection. The Nino3.4 is an index for ENSO, and DMI for IOD; see Methods section for details. Although ENSO and IOD both impact the variables analyzed here, multiple regression allows us to statistically isolate their relative impacts[28]).

The partial regression coefficient, which measures how many units the analyzed variable changes given a one-unit increase in DMI while statistically holding Nino3.4 constant, is shown in

Fig. 5a–c for marine cloudiness (shaded contours) and surface winds (vectors). Pacific wind anomalies change considerably during IOD evolution, alongside Indian Ocean convective anomalies. Before the mature phase of IOD, cloudiness anomalies in the eastern Indian Ocean are considerably stronger than in the western Indian Ocean; these are associated with westerly anomalies in the western equatorial Pacific. The Pacific wind anomalies are the strongest in the far-western Pacific, where their magnitude is comparable to that during El Niño (Supplementary Figs. 6, 7, 8). After IOD peaks, the eastern Indian cloudiness anomaly is weak (Fig. 5c); now the western Indian enhanced cloudiness (interpreted as more convective heating) is associated with easterly wind anomalies that extend to the western Pacific.

Simulations with a simple, linear atmospheric model help understand how IOD-associated convective anomalies impact surface winds over the tropical Pacific. Our model[40] (see Methods section) has a global domain and multiple vertical levels, and uses the primitive equations linearized about the observed[41] climatology for 1981–2010. We forced the model by prescribing (idealized) diabatic heating anomalies associated with a positive

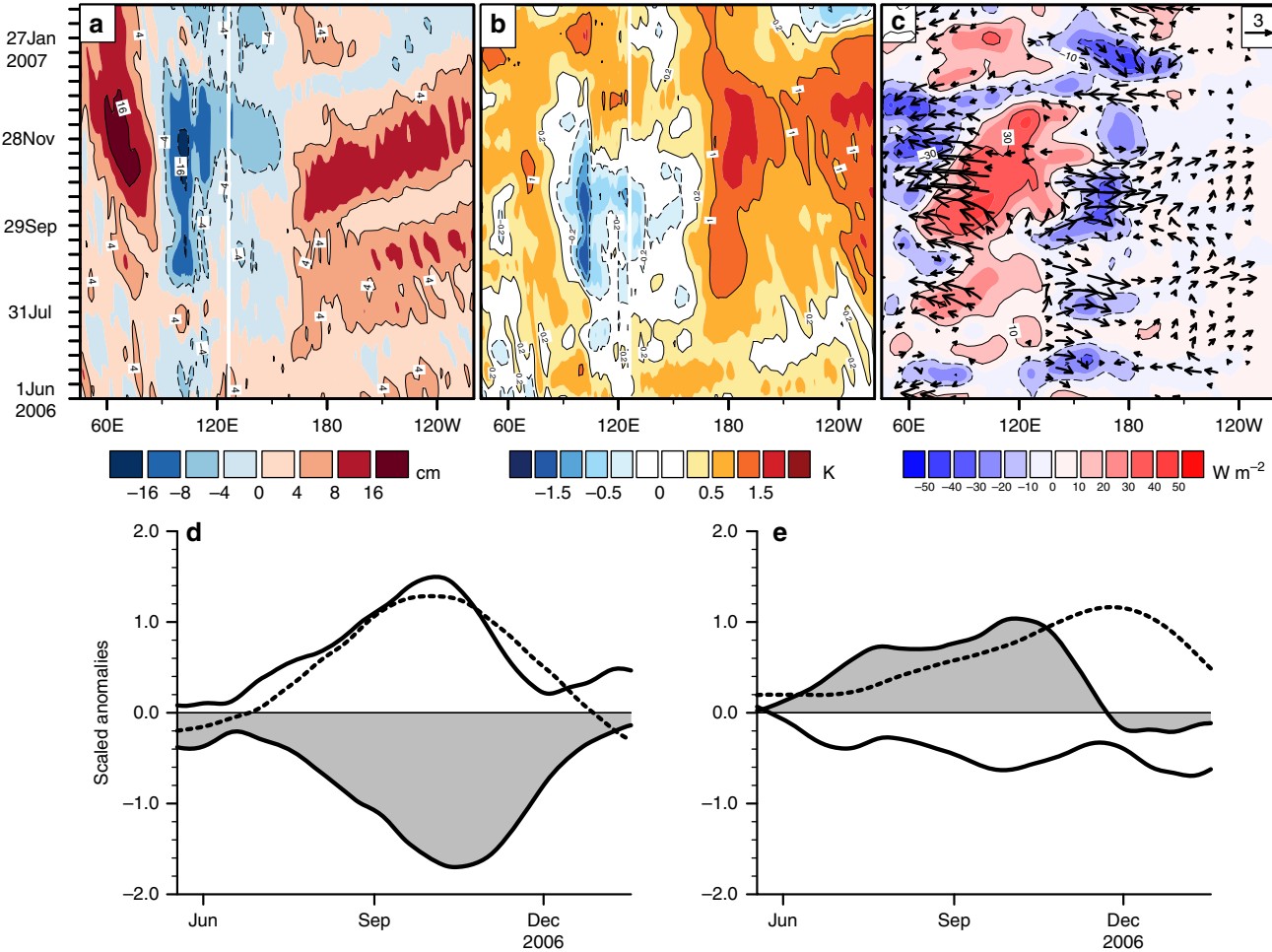

**Fig. 6** Evolution of Indo-Pacific climate anomalies during 2006. Time-longitude evolution of **a** sea surface height (SSH), **b** SST, and **c** outgoing longwave radiation (OLR) and surface zonal wind anomalies (vector, units: m s$^{-1}$). All variables were averaged between 5°S and 5°N, except over the Indian Ocean—here, SST, SSH, and OLR anomalies were averaged from 10°S to the Equator, considering their strong equatorial asymmetry during IOD. We used daily resolution data that were low-pass Lanczos filtered to remove (weather) variations shorter than 30 days. IOD strong and robust positive OLR anomalies over the eastern equatorial Indian Ocean drive easterly anomalies over the Indian Ocean, as the Rossby wave response, and westerly anomalies over the Pacific, as the Kelvin wave response. Area-averaged anomalies over selected regions contrast the nature of ocean–atmosphere coupling in the Indian (**d**) and Pacific (**e**) oceans—the time series were further smoothed with a 31-day running mean. In **d**, DMI (dashed line) evolves tightly in step with equatorial Indian Ocean zonal wind (shaded curves, 80°E–100°E) and OLR anomalies (solid line, 80°E–110°E). In sharp contrast, over the equatorial Pacific (**e**), Nino3.4 SST (dashed line) anomalies significantly lag both equatorial zonal wind (shaded curve, 150°E–180) and OLR (solid line,150°E–180°E) anomalies. In **d** and **e**, zonal wind was divided by a factor of 2 and OLR by 25

IOD event in the tropical Indian Ocean. This forcing (shaded contours in Fig. 5d–f) consisted of two elliptical patches: the negative one represents anomalously low rainfall in the eastern Indian Ocean, and the positive patch represents anomalously enhanced rainfall in the western Indian Ocean. Mimicking the structural change of convective anomalies during IOD, the negative patch had a stronger amplitude prior to IOD peak phase. After the peak phase of IOD, we made the positive patch the dominant heating pattern. Here, we assumed that diabatic heating is exclusively associated with large-scale tropical deep convection[42], and its vertical distribution is a sine curve with a maximum at the mid-troposphere and zero at the surface (for simplicity, we neglected secondary effects on surface winds induced by low-level heating, for example, the influence of SST gradients[43]).

The simulated wind anomalies (Fig. 5d–f) agree well with the observations (Fig. 5a–c), considering the experimental simplicity. The Pacific response is shaped by atmospheric Kelvin waves forced by IOD convective anomalies. However, since these waves

are distorted by the model's background flow[44], we repeated the experiments under a resting basic state (Supplementary Fig. 5) to clarify that atmospheric Kelvin waves do indeed play a central role in transmitting IOD influence to the tropical western Pacific.

We found that the structure of the prescribed heating impacted western Pacific wind anomalies strongly. With a dominant negative patch (Fig. 5d, e), westerly wind anomalies were forced over the western Pacific. However, when the positive patch is dominant, as in Fig. 5f, easterly wind anomalies were forced over the western Pacific.

The conclusion of Annamalai et al.[38] that IOD did not influence the tropical Pacific can now be explained. Because they inappropriately prescribed a zonal dipole to represent IOD diabatic heating anomaly in their atmospheric simulation, the oppositely signed Kelvin waves, generated by the dipole, largely canceled each other over the western Pacific (such a heating pattern may exist during the peak phase of IOD, but it is the eastern Indian Ocean negative heating anomaly that is dominant during most of the lifecycle of a positive IOD (Figs. 5a, b and 6c)).

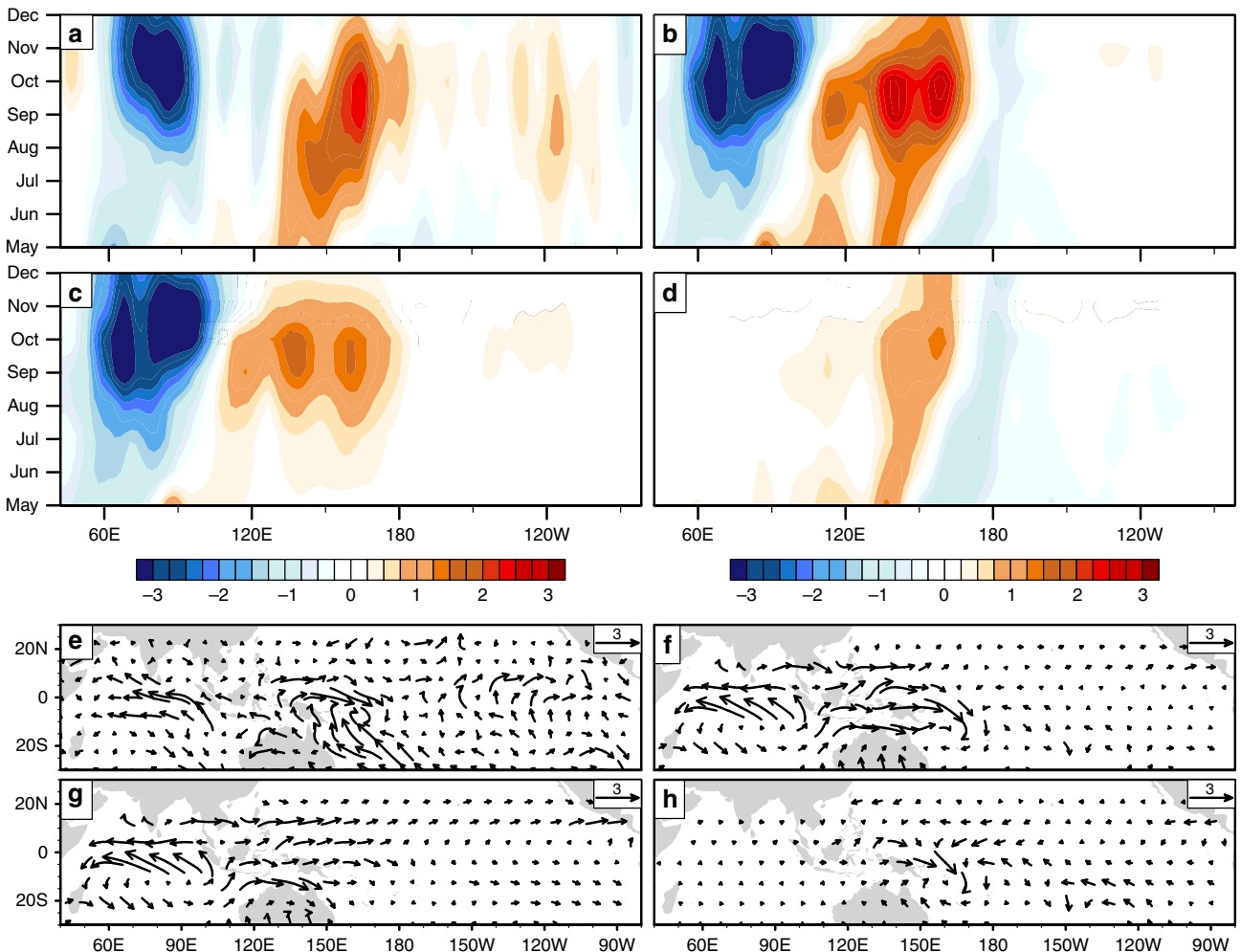

**Fig. 7** Simulation of monthly tropical surface wind anomalies during 2006. **a**–**d** Longitude time evolution of monthly equatorial zonal wind anomalies from observations (**a**) and numerical experiments (**b**–**d**). In the control experiment shown in **b**, the atmospheric model was forced with diabatic heating anomalies over the tropics covering the Indian and Pacific Oceans. In a sensitivity experiment, the forcing was prescribed only over the Indian (**c**), and in another, the forcing was prescribed only over the Pacific Ocean (**d**). **e**–**h** Surface wind anomalies, averaged between July and October, for the observations (**e**) and the numerical experiments (**f**–**h**); here, **f** is the control experiment, and **g** and **h** are the counterparts of **c** and **d** above. Comparing the surface wind anomalies from **g** against that from **f**, we estimated that the Indian Ocean forcing alone accounted for 84% of the anomalous zonal westerly amplitude over the equatorial western Pacific (160°E–170°W). While the Pacific convection also contributes a minor amount, oppositely signed surface wind anomalies to its west and east (**h**) weaken the the impact of these winds on the eastern-ocean. All the data were lightly smoothed with a 3-month running mean filter. The maps in the figure were rendered with the NCAR Command Language software (https://doi.org/10.5065/D6WD3XH5) from the Global Self-consistent, Hierarchical, High-resolution Geography Database (GSHHG). The GSHHG is available online at https://www.ngdc.noaa.gov/mgg/shorelines/gshhs.html

**IOD influence in 2006**. Having shown that IOD can impact Pacific winds, we next assess the significance of this impact for Pacific SST by analyzing a transient El Niño-like phenomenon observed during 2006.

During the transient El Niño, coherent coupled air–sea interactions are not clear in the tropical Pacific. In Fig. 6, Pacific SST anomalies are preceded by sea level anomalies propagating eastward from the western Pacific—in turn, the sea level anomalies are preceded by westerly wind anomalies over the far-western Pacific. However, it is debatable whether the SST anomalies feedback positively to the surface wind and convective anomalies: the phase relation between surface wind and SST anomalies show that although the wind anomalies forced the SST anomalies, the latter did not influence the wind anomalies (Fig. 6e); further, convective anomalies are weakly developed (Figs. 4c and 6c, e) and lead SST anomalies. It is therefore questionable whether the

Bjerknes feedback operated in the tropical Pacific during the transient 2006 El Niño.

It is likely that the prominent eastern Indian Ocean convective anomalies during the 2006 IOD drove the western Pacific wind anomalies remotely. These convective anomalies—of stronger amplitude and longer zonal scale than the Pacific convective anomalies—are strongly coupled to SST and surface wind anomalies in the Indian Ocean (Fig. 6d). To check if Indian Ocean convective anomalies drove the Pacific wind anomalies, we carried out an atmospheric simulation where the diabatic heating anomaly forcing was confined to the tropical Indian Ocean (the forcing was estimated from observed OLR anomalies during 2006). Then, we compared the surface wind anomalies (over 160° E–170°W, between July and October) from this experiment with another where the forcing was applied over both the Indian and Pacific Oceans. From the comparison, we found that the Indian

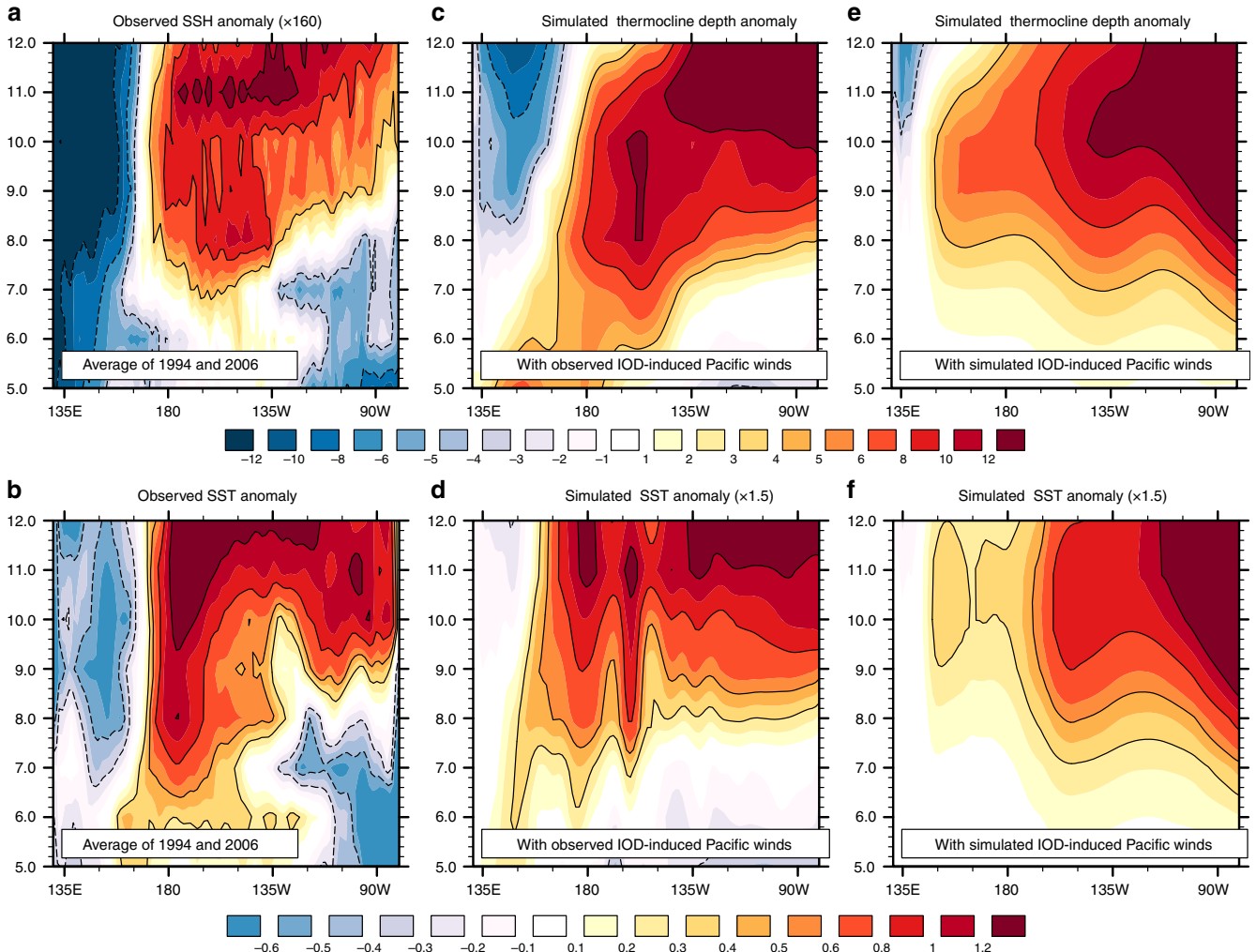

**Fig. 8** The impact of IOD-induced western Pacific wind anomalies on the equatorial Pacific Ocean. Here, we conducted two set of ocean simulations to examine how the equatorial Pacific Ocean responded to IOD-induced western Pacific surface wind anomalies. In the first set (**c**, **d**), the ocean model was driven by observed IOD-induced Pacific wind anomalies, which were estimated by compositing observed wind anomalies during 1994 and 2006. In a second set (**e**, **f**), the ocean model was driven by simulated IOD-induced surface wind anomalies, which were from an atmospheric model forced with diabatic heating anomalies over the tropical Indian Ocean (the diabatic heating anomalies were, in turn, estimated from composite OLR anomalies for 1982, 1994, 1997, and 2006). The ocean simulations may be compared with observed SSH (**a**) and SST (**b**) anomalies—both were composited from data during 1994 and 2006. The observed SSH anomalies were multiplied by 160 to facilitate comparison with the simulated thermocline anomalies (**c**, **e**): the factor of 160 is consistent with the regression coefficient between observed monthly sea level and the depth of the 20 °C isotherm in the equatorial Pacific[62]. Panels **b**, **d**, **f** show SST anomalies—the simulated SST anomalies in **d**, **f** were multiplied by 1.5, so that a single color bar can be used in the SST anomaly plots. All anomalies were averaged from 2.5°S to 2.5°N

Ocean forcing alone accounted for 84% of the strength of simulated western Pacific wind anomaly (Fig. 7) in the second experiment.

However, because the simulations only capture the low-frequency, interannual component of the wind anomalies associated with IOD, the numerical experiments cannot rule out a role for higher-frequency wind variations[45] associated with intraseasonal variations and westerly wind bursts (WWBs). Note that there are significant high-frequency variations in the wind anomalies in Fig. 6c. To address this concern, we estimated the amplitude of intraseasonal variations from daily observed[41] surface wind anomalies, using the Wheeler–Kiladis space–time filter[46], and used a WWB scheme (see Methods) to detect WWB activity. Although we did not detect any WWB events, we found that the variance associated with the easterly and westerly components of intraseasonal variations were about 18% of the

interannual wind variance (Supplementary Fig. 9). Wind anomalies in the far-western Pacific (160°E to 170°W, 5°S to 5°N), from July to October, were used in these comparisons. The impact of intraseasonal variations—which strengthen the inter-annual winds during their westerly phase, and weaken the interannual winds during their easterly phase—can be associated with the distinct peaks and valleys in the space–time evolution of the wind anomalies. Intraseasonal variations strongly modulate the interannual wind field in early August, but are much weaker than the interannual winds from middle of August.

The above analyses suggest that the transient 2006 El Niño was remotely driven by IOD, and that it did not arise from local air–sea interaction over the tropical Pacific or from stochastic forcing. A similar case can be made for the role of the 1994 IOD in the transient El Niño of that year (see Supplementary Figs. 10, 11).

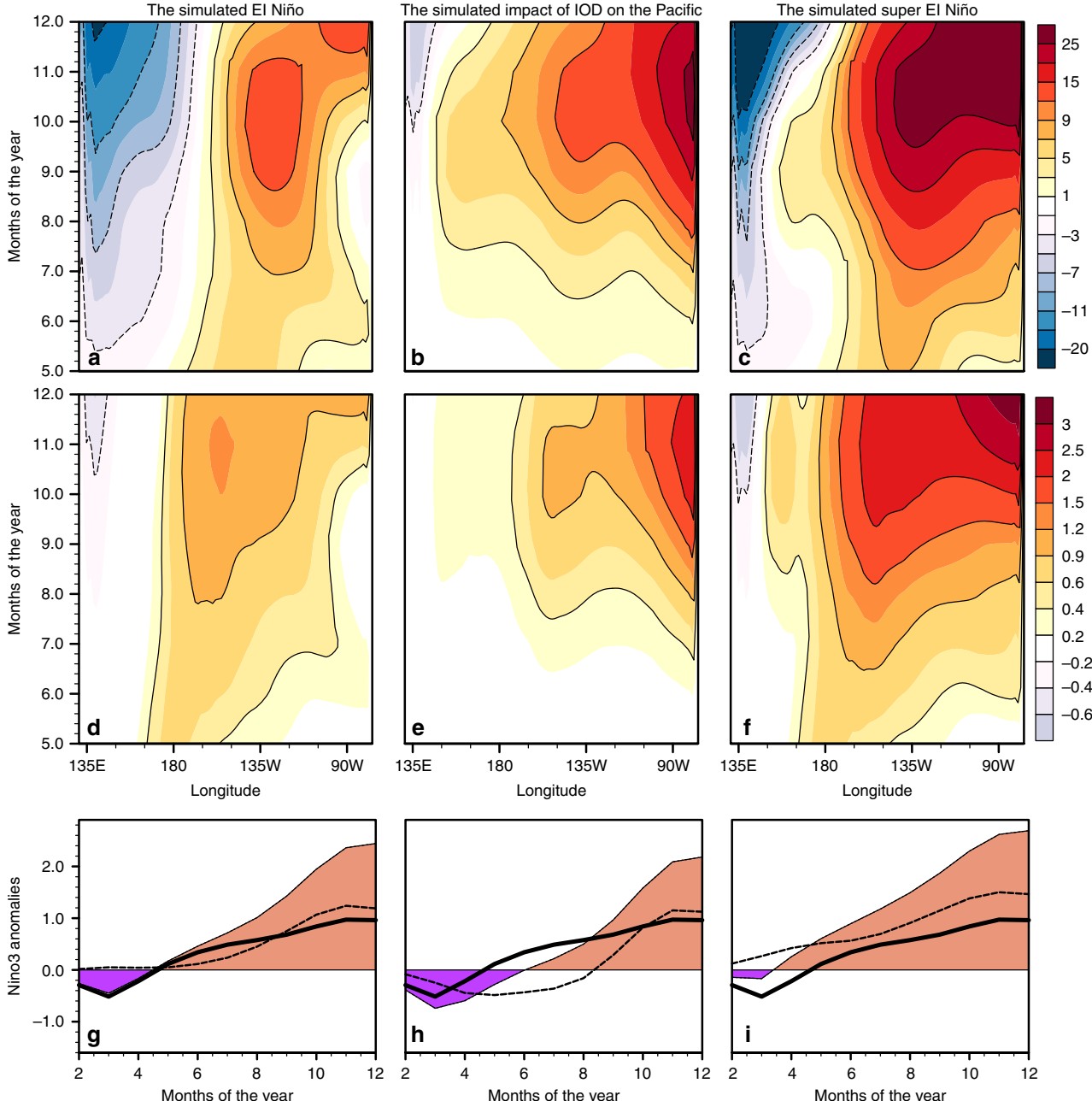

**Fig. 9** Thermocline and SST anomalies during a simulated super El Niño. In this simulation, we drove an ocean model after augmenting Pacific wind anomalies during El Niño with IOD-induced Pacific wind anomalies—the wind anomalies were from atmospheric simulations. The contributions of El Niño and IOD to the simulated super El Niño are separately shown in **a**, **d** and **b**, **e** respectively—panels **a**, **b** show thermocline anomalies, while panels **d**, **e** show SST anomalies. **c** Thermocline anomalies, and **f** SST anomalies, associated with the simulated super El Niño. **g** Temporal evolution of Nino3 anomalies calculated from the SST anomalies shown in **d**–**f**: the dashed line is the contribution from IOD, the solid line from El Niño, and the shaded curve is that for the simulated super El Niño. Panels **h**, **i** are from additional ocean simulations, where two different realizations of IOD-induced Pacific winds drove the ocean. In **h** the impact of IODs occurring without El Niño, and in **i** that of IODs co-occurring with El Niño were simulated (see text for details of the atmospheric simulations). Note that the IOD-induced part of Nino3 SST anomalies are stronger in **i** than in **h**. The observational analog of **a**, **b**, **d**, **e** are shown in the Supplementary Figs. 12–14

**The role of IOD in the evolution of super El Niños.** Now, we present oceanic simulations that elucidate additional aspects of IOD impact on the equatorial Pacific Ocean. These show that IOD-induced anomalous Pacific winds generate eastern-ocean intensified SST anomalies that, in turn, significantly augment the SST anomalies present during El Niño.

We carried out three oceanic simulations to assess how Pacific SST anomalies generated by IOD-induced western Pacific wind anomalies compare with SST anomalies during El Niño. For these experiments, we assessed IOD contribution to Pacific surface winds in two ways. One was based on observations, in which we composited observed tropical Pacific wind anomalies during 1994 and 2006—years when IOD was estimated to drive a significant part of the Pacific wind anomalies. The oceanic simulation forced with these wind anomalies are presented in Fig. 8c, d and Supplementary Figs. 12 and 13. The latter figures also contrast the

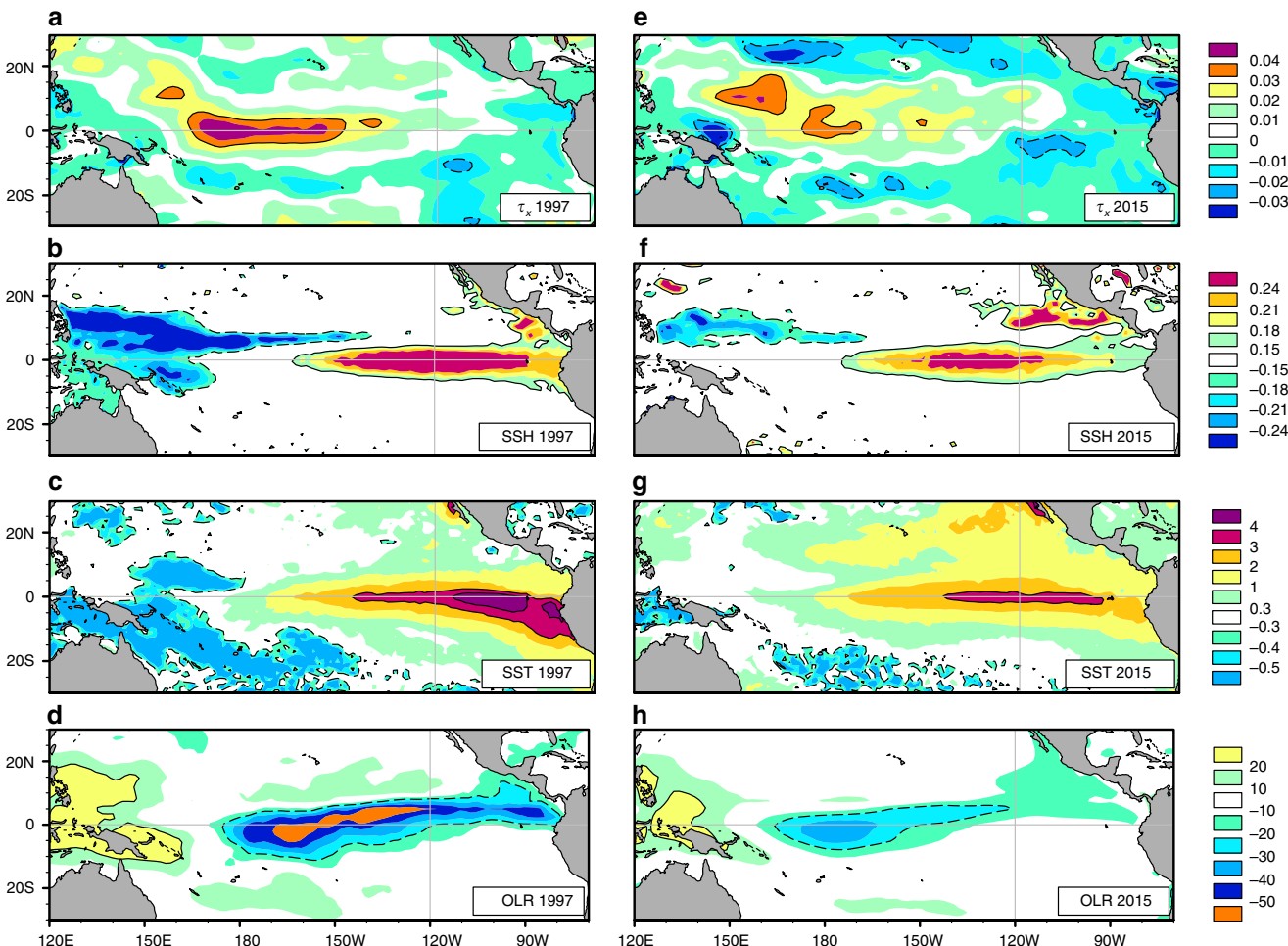

**Fig. 10** Climate anomalies during the 1997 and 2015 El Niños. The strong eastern-ocean intensified structure of SST and SSH anomalies in 1997 (**b**, **c**) contrast with the eastern-ocean damped variations in 2015 (**f**, **g**). Anomalies of zonal surface wind stress ($\tau_x$) (**a**, **e**) and OLR (**d**, **h**) are also shown. Further, an elongated, off-equatorial band of enhanced SSH and SST anomalies is seen stretching west from Baja California and Mexico in 2015; this is markedly absent in 1997. Jacox et al.[63] further contrast the nature of coastal southern California's SSH anomalies during 1997 and 2015. The absence of strong far-eastern Pacific SST anomalies in 2015 is reflected in the lack of negative OLR anomalies over this region—this may be linked to the many failed rainfall forecasts for Peru and southern California[64] during the winter of 2015/16. Further details of the differences during the two El Niños may be found in Xue and Kumar[49]. All data were averaged from October to December—this season corresponds to the peak of SST anomalies during 2015[49]. The maps in the figure were rendered with the NCAR Command Language software (https://doi.org/10.5065/D6WD3XH5) from the Global Self-consistent, Hierarchical, High-resolution Geography Database (GSHHG). The GSHHG is available online at https://www.ngdc.noaa.gov/mgg/shorelines/gshhs.html

IOD-induced oceanic solutions with that forced by observed composite El Niño wind anomalies (also see Fig. 3a–c).

Next, we estimated IOD-induced Pacific wind anomalies in a different manner using an atmospheric simulation: here, we forced the atmospheric model with diabatic heating anomalies confined to the tropical Indian Ocean (the forcing was estimated from observed, composite OLR anomalies for the 1982, 1994, 1997, and 2006 IOD events). We converted the wind anomalies from the lowest sigma level of the atmospheric model ($\mathbf{u}' = (u', v')$) to effective pseudostress anomalies ($\tau'_{\text{psx}}, \tau'_{\text{psy}}$) using[47]:

$$\tau'_{\text{psx}} = |\bar{\mathbf{u}} + \mathbf{u}'|(\bar{u} + u') - |\bar{\mathbf{u}}|\bar{u} \qquad (1)$$

$$\tau'_{\text{psy}} = |\bar{\mathbf{u}} + \mathbf{u}'|(\bar{v} + v') - |\bar{\mathbf{u}}|\bar{v}. \qquad (2)$$

Here, $\bar{\mathbf{u}} = (\bar{u}, \bar{v})$ is the mean velocity, which was taken from the observed surface wind climatology[41]. Figure 8e, f presents the oceanic simulation corresponding to this experiment.

For the third experiment, we estimated Pacific wind anomalies during El Niño using an atmospheric simulation forced by diabatic heating anomalies confined to the tropical Pacific Ocean (the heating was derived from observed, composite OLR anomalies for the 1986, 1987, 1991, and 2002 El Niño events). Figure 9a, d presents the oceanic simulation corresponding to this experiment.

In a fourth experiment, we drove the ocean model combining the simulated contributions of IOD and El Niño to tropical Pacific wind anomalies (Fig. 9, Supplementary Fig. 15).

Collectively, the experiments suggest how co-occurrence with a positive IOD helps an El Niño overcome its self-limiting dynamics. IOD-induced Pacific wind anomalies generate an eastern-ocean intensified oceanic response, which bears explanation: since IOD-induced Pacific wind anomalies are of the same sign throughout the basin (e.g., see Fig. 7), the oceanic Kelvin wave that they generate travels unhindered all the way to the eastern coast; the Rossby waves that consequently reflect off the eastern coast increase the eastern boundary response from that of the incoming Kelvin wave; this makes IOD-induced oceanic

variations eastern-ocean intensified, with SST and thermocline anomalies first appearing at the eastern boundary in mid-summer (Fig. 8e, f), and subsequently propagating westward; the IOD-induced wind anomalies also generate modest equatorial SST anomalies close to the dateline through zonal advection. Thus, co-occurrence with a positive IOD significantly enhances the SST anomalies generated by an El Niño: from boreal summer, the anomalies double in amplitude from the central to the eastern Pacific, doubling the rate of SST growth experienced during regular El Niños (Fig. 9c, f, g, h, i; Supplementary Figs. 12–15).

In turn, the co-occurrence with an El Niño helps a positive IOD enhance its downstream Pacific circulation. This is demonstrated by carrying out oceanic simulations that were forced by two different realizations of simulated IOD-induced Pacific wind anomalies. In the first realization, we forced the atmospheric model with composite Indian Ocean OLR anomalies for 1982 and 1997, with composite anomalies for 1994 and 2006 in the second. The difference between the two realizations was that in the first one, IOD and El Niño co-occurred, and in the second they did not. Although the magnitude of OLR variability (Supplementary Fig. 16) was similar in the two atmospheric simulations, the first set of wind anomalies drove an ocean response (Fig. 9i) that was significantly stronger than the second (Fig. 9h).

El Niño's influence on Indian Ocean convection likely helps a co-occurring positive IOD induce stronger Pacific wind anomalies. During boreal summer, IOD convective anomalies have strong meridional asymmetry about the equator (Supplementary Fig. 16c, 16f): suppressed convective anomalies, south of the equator, occur alongside moderately enhanced anomalies to its north[48]. The strongly asymmetric anomalies have a relatively weak symmetric component and thus relatively weak IOD-induced Pacific wind anomalies. However, during El Niño convection is suppressed over the Indian Ocean, with a stronger reduction over the northern Indian Ocean[48]. This effect greatly reduces the meridional asymmetry (Supplementary Fig. 16c, 16i) of OLR anomalies during boreal summer, and may explain why IOD induces stronger Pacific wind anomalies when it co-occurs with El Niño.

The IOD–ENSO interactions outlined above provide a plausible mechanism through which an El Niño develops into a super El Niño. This model not only explains why super El Niños are eastern-ocean intensified, but also their rapid growth in the boreal summer[8,15,35]; their rapid weakening[8] is likely explained by the observation that IOD-induced Pacific wind anomalies rapidly terminate or even reverse (Fig. 5f) after the peak phase of IOD.

## Discussion

In the rest of the article, we discuss the the 2015 El Niño which also appears to be a super El Niño[20] due to its extremely strong SST anomalies[49]. However, a closer examination shows a number of differences from the 1972, 1982, and 1997 El Niños.

First, eastern-ocean intensification is absent during the 2015 event (Fig. 10f, g)—climate anomalies during the 1997 El Niño, with which the 2015 event compares in terms of SST anomaly[49], offer a contrast (Fig. 10c, d).

Second, OLR anomalies during 2015—one of the weakest among the El Niños in the OLR record since 1979 (Supplementary Fig. 17, Fig. 4)—imply a weak coupling between SST and convective anomalies. In turn, the weaker convective anomalies imply weaker surface wind anomalies, as observed[49] (cf. Fig. 10a, e).

The weaker surface wind anomalies during 2015 must produce weaker equatorial SST anomalies by oceanic horizontal advection

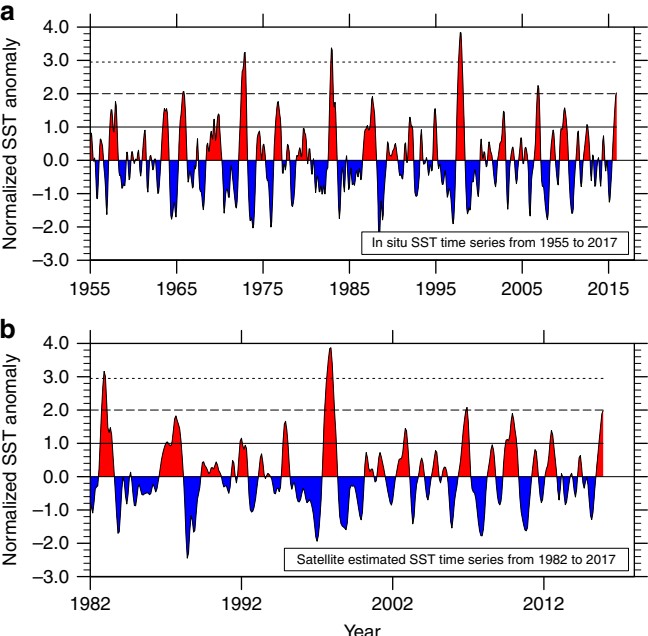

**Fig. 11** Amplitude of filtered monthly SST anomalies over the Nino3 region. Interannual Nino3 SST anomalies **a** from an in situ dataset (HadSST3)[58] for the period 1955 to 2015, and **b** from a satellite-based dataset (OISST)[55] since Jan 1982. Decadal anomalies were removed from the time series shown using a low-pass Lanczos filter with 49 weights. Only the 1972, 1982, and 1997 El Niño events have Nino3 SST anomalies that attain or exceed $3\sigma$, well separated from other events by $1\sigma$. The time series shown in **b** was padded with HadSST3[58] SST anomalies from Jan 1980 to Dec 1981, so as to prevent the super El Niño event during 1982 being removed from the analysis due to the end effects of the filter. Data after Nov 2015 were removed from **a**, **b**, because these are unreliable due to the filter's end effects. The data shown were normalized by using the standard deviation of the interannual anomalies from the respective datasets

than during 1982 and 1997. Similarly, the weaker sea surface height anomalies in the eastern Pacific (Fig. 10f) during 2015 must produce significantly smaller SST anomalies in that region—Xue and Kumar[49] note that eastern Pacific thermocline anomalies in 2015 were half of that during 1982 and 1997. Nevertheless, the SST anomalies observed during 2015 are of similar amplitude to those during 1982 and 1997 over most of the equatorial Pacific[49]. Further, its weak OLR anomalies pose a challenge in explaining the observed extreme SST anomalies in terms of the Bjerknes feedback alone.

These inconsistencies may be reconciled by the observation that the 2015 El Niño occurred during a period of rapidly rising decadal-scale SST anomaly worldwide, in particular over the tropical Pacific. Supplementary Fig. 18 reveals a prominent decadal, warming trend in globally averaged SST anomalies, following the hiatus[50] during 2004 to 2011. Although tropical Pacific decadal SST anomalies sharply trend downwards during the hiatus, they reverse course and rapidly trend upward from 2012, warming about 0.5 °C in the 3 years leading to the 2015 El Niño. Once decadal anomalies are accounted for, the SST anomaly amplitude (Fig. 11) of the 2015 event is 1 standard deviation lower than the super El Niños mentioned here.

Although DMI from monthly SST anomalies is weakly elevated during 2015[51], the actual anomalous conditions over the Indian Ocean are not consistent with a positive IOD event (Supplementary Fig. 20). Note that DMI was originally defined using interannual anomalies[26]. Further, Saji and Yamagata[27] showed that basin-wide SST anomalies induced by ENSO variations

should also be removed from SST anomalies before computing the DMI (see Methods)—an IOD index created as suggested by Saji and Yamagata[27] is shown in Supplementary Fig. 21, which confirms that prominent IOD variations were not present during 2015, consistent with the actual conditions shown in Supplementary Fig. 20.

So far as we can tell, the mechanisms elucidated here are consistent with available observations. However, they are based on simple models and hence should be regarded as preliminary until verified against more complex models. We hope that, since these mechanisms involve only the basic elements of equatorial dynamics, such future investigations will uphold the validity of our model. Further, due to data limitations, a less optimal Lanczos filter was used to remove decadal variations from recent data (Supplementary Fig. 19), which may leak in a fraction of interannual anomalies into the estimated decadal signal. However, the observed decadal trend post 2011 does not show the existence of any interannual variations, suggesting that our estimate of the decadal signal during 2015 is robust.

Our study reinforces the view that the Indian Ocean is an important factor for ENSO predictability: it is documented that the then available real-time prediction models were largely unable to predict the onset of tropical Pacific SST anomalies, until the onset had itself occurred, during the IOD years of 1994[52] and 1997[2,53] (Luo et al.[30] find the same with their retrospective coupled prediction experiments). Therefore, examining and improving the representation of IOD in the state-of-the-art coupled climate models is important. The failed predictions on average for the 2006 El Niño[45] (Supplementary Fig. 22) suggest that many forecast models may not adequately represent IOD–ENSO interaction as observed. To evaluate these processes, detailed analyses of forecast behavior in the western Pacific during IOD years must be undertaken. The 2006 IOD event may provide a benchmark for assessing how equatorial teleconnections associated with an IOD event are resolved in such models.

## Methods

**Data sources**. The daily and monthly resolution datasets of sea surface height (SSH), SST, OLR, marine cloudiness, and surface winds/stress are from AVISO[54], OISST[55], NOAA[56], ICOADS[57] and NCEP reanalysis[41], respectively. Monthly SST from the HadSST3 project[58] was also used.

**Data processing**. Daily and monthly anomalies were constructed by removing a mean climatology for the period 1982–2010 from the data. Since SSH is available only after 1993, its climatology was calculated over the period 1993–2014. The daily climatology was calculated as follows. First, an unsmoothed daily climatology was constructed from daily mean data. Thereafter, a smoothed daily climatology was constructed from the sum of the mean and the first three harmonics of the unsmoothed daily climatology. The daily anomalies are the difference of the daily mean data from the smoothed daily climatology.

**Climate indices**. All the indices described below were calculated from monthly anomalies. Prior to the calculations, the data were detrended and decadal anomalies, defined as periodicities lower than 7 years, were removed using Lanczos filtering. The Nino3 SST index was calculated by averaging SST anomalies in the eastern Pacific (5°S–5°N, 150°W–90°W). The Nino3.4 SST index is an areal average of SST anomalies over (5°S–5°N, 170°W–120°W). The DMI[27] was calculated as a difference of SST anomalies in the western (10°S–10°N; 60°E–80°W) equatorial Indian Ocean from the eastern (10°S–Eq; 90°E–110°E) equatorial Indian Ocean.

A basin-wide SST anomaly induced by ENSO[27,59] was removed prior to calculating the DMI. To estimate the basin-wide anomaly, we regressed Indian Ocean SST anomalies with Nino3.4 index at various lags. It was found that the regression maximized and were positive when the former lagged Nino3.4 by several months (ranging from 3 to 6 months depending on location[27,59]). Subtracting the lagged anomalies effectively removes warm Indian Ocean SST anomalies that lag El Niño by several months (also cool SST anomalies that lag La Niña). El Niño also induces cool SST anomalies off Java during late fall to early winter[27]—this teleconnection is not affected by the procedure employed[33].

**Westerly wind bursts**. WWB events were detected from daily surface zonal wind anomalies. Prior to the analysis, we removed interannual anomalies from the daily

anomalies using a low-pass Lanczos filter that retained only periodicities longer than 100 days. WWB events were detected from the residual high-frequency daily anomalies, using an intensity threshold of 4 m s$^{-1}$—for simplicity, we did not apply a threshold in event duration.

**The oceanic simulations**. The oceanic simulations were conducted using the ocean component of the GModel-3.0[36,47,60], developed by Gerrit Burgers at the Netherlands Center for Climate Research. It consists of a linear, 1.5-layer (shallow-water) model, which simulates the first-baroclinic-mode response of the ocean, as well as a linear, empirical SST equation.

The model equations describe the thermocline depth field $h$, the zonal velocity field $u$, and the meridional velocity $v$ of a shallow-water layer of mean thickness $H$ and density $\rho$ over a motionless bottom layer of density $\rho + \delta\rho$.

$$\frac{\partial u}{\partial t} - fv + g'\frac{\partial h}{\partial x} + F_M(u) = \tau_x, \qquad (3)$$

$$\frac{\partial v}{\partial t} + fu + g'\frac{\partial h}{\partial y} + F_M(v) = \tau_y, \qquad (4)$$

$$\frac{\partial h}{\partial t} + H\left(\frac{\partial u}{\partial x} + \frac{\partial v}{\partial y}\right) + F_H(h) = 0, \qquad (5)$$

the wind stress forcings are $\tau_x$ and $\tau_y$, $f = \beta y$ is the Coriolis parameter in the equatorial beta-plane approximation, $g' = g\rho^{-1}\delta\rho$ is the reduced gravity, and $F_M(u)$, $F_M(v)$ and $F_H(h)$ are frictional terms. The value of the shallow-water wave speed $C_o = \sqrt{g'H}$ was 2.5 m s$^{-1}$, and the mean depth $H$ was set to 150 m. The frictional terms consist of several parts. The main part is the harmonic part, with an eddy viscosity of $2 \times 10^4$ m$^2$ s$^{-1}$ in $F_M$ and an eddy diffusivity of $2 \times 10^3$ m$^2$ s$^{-1}$ in $F_H$. A linear damping term, which is only sizeable near the northern and southern boundary of the basin, prevents waves propagating along these boundaries. Further, small, fourth-order terms are used to suppress short wavelength numerical instabilities near the equator.

A simplified SST equation is used in the model to simulate equatorial Pacific SST anomalies associated with ENSO. The formulation considers SST anomaly tendency due to thermocline anomalies $h$ (thermocline feedback) and that due to advection of mean zonal temperature gradients by anomalous ocean zonal currents (zonal advection feedback). The linear equation for SST anomalies ($T$) has the following generic form:

$$\frac{dT}{dt} = \alpha(x)h(x,y) + \beta(x)\tau_x(x,y) - \gamma(x)T(x,y). \qquad (6)$$

The first term on the right hand side models the thermocline feedback by a term linear in the thermocline anomaly. However, the efficiency of this feedback is affected by the mean thermocline depth, which is much shallower at the eastern than at the central and western Pacific. To account for this, the coefficient $\alpha$ is a function of longitude, increasing in strength from the central towards the far-eastern Pacific (see Fig. 1 of Burgers and van Oldenborgh[36]). In the central Pacific, the zonal advection feedback is important[42]. In the ocean model, this feedback is modeled by a term linearly related to the zonal wind stress $\tau_x$. The reason why $\tau_x$ is used, instead of model simulated zonal current anomalies ($u$), is because the representation of $u$ by a 1.5-layer shallow-water model is rather poor, even with the addition of an Ekman layer. Further, observational evidence has suggested that an empirical linear relation between the surface wind and observed zonal surface velocities in the ocean is slightly more accurate than that between the observed zonal surface velocity and the velocity of the 1.5-layer linear model[36]. The coefficient $\beta$ for this term also depends on longitude and reaches a maximum in the central Pacific. The last term in the SST tendency equation is a simple relaxation term that models negative feedbacks, such as anomalous surface heat fluxes. Burgers and van Oldenburgh[36] tuned the factors $\alpha$, $\beta$, and $\gamma$ empirically, such that monthly observed SST anomalies along the equator over the period 1982–1999 was reproduced best, when the model was forced by Florida State University pseudostress anomalies[61]. Note that all of the coefficients in the SST Eq. (5) all depend on longitude (their longitudinal variations are depicted in Fig. 1 of Burgers and van Oldenborgh[36]). Ideally, these should also vary as a function of latitude. However, for simplicity such a dependence was not incorporated, considering that the focus of this investigation is confined to SST variations close to the equator; simulated and observed Nino3 anomalies for the period 1966 to 1999 are shown in Supplementary Fig. 23, and the correlation between simulated and observed SST anomalies, and its sensitivity to the $\alpha$ and $\beta$ terms in Eq. (6), are shown in the Supplementary Fig. 24.

The ocean simulations were always started from a state of rest, and were forced with monthly surface wind stress anomalies. The model has a limited domain, with a grid spacing of 2° in the zonal and 1° in the meridional direction, and with closed boundaries at 30°S and 30°N and realistic closed western and eastern boundaries. The western boundary and eastern boundary follow approximately the coastlines of Australasia and America. The model resolution is relatively coarse. However, here we focus on the ocean's response of relevance to ENSO, which is largely accounted

for by the equatorial Kelvin wave and the first meridional mode Rossby wave with meridional decay scales of the order of $\sqrt{C_o/\beta} \approx 330$ km. These are well resolved with a meridional grid spacing of 1°, which is about 3–4 times smaller than the meridional decay scale of the equatorial waves of interest. Also, at this resolution, there are no noticeable distortions by finite-difference effects to the the wave structure and zonal currents.

**The atmospheric simulations**. The atmospheric model, hereafter Linear Baroclinic Model or LBM[40], based on the primitive equations on a sphere, is a much simplified version of an Atmospheric General Circulation Model (AGCM)—the Center for Climate System Research (University of Tokyo), National Institute for Environmental Studies and Frontier Research Center for Global Change (CCSR/NIES/FRCGC) AGCM. The LBM is derived by linearizing the AGCM about a specified background state, $\psi_b(x, y, z, t)$, eliminating water vapor, and dropping the model's radiation scheme. The background state $\psi_b(x, y, z, t)$ was set to the seasonal climatology of the NCEP reanalysis data[41]. The diabatic heating that forces the atmosphere is represented by an externally prescribed function, $Q(x, y, z, t)$. The model employs a linear drag that mimics Rayleigh friction and Newtonian damping: the damping time scale was set at 1 day for the lowest three levels and the topmost two levels, 5 and 15 days for the fourth and fifth levels, and 30 days elsewhere. Biharmonic horizontal diffusion with a time scale of 4 h for the smallest-scale wave was also used.

The LBM was integrated at T42 horizontal resolution and with 20 vertical sigma levels. The atmospheric simulations are shown at monthly resolution: the model was forced with monthly diabatic heating anomalies $Q(x, y, z, t)$ that were kept unchanged throughout the model integration. The steady-state response to a prescribed $Q(x, y, z, t)$ was found using the time integration method; since the solutions did not change significantly after 8 days (also cf. Jin and Hoskins[44]), the solution at day 10 was considered as equivalent to the steady-state solution. During the LBM runs, $Q$ was switched on at the beginning of, and remained steady throughout, the integration. While $Q$ had an idealized form in the integrations discussed in Fig. 5d–f, for the rest of the LBM experiments its horizontal distribution was estimated from OLR anomalies: the OLR anomaly was converted to a rainfall anomaly using an empirical conversion factor of −6.0 mm per day for 1 W m$^{-2}$ of OLR variation; further, a rainfall rate of 10 mm per day was considered equivalent to a column-averaged diabatic heating rate of 2.5 K per day as in Jin and Hoskins[44].

**Code availability**. The atmospheric model and its source code are available from The University of Tokyo (http://ccsr.aori.u-tokyo.ac.jp/hiro/sub/lbm.html). The source code for the ocean model is available from Dr. Gerrit Burgers (burgers@knmi.nl).

**Data availability**. The observational datasets used in this analysis are publicly available and were downloaded (except for SSH and SST from HadSST3) from NOAA-ESRL Physical Sciences Division, Boulder Colorado from their website at https://www.esrl.noaa.gov/psd. The SSH data are available from CMEMS (http://marine.copernicus.eu). The HadSST3 is available from the Met Office Hadley Center (https://www.metoffice.gov.uk/hadobs/hadsst3/). Data for the real-time Nino3.4 forecast plumes can be downloaded from IRI (http://iri.columbia.edu/forecast/ensofcst/Data/). The model simulations are available from the corresponding author upon request.

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

## Acknowledgements

We thank Ms. K.-H. An and Drs. C. Ummenhoffer, M. K. Davey, J. McCreary, N. Schneider, D. Sengupta, and T. Yamagata for their critical inputs, Drs. M. Watanabe and G. Burgers for the model codes, and Drs. Anthony Barnston and Simon Mason for the Nino3.4 prediction plumes. We thank Professor John Blake (University of Aizu) for his generous help and advice with the language editing. The figures, including those containing map outlines, were made with the NCAR Command Language (Version 6.4.0) (see https://doi.org/10.5065/D6WD3XH5).

## Author contributions

S.N.H. conceived the experiment(s) and wrote the paper; V.T. and D.J. conducted the experiment(s); S.N.H., D.J., and V.T. analyzed the results. All authors reviewed the manuscript.

## Additional information

**Competing interests:** The authors declare no competing interests.

