## [Peer Review File · Nature Communications]

Reviewers' comments:

Reviewer #1 (Remarks to the Author):

The IOD's influence on the tropical Pacific as well as on the ENSO has already been discussed in several previous studies. I could not find any substantial new findings in this study to recommend its publication in the Nature Geoscience. Also, I find that the discussions are somewhat cursory. For example, authors have neither used an objective criterion to define a super El Nino nor tried to clarify why a particular year is so different from others. Why the season of October-December was used to compare SST anomalies associated with 1997-98 and 2015-16 events? Could only SST anomaly define an El Nino and distinguish an El Nino from a super El Nino? What was the contribution of the subsurface? Are subsurface heat content and vertical advection not necessary factors to be considered in the development of El Nino?

Why the El Nino of 2015-16 is not considered as a super El Nino? The rainfall and SST anomalies of that year as argued in the manuscript are not so significantly different from other recent El Ninos that were accompanied with IODs. Other studies attribute decadal and other regional climate variabilities to explain that difference in the observed anomalies. Authors need to discuss those and separate IOD's influence from others for clarity of the concept.

The discussions on why the 1994 and 2006 IODs could not lead to super El Ninos of those years are not convincing. If the co-existence of IOD in those years did not contribute to their super growth, perhaps then IOD is not a real factor in the process of an El Nino reaching the stature of super El Nino (again the status is not defined objectively in this study to clearly argue the mechanism). At least, the arguments presented here are not convincing enough to accept the hypothesis.

Page 2, The meaning of sentence "Anomalous flows develop..." is not clear. Please rewrite. The manuscript reads well but some improvements are necessary in the writing style.

Reviewer #2 (Remarks to the Author):

In this article the authors argue that the state of the Indian Ocean Dipole (sea surface temperatures [SST] in the east and west Indian Ocean) influences the atmospheric circulation over the equatorial Pacific and thus influences the evolution of El Nino events, possibly enhancing some such events to become 'super El Ninos'. They use a mixture of analysis of observations and experiments with a dynamical atmospheric model.

The idea is a good one, and the topic is likely to be of considerable interest to climate scientists. However the evidence presented is not easy to understand, partly because explanatory detail is missing. The presentation also assumes the reader has substantial knowledge of the field.

Specific comments:

Introduction:

Paragraph 2: The 2015/16 El Nino is widely referred to as a 'super El Nino', but the authors classify this as 'strong' but not 'super' on the basis of the large near-equatorial SST anomalies not extending to the South American coast. The authors need to make clear from the outset (and in the abstract) that their definition refers to a particular sub-type. It would help to cite more examples of 'super El Ninos' (1982/83, other?).

Is extension to the South American coast a distinguishing feature of 'super' events?

(Is the classification of events as 'Central' and 'East' Pacific type relevant here?)

Para 3: outgoing longwave radiation (OLR) is used as a proxy for rainfall: it should be stated that negative OLR anomalies are associated with positive rainfall anomalies, and vice versa.

Para 6: it could be mentioned that while forecasts strongly favoured development of a large El Niño in 2014, a minority of forecasts (e.g. in dynamical system ensembles) were neutral.

Para 7: it would be helpful to define the Indian Ocean Dipole here (e.g. as an east-west contrast in Indian Ocean SST anomalies; with the index positive when e.g. west is warmer and east is cooler than usual).

Results:

State dependence: it is well known that the actual state of the system influences its evolution, and well understood that 'at rest' idealised models offer important but limited insight, so para 1 could be omitted.

Para 2: more information should be provided about the atmospheric model: e.g. does it include rainfall? How is the basic state specified: e.g. as a relaxation to re-analysis u,v,T seasonal climatology?

Para 3: vertical distribution of heating anomaly: is this $\sin(\sigma)$? Comment on the lack of surface heating (which can be important in some regions) in this formulation.

Figure 3a: the contour labels are not legible.

Fig. 3B: the base state seems to have no easterlies in the central-east Pacific, contrary to observed wind climatology. This needs explanation, as it casts doubt on the ability of the 'base simulation' to represent observed conditions. (Perhaps the observed profile should be included for comparison?) It also casts doubt on the ability of the model to represent wind anomalies in the cold tongue region. The 'thermocline feedback' axis label does not fit the 'zonal wind' caption - perhaps there is an error in this figure?

Page 4, below Fig. 4: need to explain what is meant by 'thermocline effect'. Does the simulated reversal of zonal winds by El Niño east of 150W fit observed behaviour?

Fig. 4: although the use of squared OLR anomalies is explained eventually, it is easy to misunderstand this figure. Perhaps label the axis as OLR anomaly amplitude, or RMS OLR anomaly? And start the caption with 'Amplitude of convective variations...'.

The impact of IOD on surface winds

Para 2: explain why amplitude is used in Fig. 4. The sign of the anomaly is also important: why not just average anomalies over smaller indicative areas?

Fig. 5: need to explain 'partial linear regression controlling for Niño3' (what is the motivation and implication of such a control?), and need to define the Dipole Mode Index. Also make clear that brown colours mean less cloudiness in a,b,c (by inference, less rainfall) and less atmospheric heating in d,e,f; while green indicates more rainfall (heating).

Para 4: make clear the synthetic thermal forcing in Fig. 5def is intended to represent the convection patterns in Fig 5abc.

Below Fig. 7: explain that Kelvin waves tend to induce easterly surface winds to the east of near-equatorial atmospheric heating.

Page 7 para 4: the evidence described does not exclude westerly wind bursts from having a role in 2006.

Page 8 para 1: while the evidence supports IOD influence on atmospheric variability over the equatorial Pacific, an impact on oceanic variability is an inference.

The 'model for super El Niños' seems somewhat hypothetical, and relies on the IOD influence on the central Pacific winds. The 2006 case study is suggestive of such influence, but more cases would be

welcome. Fig. 5 is not very convincing of such influence.

Overall: I recommend the authors submit a revised version.

Reviewer #3 (Remarks to the Author):

Review of a manuscript, entitled "A model for super El Niños" by Hameed et al.

In this paper, the authors investigate the dynamics of super (very large amplitude, eastern ocean intensified) El Niños. Their primary tool is to obtain solutions to an atmospheric model. Forcing is by externally prescribed heating sources, $Q(x,y)$, that are spread sinusoidally throughout the troposphere and have a horizontal structure similar to those for ENSO and IOD events. Further, the model allows a background state, $\psi_b(x,y,t)$, to be externally prescribed, so that its solutions are the anomaly fields with respect to ψ_b that are driven by Q . The authors vary ψ_b from the observed, climatological annual cycle to a state that includes developing El Niños of various strengths.

One problem I have with the paper is that the approach in the previous paragraph is not well described. (I had to read the paper several times before I figured out what is going on, and even now I am not completely sure what I wrote above is correct.) So, I think the paper the Methods section must immediately follow the Introduction. There, details of the anomaly model must be clearly described. For example, is the model linearized about ψ_b , or are some nonlinear terms retained? My guess is that it must be linearized. Also, in this section the concept of "state dependence" must be clearly defined (perhaps similar to what I wrote above).

A conceptual problem with their approach (if what I wrote in the first paragraph is correct) is that they wish to investigate the response due to ENSO (and IOD) forcing Q but at the same time build in the ENSO response by including it in ψ_b . The logic of this approach seems circular to me. It seems like a way to introduce the nonlinear effects into the system through ψ_b , while at the same time keeping the anomaly model linear. Wouldn't it be more logical to run a nonlinear version of the model without any background state and simply vary the strength of Q ?

Another issue is that, although the title and introduction of the paper suggest that it will explain the cause of super El Niños, there is no solution reported for a super El Niño (the solutions reported in the Results section are only for ordinary El Niños and IOD events). All discussion of super El Niños takes place in the Discussion section, where only a hypothesis about the cause of super El Niños is presented: They occur because of ocean-atmosphere feedbacks enhanced by the simultaneous occurrence of ENSO and IOD events. Apparently, then, their modeling approach cannot generate super El Niños, so their dynamics can only be inferred from their ENSO and IOD solutions. Thus, their paper does not really determine the underlying dynamics that cause of super El Niños, only presents a plausible hypothesis. In any case, the text about super El Niños in the Discussion section should be moved to a final Results subsection.

Figure 3 is confusing. Figure 3(a) describes $\psi_b(x,y,t)$ for the equatorial winds. Are they averaged over a latitude band, like 2S to 2N? Since ψ_b depends on t , what time is shown in the figure? In Figure 3(b), to what does the "Thermocline feedback" axis refer? I don't think you defined it. In Figures 3(b) and 3(c), I guess the dotted and solid curves are from solutions forced by the same Q but different ψ_b . Perhaps state that explicitly. What are the values of the ENSO state in Figure 3(a) for the ψ_b 's that you used?

Given these problems, I cannot recommend publication of the manuscript in its present form. It may be that an improved version of the paper, which clearly addresses the above issues, will be publishable. Certainly, some of the issues can be addressed through a modest reorganization (modification) of the text. Others are more serious.

Response to Reviewers

Re: Reviews for NCOMMS-17-05765 "A model for super El Ninos"

Please accept our sincerest thanks for the reviewers comments on our manuscript. We have modified the paper in response to the extensive and insightful reviewer comments. We have added additional experiments using an ocean model to fully address the reviewers comments. Furthermore we have extensively rewritten sections of the manuscript and added supporting data as Supplementary information. We hope that these comply with the referees remarks. We will respond to the referee comments point by point below this. To distinguish the referee comments from our response, the former are enclosed within a frame and the text is typed with italics on a gray background.

Response to Reviewer #1

We thank the reviewer for the careful perusal of our manuscript and for the critical comments.

1. The IODs influence on the tropical Pacific as well as on the ENSO has already been discussed in several previous studies. I could not find any substantial new findings in this study to recommend its publication in the Nature Geoscience.

Thank you for the comment. We believe that you have raised an important issue, by pointing out that originality of our new findings is unclear.

You are right in saying that a number of studies have tried to link IOD's influence on the ENSO. Among these, Saji and Yamagata¹ compared the evolution of Nino3 SST anomalies for El Ninos that co-occurred with IOD and those that did not. They found that the El Ninos that co-occurred

had larger amplitude and faster buildup and termination. Apart from these, the major studies that have explored this connection are Behera et al (2006)² and Luo et al (2010)³. Both are coupled model investigations and the conclusions regarding IOD's influence on ENSO are based purely on model simulations. In particular, the former noted that the periodicity of ENSO was changed when an active Indian Ocean was included. The latter one performed prediction experiments and concluded that an active Indian Ocean was required in their model to simulate the onset of the El Nino events in 1994, 1997 and 2006.

There is one more important study, investigating the lagged impact of IOD on the following year's ENSO⁴. Here our concern is on the concurrent impact of an IOD.

While these studies are important, they only provide indirect evidence for the impact of IOD on the Pacific. This is either because the evidence is from data analysis, from which the effect of IOD and ENSO are hard to separate in a universally convincing manner. Or in the case of Behera et al (2006) and Luo et al (2010), the evidence is an interpretation of results from a single model. Further since both the above did not perform any detailed analysis of the dynamical mechanisms underlying such a perceived impact, it is not clear whether the teleconnections of the model simulated IOD are the same as that operates in the real-world, or whether the results in their sensitivity experiments are really due to the effect of the IOD simulated in the model and not due to other factors such as the Indian Monsoon.

We think that the issue of clearly separating IOD's influence on ENSO (or the converse) is a harder problem than the reviewer tends to believe. So we beg to differ with the reviewer's judgement that no substantial new findings are present in this study. In our opinion, one of the substantial new findings of the paper is the unambiguously clear demonstration of the impact of the 2006 IOD in forcing El Nino like SST anomalies over the Pacific. The rest of the results are simply a logical extension of this finding.

In the original manuscript, we left it at the place where the logical extension of this result was put forth as a hypothesis. However we realized that it is important to demonstrate the key role of IOD for super El Ninos in a more careful and tangible manner. In the revised manuscript we have used

an ocean model to extend the results and show that IOD does indeed play a key role in generating super El Ninos.

We hope that you will find our extended results illuminating and convincing.

2. Also, I find that the discussions are somewhat cursory.

We are sorry that we have given such an impression. We had tried our best to tie down any statements to an observational fact, dynamical reasoning or simulated results, but have obviously failed in this.

Following on your critique, we have worked hard to make sure that such cursory discussions are not present in the revised version.

3. For example, authors have neither used an objective criterion to define a super El Nino nor tried to clarify why a particular year is so different from others. Why the season of October-December was used to compare SST anomalies associated with 1997-98 and 2015-16 events?

Thank you for the critique. Yes, we did not ourselves develop an objective criterion to define a super El Nino. This was because such an objective definition was already available from the work of Hong et al (2015)⁵ and we had simply used the years found in their study. Further the same years appear in the independent analysis of Takahashi and Dewitte⁶, and further are discussed for their extraordinary spatio-temporal evolution in McPhaden (1999)⁷ and Cai et al (2014)⁸.

Thus the concept of super El Nino has been around at least since 1999, and we thought it was not necessary to strictly (re)define a super El Nino by ourselves. In the revised manuscript, we have taken care to provide a more detailed discussion of what we mean by a super El Nino, based on the above studies. We also think that the concept of super El Nino will be an evolving one, and our study will contribute to further refining the concept. Further, we have also discussed

in more detail the differences between the 1997-98 and 2015-16 events (see revised Fig. 1 and accompanying discussion). This discussion, along with the underlying mechanisms revealed here, will help crystallize the concept of super El Nino further.

The season of October-December is around the peak of El Nino. Our original intention in showing the variations during this period was to include a discussion of the state of IOD. Since the IOD rapidly vanishes after boreal fall, using a season around boreal fall was helpful to show both the state of IOD as well as El Nino during those years.

4. *Could only SST anomaly define an El Nino and distinguish an El Nino from a super El Nino? What was the contribution of the subsurface? Are subsurface heat content and vertical advection not necessary factors to be considered in the development of El Nino?*

We think this is an excellent question, and it was exactly our point that SST anomaly alone should not be used to distinguish between El Ninos. We have enlarged upon this point in the revised version to show that despite similar amplitude of SST anomaly during 1997 and 2015, the structural differences lead to the effects of both being vastly different.

Using SST alone would lead to classifying events such as 1994 and 2006 as El Nino years. However as shown, the atmospheric variations, especially convective variations (and therefore impact on weather and climate) during these years are vastly different from regular El Ninos.

5. *Why the El Nino of 2015-16 is not considered as a super El Nino? The rainfall and SST anomalies of that year as argued in the manuscript are not so significantly different from other recent El Ninos that were accompanied with IODs. Other studies attribute decadal and other regional climate variabilities to explain that difference in the observed anomalies. Authors need to discuss those and separate IODs influence from others for clarity of the concept.*

We thank you for this critical question. We have explained in the revised manuscript exactly how 2015-16 differs from other super El Ninos and based on this we have suggested that this El Nino cannot be considered as a super El Nino. We have also briefly discussed the role of anthropogenic warming on 2015-16 based on a recent work⁹.

Since this is an important question, we also provide a somewhat detailed response below. There is in fact considerable differences in the rainfall structure of the 1997 and 2015 El Nino events, especially their eastward extension and strength over the eastern Pacific. These may seem minor to the reviewer, but the 1997 El Nino costed billions of dollars in damage for California and Peru. Based on the large amplitude SST anomaly of the 2015 El Nino, it was widely expected that it would have a similar impact on the eastern Pacific as the 1982 and 1997 El Nino. The immense destruction associated with the 1972 El Nino on the eastern Pacific is also widely noted¹⁰⁻¹². These led to flood preparedness actions being organized in advance in California and Peru^{13,14}. However this turned out to be a false alarm. As noted by Cohen¹³, Southern California was plagued by heatwaves and wildfires instead of the much anticipated deluges, while Seattle which was expecting a worsening drought, instead had the wettest winter on record.

6. The discussions on why the 1994 and 2006 IODs could not lead to super El Ninos of those years are not convincing.

We are sorry, but maybe the reviewer misinterpreted some of our analysis. In fact, we did not include any discussion about why 1994 and 2006 IODs could not lead to super El Ninos. We did discuss the impact of the 2006 IOD on the Pacific using observational analysis and numerical simulations. Based on this we clearly demonstrated that the transient El Nino-like conditions during 2006 were driven by IOD teleconnection, and that these conditions did not arise from coupled air-sea interactions within the Pacific Ocean.

7. If the co-existence of IOD in those years did not contribute to their super growth, perhaps then IOD is not a real factor in the process of an El Nino reaching the stature of super El Nino (again the status is not defined objectively in this study to clearly argue the mechanism). At least, the arguments presented here are not convincing enough to accept the hypothesis.

We thank you for the critical question. But perhaps this is a misunderstanding of our analysis and results. The main crux of our argument was the combined effect of IOD and ENSO. The years 1994 and 2006 are not years where IOD and ENSO coexisted, in the sense that the transient signals over the Pacific were forced by IOD and not due to the coupled air-sea interactions within the Pacific basin.

However with pre-existing El Nino (understood here as a coupled mode arising out of ocean-atmosphere interactions over the Pacific) conditions present, the additional forcing provided by IOD forcing can transform the El Nino into a super El Nino.

8. Page 2, The meaning of sentence Anomalous flows develop is not clear. Please rewrite. The manuscript reads well but some improvements are necessary in the writing style.

Thank you for the comment. We have spent considerable time to make sure that such unclear sentences are not present in the revised manuscript.

We agree that there were certain shortcomings in our original presentation. Especially, since we only relied on atmospheric modeling, we could not demonstrate the impact of IOD on the equatorial Pacific ocean in a convincing manner. We sincerely thank you for the critical comments that gave us an opportunity to think hard about our work and its limitations. In the revised version, we have made great efforts to overcome these shortcomings, and we hope that you will find that our central thesis is significantly augmented by the added simulations and analyses.

Response to Reviewer #2

First of all, we sincerely appreciate the time and energy that you took out to go over the manuscript with such care and to provide your thoughtful and at the same time critical comments.

1. *In this article the authors argue that the state of the Indian Ocean Dipole (sea surface temperatures [SST] in the east and west Indian Ocean) influences the atmospheric circulation over the equatorial Pacific and thus influences the evolution of El Nino events, possibly enhancing some such events to become 'super El Ninos'. They use a mixture of analysis of observations and experiments with a dynamical atmospheric model.*

The idea is a good one, and the topic is likely to be of considerable interest to climate scientists. However the evidence presented is not easy to understand, partly because explanatory detail is missing. The presentation also assumes the reader has substantial knowledge of the field.

We are delighted to know that you appreciated our idea, and thought that the topic will be of considerable interest to the climate community. This has encouraged us to work even harder to straighten out problems with the current manuscript and to consolidate the main arguments with additional data.

We agree that the original presentation missed crucial explanatory detail. Following on your detailed advice, we have tried our best to correct this particular flaw in the revised version.

2. *Para 2 - The 2015/16 El Nino is widely referred to as a 'super El Nino', but the authors classify this as 'strong' but not 'super' on the basis of the large near-equatorial SST anomalies not extending to the South American coast. The authors need to make clear from the outset (and in the abstract) that their definition refers to a particular sub-type. It would help to cite more examples of 'super El Ninos' (1982/83, other?). Is extension to the South American coast a distinguishing feature of 'super' events? (Is the classification*

of events as 'Central' and 'East' Pacific type relevant here?)

Thank you for the comment and the suggestions on how to improve this discussion. As per your suggestion, we clarify in the introduction as well as the abstract, that we consider a particular subtype where the characteristic feature is the maxima in SST anomaly at the eastern Pacific. This aspect of super El Nino is based on the previous studies of Cai et al⁸ and Hong et al⁵.

Hong et al⁵ find using cluster analysis that 1972, 1982 and 1997 belong to the class of super El Ninos - a distinguishing feature of this class is the strong SST variations at the east Pacific. A similar conclusion on the uniqueness of these three El Ninos can be found in Takahashi and DeWitte⁶.

Historically the three (super) El Ninos mentioned above, have played a key role in bringing awareness of the phenomenon to the scientific community, to governments and common people, due to their enormous impacts. One of the motivations for this study was the coincidence of the three super El Ninos with the Indian Ocean Dipole (IOD).

We have extensively rewritten the Introduction to bring this historical perspective into light.

3. Para 3: outgoing longwave radiation (OLR) is used as a proxy for rainfall: it should be stated that negative OLR anomalies are associated with positive rainfall anomalies, and vice versa.

Thank you. We have done this in the revised manuscript.

4. Para 6: it could be mentioned that while forecasts strongly favoured development of a large El Nino in 2014, a minority of forecasts (e.g. in dynamical system ensembles) were neutral.

Thank you. We have added this (see line 79, pg. 4).

5. *Para 7: it would be helpful to define the Indian Ocean Dipole here (e.g. as an east-west contrast in Indian Ocean SST anomalies; with the index positive when e.g west is warmer and east is cooler than usual).*

Thank you for the kind suggestion. We have incorporated this in the revised version (starting line 89, pg. 4). The Dipole Mode Index (DMI) first appears in Fig. 6. We have added the definition of DMI to the figure caption.

6. *State dependence: it is well known that the actual state of the system influences its evolution, and well understood that 'at rest' idealised models offer important but limited insight, so para 1 could be omitted.*

Yes, we agree and have omitted the paragraph. Also, we have decided to remove the computations and discussions related to State dependence, following on Reviewer #3's critique. For the purposes of this manuscript, the self-limiting ENSO dynamics can be explained in terms of the atmospheric Kelvin wave response to ENSO convection, and the seasonal cycle of SST cooling at the equatorial eastern Pacific from boreal summer to fall. We think that state-dependence plays a second-order role, and we further suspect that this factor may be important in providing an explanation for the occurrence of central Pacific El Nino/El Nino Modoki events. We plan to work on this aspect, after implementing an algorithm known as the Trajectory Piecewise Linear (TPWL) scheme of Rewiński^{15,16} to approximate the non-linearity associated with this problem.

7. *Para 2: more information should be provided about the atmospheric model: e.g. does it include rainfall? How is the basic state specified: e.g. as a relaxation to re-analysis u, v, T*

seasonal climatology?

We have provided a brief description about the atmospheric model in the revised manuscript (starting line 212, pg. 9). More details including the integration scheme are mentioned in the Methods section.

The model equations (primitive equations) are linearized about the basic state. So they enter into the terms of the linearized equations directly. The basic state from the NCEP Reanalysis data is specified as an input to the model, along with the thermal forcing.

No, the model does not have a precipitation scheme. It is just a dynamical core forced by a prescribed thermal forcing. When a resting state atmosphere is prescribed as the basic state, the model reproduces the Gill solutions (e.g. Supplementary Fig. 4)

8. *Para 3: vertical distribution of heating anomaly: is this $\sin(\sigma)$? Comment on the lack of surface heating (which can be important in some regions) in this formulation.*

Yes, you are right. This is $\sin(\sigma)$. In the revised version we have commented on the lack of surface heating in this formulation (starting line 218, pg. 9).

9. *Figure 3a: the contour labels are not legible. Fig. 3B: the base state seems to have no easterlies in the central-east Pacific, contrary to observed wind climatology. This needs explanation, as it cast doubt on the ability of the 'base simulation' to represent observed conditions. (Perhaps the observed profile should be included for comparison?) It also casts doubt on the ability of the model to represent wind anomalies in the cold tongue region. The 'thermocline feedback' axis label does not fit the 'zonal wind' caption - perhaps there is an error in this figure? Page 4, below Fig. 4: need to explain what is meant by 'thermocline effect'.*

This figure was removed. The discussion on the impact of IOD is now made using an ocean model simulation. The ocean model combines a shallow-water equation ocean model and a linear SST equation. A brief description of the ocean model is on page 5 (starting line 112) and details are provided in the Methods section.

10. *Does the simulated reversal of zonal winds by El Nino east of 150W fit observed behaviour?*

The original inspiration for considering state-dependence was based on the observation that there are considerable easterly winds at the eastern Pacific during El Nino. We have added a new figure (Fig. 3) in the revised version to demonstrate this. As far as we know, the impact of these easterlies are not considered to be important. We show for the first time using an ocean model simulation (Fig. 4) that these winds have an important role in determining the structure of SST anomaly during regular El Nino years. Not only do they explain the weakening of SST variance at the far east Pacific, but the curl associated with these also explain the warm (high) SST (SSH) anomaly extending west from the south of Baja California and the equator.

11. *Fig. 4: although the use of squared OLR anomalies is explained eventually, it is easy to misunderstand this figure. Perhaps label the axis as OLR anomaly amplitude, or RMS OLR anomaly? And start the caption with 'Amplitude of convective variations...'*

Thank you for this suggestion. We have revised the figure labels and added more details in the figure caption.

12. *Para 2: explain why amplitude is used in Fig. 4. The sign of the anomaly is also important: why not just average anomalies over smaller indicative areas?*

We have added an explanation about this in the figure caption. Yes, we agree that the sign of the anomaly is important. Both ENSO and IOD are associated with OLR anomalies that do not have the same sign throughout the domain. Taking the squared amplitude lets us capture the strength of both phases using a single index.

Interestingly, the structure of anomalies (for e.g. its asymmetry/symmetry) also plays an important role (see Supplementary Fig. 11 and accompanying discussion).

13. *Fig. 5: need to explain 'partial linear regression controlling for Nino3' (what is the motivation and implication of such a control?), and need to define the Dipole Mode Index. Also make clear that brown colours mean less cloudiness in a,b,c (by inference, less rainfall) and less atmospheric heating in d,e,f; while green indicates more rainfall (heating).*

Thank you for the suggestions. We have explained this particular terminology related to multiple regression analysis in the revised manuscript (starting line 207, pg. 8). We have also clarified the coloring scheme for cloudiness and its interpretation in the caption of Fig. 6 of the revised manuscript).

14. *Para 4: make clear the synthetic thermal forcing in Fig. 5def is intended to represent the convection patterns in Fig 5abc.*

Yes. Thank you. We have done this (Fig. 6 of revised manuscript).

15. *Below Fig. 7: explain that Kelvin waves tend to induce easterly surface winds to the east of near-equatorial atmospheric heating.*

Thank you. We have added this explanation to the figure caption below Fig. 7.

16. *Page 7 para 4: the evidence described does not exclude westerly wind bursts from having a role in 2006.*

The role of westerly wind bursts during 2006 is mentioned in McPhaden (2008)¹⁷. This was (presumably) to account for the unusual lack of air-sea coupling over the Pacific during the 2006 event. However the attribution is based on the presence of a number of wind events in their longitude-time section of surface winds and OLR anomalies. The magnitude of these were not estimated by the authors. For this paper, we did such a calculation, measuring the contribution from both intraseasonal events as well as westerly wind bursts. While we did not detect any westerly wind bursts (definition in the Methods section), intraseasonal variations accounted only for a negligible 7%.

The dominance of low-frequency components in the surface wind anomalies is also clear in Fig. 7c. Note that daily wind anomalies (low passed to remove weather scale fluctuations) was used in this figure.

17. *Page 8 para 1: while the evidence supports IOD influence on atmospheric variability over the equatorial Pacific, an impact on oceanic variability is an inference.*

Yes. When we worked on the original manuscript, we did not have access to an ocean model, and we were thus compelled to leave this as an inference. However having carefully gone through the editorial and reviewer comments, we decided to demonstrate the IOD impact on the equatorial Pacific using an ocean model. We have now replaced all such inferences with simulated ocean behavior.

18. *The 'model for super El Ninos' seems somewhat hypothetical, and relies on the IOD*

influence on the central Pacific winds. The 2006 case study is suggestive of such influence, but more cases would be welcome. Fig. 5 is not very convincing of such influence.

You are right. The “model for super El Ninos” was presented as a hypothesis, due to our not having an ocean model simulation at the time of the original submission. Having added an ocean component to our simulation, we are able to explore in greater detail how the IOD impact on Pacific surface winds affect the equatorial Pacific ocean.

Originally we had hoped that we made a convincing case about the impact of the 2006 IOD on the equatorial Pacific, and that we could use this as a strong argument in support of our hypothesis. With the ocean model experiments, we can now confirm that the IOD does indeed play a key role in super El Nino evolution. Further the ocean model is also used to demonstrate the self-limiting dynamics of ENSO.

Thank you for suggesting that we should add more cases like 2006. We have added the 1994 case study in the Supplementary Information. However this is also discussed in Luo et al 2010³, on the basis of the coupled prediction experiments. It is more difficult to do this from historical observations because of sampling issues.

Once again, thank you for giving us the opportunity to strengthen our manuscript with your valuable comments and queries. We have worked hard to incorporate your feedback and hope that these revisions persuade you to accept our submission.

Response to Reviewer #3

First of all, we sincerely thank the reviewer for the careful reading of our paper and for the extensive, critical and insightful comments.

- 1. In this paper, the authors investigate the dynamics of super (very large amplitude, eastern ocean intensified) El Niños. Their primary tool is to obtain solutions to an atmospheric model. Forcing is by externally prescribed heating sources, $Q(x,y)$, that are spread sinusoidally throughout the troposphere and have a horizontal structure similar to those for*

ENSO and IOD events. Further, the model allows a background state, $\psi_b(x,y,t)$, to be externally prescribed, so that its solutions are the anomaly fields with respect to ψ_b that are driven by Q . The authors vary ψ_b from the observed, climatological annual cycle to a state that includes developing El Niños of various strengths.

Thank you for the nice summary of our work. It is particularly well put. We have borrowed aspects of the writing to improve the description of the experimental design in the revised version.

2. One problem I have with the paper is that the approach in the previous paragraph is not well described. (I had to read the paper several times before I figured out what is going on, and even now I am not completely sure what I wrote above is correct.) So, I think in the paper the Methods section must immediately follow the Introduction. There, details of the anomaly model must be clearly described. For example, is the model linearized about b , or are some nonlinear terms retained? My guess is that it must be linearized. Also, in this section the concept of state dependence must be clearly defined (perhaps similar to what I wrote above).

Yes, what you have wrote above is correct. We are sorry about the ambiguous description about aspects of the modeling. We have addressed this issue carefully in the revised manuscript.

In Nature Communications, the Methods section comes at the end, and here we have described the numerical models in greater detail. However, as per your suggestion, we have included an appropriately brief description when the model is first introduced in the text.

In the revised version, we decided not to incorporate the effects of State-dependence (please see response # 3 below).

3. *A conceptual problem with their approach (if what I wrote in the first paragraph is correct) is that they wish to investigate the response due to ENSO (and IOD) forcing Q but at the same time build in the ENSO response by including it in ψ_b . The logic of this approach seems circular to me. It seems like a way to introduce the nonlinear effects into the system through ψ_b , while at the same time keeping the anomaly model linear. Wouldn't it be more logical to run a nonlinear version of the model without any background state and simply vary the strength of Q ?*

We deeply thought about this critique and agree with your judgement, that there may be some circular logic in our approach. Currently we do not have a nonlinear version of the model that works, so we are stuck with the linear version. However, we will investigate the issue of state-dependence in the future, by using the Trajectory Piecewise Linear (TPWL) scheme developed by Rewiński^{15,16}. The modification of ψ_b by ENSO or IOD is significant, and therefore the modifications to ψ when ψ_b is markedly modified by ENSO/IOD would be very interesting to examine.

4. *Another issue is that, although the title and introduction of the paper suggest that it will explain the cause of super El Niños, there is no solution reported for a super El Niño (the solutions reported in the Results section are only for ordinary El Niños and IOD events). All discussion of super El Niños takes place in the Discussion section, where only a hypothesis about the cause of super El Niños is presented: They occur because of ocean-atmosphere feedbacks enhanced by the simultaneous occurrence of ENSO and IOD events. Apparently, then, their modeling approach cannot generate super El Niños, so their dynamics can only be inferred from their ENSO and IOD solutions. Thus, their paper does not really determine the underlying dynamics that cause of super El Niños, only presents a plausible hypothesis. In any case, the text about super El Niños in the*

Discussion section should be moved to a final Results subsection.

Our original intention, keeping in mind that we based our investigation solely on atmospheric modeling, was to only present a plausible hypothesis for the cause of super El Ninos. Reviewer #2 and the editor as well have raised similar concerns. In light of this, we have enhanced the paper by adding ocean modeling. We are now able to demonstrate why IOD forcing is a key factor in the development of super El Ninos. Further we have provided substantial support for our other key argument that self-limiting ENSO dynamics prevents the development of super El Ninos through coupled air-sea interactions in the Pacific alone. We hope the revisions give greater clarity to the ideas that we are putting forth.

5. *Figure 3 is confusing. Figure 3(a) describes $\psi_b(x,y,t)$ for the equatorial winds. Are they averaged over a latitude band, like 2S to 2N? Since ψ_b depends on t , what time is shown in the figure? In Figure 3(b), to what does the "Thermocline feedback" axis refer? I don't think you defined it. In Figures 3(b) and 3(c), I guess the dotted and solid curves are from solutions forced by the same Q but different ψ_b . Perhaps state that explicitly. What are the values of the ENSO state in Figure 3(a) for the ψ_b 's that you used?*

We are sorry that our inadequate description of Fig. 3 made it confusing. In that figure, equatorial winds were averaged from 5S to 5N, and ψ_b for August-September-October was used (*this was mentioned in the text*). This season was used since the IOD forcing peaked at this time. The "Thermocline feedback" referred to the processes by which zonal wind stress anomalies affected SST by changing the depth of the thermocline. We are again sorry that some crucial definitions were left out.

In the revised we have replaced Fig. 3 and other figures discussing the atmospheric solutions to ENSO/IOD forcing, with results from ocean modeling. We hope that you will find that we have exercised great care in defining terms/concepts before they are discussed.

6. *Given these problems, I cannot recommend publication of the manuscript in its present form. It may be that an improved version of the paper, which clearly addresses the above issues, will be publishable. Certainly, some of the issues can be addressed through a modest reorganization (modification) of the text. Others are more serious.*

Thank you again for pointing out the key issues with the original manuscript and through your positive judgement, an opportunity to strengthen the manuscript with your valuable comments and queries. We have worked hard to incorporate your feedback and hope that these revisions persuade you to accept our submission.

References

- [1] N. H. Saji and T. Yamagata. “Structure of SST and Surface Wind Variability during Indian Ocean Dipole Mode Events: COADS Observations”. In: *Journal of Climate* 16.16 (2003), pp. 2735–2751. DOI: [10.1175/1520-0442\(2003\)016<2735:SOSASW>2.0.CO;2](https://doi.org/10.1175/1520-0442(2003)016<2735:SOSASW>2.0.CO;2).
- [2] S. K. Behera et al. “A CGCM study on the interaction between IOD and ENSO”. In: *J. Clim* 19 (2006), pp. 1688–1705.
- [3] J.-J. Luo et al. “Interaction between El Nino and Extreme Indian Ocean Dipole”. In: *J. Clim.* 23 (2010), pp. 726–742.
- [4] T. Izumo et al. “Influence of the state of the Indian Ocean Dipole on the following years El Nino”. In: *Nature Geoscience* 3.3 (2010), pp. 168–172.
- [5] L.-C. Hong, L. Ho, and F.-F. Jin. “A southern hemisphere booster of super El Nino”. In: *Geophysical Research Letters* 41.6 (2014), pp. 2142–2149.
- [6] K. Takahashi and B. Dewitte. “Strong and moderate nonlinear El Niño regimes”. In: *Climate Dynamics* (2015), pp. 1–19.

- [7] M. J. McPhaden. “Genesis and Evolution of the 1997-98 El Nino”. In: *Science* 283.5404 (1999), pp. 950–954. DOI: [10.1126/science.283.5404.950](https://doi.org/10.1126/science.283.5404.950).
- [8] W. Cai et al. “Increasing frequency of extreme El Nino events due to greenhouse warming”. In: *Nature Climate Change* 4.2 (2014), pp. 111–116.
- [9] I.-H. Park, S.-K. Min, S.-W. Yeh, E. Weller, and S. T. Kim. “Attribution of the 2015 record high sea surface temperatures over the central equatorial Pacific and tropical Indian Ocean”. In: *Environmental Research Letters* 12.4 (2017), p. 044024.
- [10] C. N. Caviedes. “El Nino 1972: Its Climatic, Ecological, Human, and Economic Implications”. In: *Geographical Review* 65.4 (1975), pp. 493–509.
- [11] M. Glantz. *Currents of Change: Impacts of El Niño and La Niña on Climate and Society*. Cambridge University Press, 2001.
- [12] B. Fagan. *Floods, famines, and emperors: El Niño and the fate of civilizations*. Basic Books, 2009.
- [13] J. Cohen. “Weather forecasting: El Nino dons winter disguise as La Nina”. In: *Nature* 533.7602 (2016), pp. 179–179.
- [14] R. Emerton et al. “Complex picture for likelihood of ENSO-driven flood hazard”. In: *Nature Communications* 8 (2017).
- [15] M. Rewienski and J. White. “A trajectory piecewise-linear approach to model order reduction and fast simulation of nonlinear circuits and micromachined devices”. In: *IEEE Transactions on computer-aided design of integrated circuits and systems* 22.2 (2003), pp. 155–170.
- [16] D. Gratton. “Reduced-order, trajectory piecewise-linear models for nonlinear computational fluid dynamics”. PhD thesis. Massachusetts Institute of Technology, 2004.
- [17] M. J. McPhaden. “Evolution of the 2006-2007 El Niño: the role of intraseasonal to interannual time scale dynamics”. In: *Adv. Geosci.* 14 (2008), pp. 219–230.

Reviewers' comments:

Reviewer #1 (Remarks to the Author):

The revised manuscript reads better than the original version. However, I still have several major concerns. It is difficult for me to recognize certain events as super El Ninos in the absence of an objective basis for the classification. The cited studies in authors responses were earlier than 2015-16 event and hence that year does not appear in their studies as a super El Nino.

The difference between 1997-98 and 20115-16 discussion is not convincing. There is no index in this study to identify 1972, 1982 and 1997 as super El Ninos. There is also no way to compare amplitudes of these events with other El Nino events. Additionally, what are the statistical significances of the composites of those three events shown in Fig. 2? Authors have only compared 2015 event with 1997 event for October-December season for a selected few oceanic and atmospheric fields without explaining the importance of the season and those fields for ENSO classifications.

There are several recent studies that have discussed super El Ninos including the 2015 event (e.g. Chen et al. 2017, Scientific Reports, Latif et al. 2015, Climatic Change, McPhaden 2015, Nature Climate Change, Guilyardi et al. 2016, BAMS, Levine et al. 2016 J. Climate). Please refer those and develop a criterion that would objectively identify super El Ninos.

How the Indian Ocean condition differed in these years? Why the IOD of 2006 and 2015 did not help the El Ninos of those years to culminate into super El Ninos? This was my concern earlier and I have not found any clear response from the authors. The Indian Ocean connection shown in Fig. 6 is a derivative of all the El Nino events. Please plot the evolution of IOD based on observed data in those classified super El Nino years and compare those with other events such as 2006, 2009, 2012 and 2015. The 2006 event is discussed using model simulations before showing any results from observation. I believe if one strong IOD can lead to a strong El Nino, it should be repeated in other strong IOD cases. If other factors are responsible for diminishing the role of IOD in some of the years, the presence of IOD in 1982 and 1997 could be coincidence unless we clearly separate the IOD influences from other incidental factors that could lead El Ninos to super El Ninos.

Reviewer #2 (Remarks to the Author):

Recommend: minor revision

The authors have made substantial changes in response to the reviewers' comments, thus improving their article considerably.

Their definition of a 'super El Nino' is now made clear, and is acceptable. Results using a simple tropical ocean model have been added, to good effect. The result is an interesting paper, worthy of publication with minor revision.

Specific comments:

It is important to distinguish between actual and anomaly values. The notation itself (SSH, OLR etc) does not make this clear, so 'anomaly' should be specified throughout the text wherever appropriate.

Introduction ~line 60: wetter southern California and drier NW America is the favoured (but not guaranteed) response to El Nino events in general, not just super events. Is it known why 2015 was

different?

Introduction ~line 78: it would be interesting (but not essential) to say something about the predictability of very strong/super El Nino events.

Introduction line 92: define/describe negative and positive phases

Figure 2: the colour scale is rather misleading, as at a glance it suggests positive and negative values as in other figures. The message would be clearer with e.g. shades of a single colour. The caption would be better as 'seasonal cycle of the standard deviation of interannual SST and OLR anomalies'.

Results line 104: a bit of explanation about how the convective anomalies arise would be useful (e.g. growth of small anomalies by Bjerknes feedback)

line 106: explain here that easterly anomalies favour local equatorial surface divergence and upwelling, which offsets the downwelling effect of Kelvin waves arriving from the west. Worth noting that in Fig. 3 the equatorial easterly anomalies are largely confined east of 90W until the event is well developed near the end of the year.

Paragraph 2: how (and when) was the ocean model started? What are the initial conditions? Relatedly, what causes the prominent equatorial thermocline anomaly in May? Point out that the SST anomalies are not interactive and do not affect the ocean.

Impact of IOD line 207: explain here what is meant by 'control for the effect of ENSO'. The caption to Fig. 6 has a lot of associated detail that belongs better in the main text.

Line 210: point out here the key features of Fig 6abc: i.e. in Fig 6b substantial east Indian reduced cloudiness interpreted as less convective heating which is associated with westerlies in west equ Pac; in Fig 6c the east Indian anomaly is weak and west Indian enhanced cloudiness (more heating) is associated with easterly equatorial winds that extend to the west Pacific.

Discussion line 338: I am not sure there is a case for an inherent coupled mode; rather ENSO and IOD are interacting phenomena.

Supplement: in Fig. 1 the warming trend in 2015 is also prominent in Nino3.4

Fig. 2: is the convective threshold relevant to the east Pacific, where surface temperature gradients are also influential in convection location?

Fig. 9: bottom panels should be labelled e,f,g

Some wording suggestions:

last line of abstract: 'tropical Pacific dynamics' rather than 'ENSO dynamics'

introduction line 21: '... made El Nino a term familiar to all people'

line 23: 'strong' rather than 'astonishing'

line 28: 'unusually strong zonal wind stress anomalies...'

line 52: '... SST is normally relatively cold...'

results line 102: westerly wind anomalies (and likewise elsewhere)

line 103: .. central equatorial Pacific ... deepen the equatorial thermocline ...

line 128: ... the atmospheric Kelvin wave response ...

line 129: ... the northeast tropical Pacific ...

line 133: ... strong eastern equatorial Pacific ...

impact of IOD line 223: ... mid-troposphere and zero at the surface.

Line 232: ... three dimensional background flow ...

Fig. 4: the arrows have units of pseudostress, please explain.

Reviewer #3 (Remarks to the Author):

Please see next page for comments.

Review of a manuscript, entitled “A model for super El Niños” by Hameed et al.

SUMMARY: This paper is a resubmission of a previously-submitted manuscript concerning the dynamics of super (very large amplitude, eastern-ocean intensified) El Niños. It is much improved scientifically over the previous version. Here are a list of the improvements. 1) The atmospheric model is now supplemented by an ocean model. 2) The externally prescribed, background state of the atmospheric model, $\psi_b(x, y, z, t)$, is now fixed to the climatological annual cycle. 3) The physics of both models are described in sufficient detail (although I have concerns about both of them; see below). 4) A solution for a super El Niño is now presented, and the contributions of both Pacific and IOD winds explicitly shown.

As noted by the authors, new ideas in this paper are: *i*) the limiting impact on ordinary El Niños of westerly winds in the far-eastern Pacific; *ii*) a more-extended analysis of the 2006 IOD event than done previously; and *iii*) the importance of contemporaneous IOD events in generating super El Niños. My reservation about these results is whether they are in fact completely new, as there has been so much published on the IOD/ENSO interactions (see, for example, Comment 16). At least to me, the above ideas are new and noteworthy, and they are now presented much more clearly than in the previous manuscript. So, I believe that with a bit more improvement the paper will be publishable.

There are several scientific issues, however, that the authors will need to address. In particular, I wonder if the authors have adequately reviewed previous literature (Comment 16), and note that the descriptions of the oceanic and atmospheric models are unclear (Comments 17–29). The paper also has many editorial problems that should be eliminated. I point out many of them, but by no means all. Editorial issues include: use of past tense when present tense is better; lack of spaces between words; paragraphs that are too long or should be combined. In some cases, I lost the logical flow of the text, because unrelated (weakly related) text was included.

A list of specific comments follows. The list is lengthy, but many of the comments are very easy to respond to. I would like to see the authors’ response to all of them, especially the scientific ones.

SPECIFIC COMMENTS:

1. abstract, line 7: It is dangerous (and egotistical) to use the phrase “for the first time,” since you may (are likely) to be wrong. In the present case, you discuss the impact of easterly winds in far-eastern ocean associated with ENSO. Such winds have been noted for decades, and I would not be at all surprised if someone has commented on their possible impact on ENSO. I would drop all such statements from the manuscript (there are a couple of other instances).

2. line 21: Perhaps “general populace” is better than “regular people.”

3. lines 27–29: The sense of the impact on thermocline depth is wrong. I guess on line 28 you mean “thermocline depth *anomalies* across the Pacific and strong zonal-wind-stress *anomalies* in the western and central, equatorial Pacific.” On line 27, start the sentence with “Further, they *were all* also characterized...”.

4. lines 37 and 44: I, for one, would use “First” rather than “Firstly,” as the former is more common usage. To provided a clue to your readers that you are changing the topic, replace “The second” with “Second(ly), there is a.”
5. lines 45–48: It would be good to refer your readers ahead to a figure where you illustrate the properties noted here.
6. line 55: The logic of this sentence is not clear. It is better if you replace “mentioned here” with “mentioned, but that was not the case for the 2015 event.”
7. lines 62–66: Does the sentence of this paragraph fit the logical flow of the text. I find it distracting. You could drop it, and write: “...measure of extremity. Given the other differences noted above, we do not consider the 2015 event to be a super El Niño. Our proposed model suggests that super El Niños occur only when there are extremely strong SST variations in the far-eastern Pacific”.
8. use of “eastern” vs. “far-eastern”: In many cases, I think you really mean “far-eastern,” rather than “eastern.” Please check all usages.
9. lines 69–70: Is this sentence really useful, or just distracting.
10. lines 76–77: Perhaps don’t start a new paragraph here. Perhaps replace “Many” with “Several.”
11. line 79: Replace “will” with “would.”
12. lines 80–81: Replace with “The event did not materialize despite the supposedly “favorable conditions” being present^{18,19}. Clearly much...” That is, drop the phrase about climate forecasts, which seems unnecessary and distracting.
13. lines 86–87: Don’t start a new paragraph here. Don’t the two paragraphs discuss the same topic?
14. lines 87–94; discussion of IOD: I realized that one of the reviewers asked you to define an IOD event. I, however, found this discussion very distracting. A simple fix is to put it in parentheses. Also the sentence on lines 92–94 seems like too much information to me, and I think you should drop it. Can you put the IOD discussion into a footnote.
15. line 95: Replace with “show that the *atmospheric* Kelvin wave.”
16. Introduction: Have adequately summarized preceding work in the Introduction (and elsewhere). For example, the following are two papers that you did not reference, which seem to me to be relevant: Annamalai et al. (2004; J. Clim., 302–319) and Annamalai et al. (2010; J. Clim., 3922–3952). I bring these two papers to your attention in particular because, as I recall, they conclude that the Pacific response to IOD events is much weaker than in your solutions.
17. lines 364–366: Before proceeding with results, I first comment on the models used. Replace with “...smoothed by a 91-day running-mean filter...” Replace with “An intensity threshold of 4 m/s is used to discriminate WWB events, but for simplicity we did not apply a threshold in event duration.”

18. The ocean model: Please reword the first two paragraphs in “The ocean model” section. Combine the two paragraphs. On lines 370–373, replace with: “It consists of a linear, 1.5-layer (shallow-water) model, which simulates the first-baroclinic-mode response of the ocean, as well as a linear, empirical SST equation. The 1.5-layer model consists of . . .”
19. lines 374–376: Replace with “The model equations are” and add commas after all the equations (they should be regarded as part of the sentence. On line 376, replace with “the wind-stress forcings are τ_x and τ_y , $f = \beta y$ is the Coriolis parameter in the equatorial β -plane approximation, $g' = g\rho^{-1}\delta\rho$ is the reduced-gravity coefficient, and . . .”
20. line 383: Why is the grid spacing so large? Surely, you could use a resolution of 25–50 km. If you use $\Delta x = 2^\circ$, $\Delta y = 1^\circ$, then I think you need to tell your readers why you think you can get away with such large values.
21. lines 383–386: It logical to discuss equation (5) here. So, move this paragraph to the end of subsection after line 398.
22. equation (5): Does (5) make physical sense? Such a representation for T is common in analytic ENSO models, but I have never seen it used in a numerical model. I can understand the impact of h on T in the eastern Pacific where the thermocline is shallow, and Δh impacts the upwelling of subsurface water. I can also understand that easterly winds ($\tau_x < 0$) could impact T by Ekman divergence and upwelling, but westerly wind ($\tau_x > 0$) should have no effect of this sort. I have no idea how τ_y can impact T . If you set $\gamma = 0$, do your results change?
So, I think you need a better justification of (5). The fact that α , β , and γ depend on x is good. For example, I suppose that $\alpha(x)$ is set so that it is appreciable on in the eastern Pacific. Shouldn’t those coefficients also be functions of y ? For example, the α -term should only apply in the Pacific cold tongue.
23. lines 404–406: The most important aspect of the LBM is its linearity, so I would mention that first. Replace with: “The LBM is derived by linearizing the AGCM about a specified background state, $\psi_b(x, y, z, t)$, eliminating water vapor, and dropping the model’s radiation scheme. The diabatic heating that forces the atmosphere is then represented by an externally prescribed function, $Q(x, y, z, t)$.”
24. lines 412–413: Replace with: “The background state $\psi_b(x, y, z, t)$ is set to the seasonal climatology of the NCEP reanalysis data.”
25. lines 414–442: Please carefully go over the entire discussion in these three paragraphs to check their correctness and to improve their clarity and logical flow.
26. line 427 and elsewhere; April: Do not “hardwire” your discussion to the month of April. Let “ ψ_0 ” be your initial state variable, rather than “ $\psi(\text{April})$,” and then you can refer to later times as “ ψ_{n-1} ” and “ ψ_n ,” and so on.
27. method of integrating the LBM: Rather than integrating the LBM forward in time continuously, the authors split the time domain into a series of “steps.” The interval between steps is not clear from what is written. Is it 10 days, one month, or some other interval? Please state the interval in the text. But, why run the model in this “steppy” manner? I

guess because their LBM code is set up that way. If so, why didn't you just modify the code to run continuously?

Let t_n indicate a particular time step. Then, at that step the LBM is run out to equilibrium with $Q(x, y, z, t_n)$ and $\psi_b(x, y, z, t_n)$, and (I guess) ψ_n is this equilibrium value. This process is okay because the Rayleigh friction and Newtonian cooling time scales are so small, that is, $\nu^{-1} = \kappa^{-1} = 1$ day. As a result, the LBM rapidly comes into equilibrium in a time scale only somewhat longer than 1 day. Since the forcing function, Q and ψ_b , vary on a much longer time scale of the order of a month (30 days), the LBM response is essentially in *quasi-equilibrium*.

28. initial conditions at step t_n : The authors assume that they need to start the integration for ψ_n with a non-zero initial condition, which (I guess) they take to be ψ_{n-1} . But, since the system is in quasi-equilibrium (see previous comment), *no initial condition is needed*. In only a few days, the system adjusts to equilibrium with the forcing functions Q and ψ_b , and completely “forgets” the initial condition.

29. equation (6): Equation (6) appears to be a method for interpolating between time steps. It is impossible to tell, however, because the equation *makes no physical sense at all* since its left- and right-hand sides have different units (units of ψ_b vs. units of $\psi_b Q$). So, I have no idea how to interpret or fix it.

On the other hand, because the LBM solution is essentially in quasi-equilibrium with the slowly-varying forcing functions, I don't think any version of (6) is actually needed. It should be dropped. In summary, the authors simply need to calculate ψ_n without any initial condition ($\psi_{n-1} = 0$ will do). Calculating ψ in this simple way should reasonably approximate the solution to a continuous integration.

Given that (6) is not clear, I can't tell whether the LBM solutions reported in the text are sensible. This issue must be addressed.

30. lines 152–160: In this paragraph you state what you will be doing in the subsequent section. The paragraph should be moved to line 162, as a introduction to the section. Suppose you are a reader trying to reread Section “The impact of IOD on the equatorial Pacific.” You would want to read the paragraph on lines 152–160, but would not find it. Generally, for this reason it is bad practice to introduce the next section in a last paragraph to the preceding one.

31. lines 169–171, 185, and elsewhere; “direct” vs. “indirect evidence: I do not understand the distinction between “direct” and “indirect” evidence. I guess you are arguing that the 2006 was a pure an IOD event (no EL Niño at all) as possible, and in that way your analysis is better (more direct) than previous ones. Really? Why not drop this confusing nomenclature, and just write something like: “Here, to supplement previous work we analyze in detail a solution during 2006, a year with a prominent IOD event but very weak or no El Niño response.”

32. line 185: Rewrite the text to avoid using “for the first time.”

33. line 192: Regarding the relative sizes of convective variations in the Indian and Pacific Oceans, is comparing their areal average values appropriate. After all, the area of the Pacific Ocean is much larger than that of the Indian Ocean.

34. lines 197–198: Don't start a new paragraph here, as the following two sentences are related to the topic of the previous paragraph.
35. line 201: Start a new paragraph with the sentence: "To demonstrate the IOD... It does start a new topic.
36. lines 203–204: Don't start a new paragraph here.
37. line 204 and many places elsewhere: You wrote "(Fig. 6(a)-(c))" here, and there are similar expressions elsewhere. For one thing, there are too many parentheses. For another, "Fig." should be plural. So, replace with "(Fig. 6a–6c)" here and similarly elsewhere.
38. line 209: You note here that the IOD winds are strong over the western Pacific. How strong relative to ENSO winds? Can you please compare them? My impression is that IOD winds are much weaker than ENSO winds.
39. lines 212–223; overview of the LBM: I expect that this discussion will change once the description of the LBM is fixed in the Methods section. In any case, try to shorten it. Move line 230, which is logically out of place where it is now located, to line 216, replacing nearby text with something like: "It is linearized about a background state, $\psi_b(x, y, z, t)$, which we set to be the annual cycle from the NCEP reanalysis⁴⁵ climatology."
40. lines 238–239: Lots of flowery words here, so much so that for me the meaning of the sentence is lost. Also, this first sentence is really a justification to study the 2006 events, so if you keep it, it belongs at the beginning of the section.
41. line 241: I have never seen the word "Noteworthily" used in an oceanographic paper before. A google search, however, confirms that it is in fact a real word, albeit one that is not commonly used. Perhaps replace it.
42. line 261: You tend to use emotional language in the text, which should be avoided in scientific writing. So, I would drop "strongly" from the sentence here. Also, on this line you wrote "west Pacific," and there are similar usages elsewhere. I think you should replace such expressions with "western Pacific." Alternately, but not as good, you might write "West Pacific."
43. line 266: Again emotional language. I would drop "overwhelming." Frankly, I don't feel overwhelmed by what you presented. I do feel, though, that you have made a strong and convincing argument. Leave the decision to be "convinced" or "overwhelmed" to your readers.
44. line 278–279: The complete sentence on these lines is not physically precise. Drop it.
45. lines 294–296: Why does the SSHA appear first at the eastern boundary? Your explanation only applies to SSTA. It must result from reflected Rossby waves, which increase the eastern-boundary response from that of the incoming Kelvin wave. You might note this property.
46. line 294 and elsewhere: You wrote "far western Pacific." Replace with "far-western Pacific," that is, add a hyphen between the two modifiers. There are many instances in this paper where strings of modifiers need either hyphens or commas between them.

47. line 304; I think the upper and lower limits in Figure 10 should be adjusted. With their present values, much of the red and blue areas are fat blobs beyond those limits. I, and your readers I expect, would like to see more color contours in these regions.

48. line 310: Drop “first.”

49. line 345: Perhaps start a new paragraph with: “Finally, we note that, although”

50. lines 348–350: The meaning of the complete sentence in these lines is unclear. I have no idea what it means.

Response to Reviewers

Response to Reviewer #1

We thank the reviewer for the comments on the revised version of the manuscript. We understand that the reviewer is largely concerned with our argument that the 2015 El Nino event is not a super El Nino, and that 2015 is not an IOD year. We have done the following to address these concerns (other questions are addressed below this part):

- To provide a more compelling support for our definition of super El Ninos, and our argument that 2015 is not a super El Nino, we have
 - Added several indices based on OLR anomalies (Supplementary Figure 2) - this provides an insightful comparison between various El Nino events since 1979 (OLR data is only continuously available from 1979); OLR anomalies are a measure of the strength of coupled variability relevant to ENSO; the weak OLR variations in 2015, relative to that in other El Ninos, further support our argument that the 2015 El Nino is not a super El Nino.
 - Enhanced paragraph two (lines 30-38 of the tracked version) of the manuscript to clearly indicate that the super El Ninos mentioned in our work were identified using objective analyses in previous studies (Hong et al., 2014; Li et al., 2015; Takahashi and Dewitte, 2014).
 - A related question, therefore, is why was the SST anomalies associated with the 2015 El Nino so large. This question is relevant, because many researchers simply use the magnitude of SST anomaly alone in deciding if an El Nino is, or not is, a super El Nino. This question is outside the scope of our main work; nevertheless, considering its relevance, we have added a paragraph of discussion in the last section (lines 525-532) that suggests that factors external to the tropical Pacific may have played a role in the large SST anomalies experienced during 2015.
- To provide more information about the state of the Indian Ocean, in particular IOD variations, we have added Supplementary Figure 1 and a Supplementary Discussion; the figure depicts longitude-time variations of climate anomalies over the tropical Indian Ocean during 1997 and during 2015;

the discussion describes the objective analyses that classified 1972, 1982, 1994, 1997, and 2006 as IOD years, and 2015 as a non-IOD year.

From here on, we address the specific issues raised by the reviewer, paragraph by paragraph.

1. The revised manuscript reads better than the original version. However, I still have several major concerns. It is difficult for me to recognize certain events as super El Ninos in the absence of an objective basis for the classification. The cited studies in authors responses were earlier than 2015-16 event and hence that year does not appear in their studies as a super El Nino.

The difference between 1997-98 and 2015-16 discussion is not convincing. There is no index in this study to identify 1972, 1982 and 1997 as super El Ninos. There is also no way to compare amplitudes of these events with other El Nino events. Additionally, what are the statistical significances of the composites of those three events shown in Fig. 2? Authors have only compared 2015 event with 1997 event for October-December season for a selected few oceanic and atmospheric fields without explaining the importance of the season and those fields for ENSO classifications.

The classification of the 1972, 1982, and 1997 events as super El Ninos are based on the objective analyses by Hong et al. (2014), Takashi and DeWitte (2015), and Li et al. (2015); the first study is also referred to by other investigators who have discussed super El Ninos, viz., Chen et al. 2017 and Latif et al. 2015. **Thus, the identification of 1972, 1982, and 1997 as super El Ninos have a strong, objective basis, and is not unique for this study.** *We have added a few lines of discussion to clarify this point (lines 30-38 in the tracked version of the manuscript).*

Further, the following were added to enhance the comparison between 2015 and other El Nino events:

- Supplementary Figure 2, which is new, compares El Ninos since 1979 using OLR anomalies.
- Supplementary Figure 1, also new, compares Indian Ocean response during the 1997 and 2015 El Ninos.

The reviewer also has asked about “the statistical significance of the composite of those three events shown in Fig. 2”. *Regrettably, the reviewer is mistaken about Fig. 2: this figure is a plot of the*

seasonal cycle of standard deviation of SST and OLR. Thus, we cannot address the question; it is not meaningful in the context of Fig. 2.

To address the reviewer's last point (in the first paragraph of the reviewer comments), we have added an explanatory clause in line 44 - "*which are key variables in the development and evolution of an El Nino event*".

2. There are several recent studies that have discussed super El Ninos including the 2015 event (e.g. Chen et al. 2017, Scientific Reports, Latif et al. 2015, Climatic Change, McPhaden 2015, Nature Climate Change, Guilyardi et al. 2016, BAMS, Levine et al. 2016 J. Climate). Please refer those and develop a criterion that would objectively identify super El Ninos.

As mentioned in the previous response, the three super El Ninos mentioned in our study were identified using objective analyses in previous studies (three different studies find the same - that 1972, 1982, and 1997 El Ninos are unique; and that a strong, eastern-intensified response is one of their unique characteristics).

We are confused that out of the 5 papers referred to by the reviewer only two (Chen et al., 2017; Latif et al., 2015) discuss super El Ninos; the others have no bearing on super El Ninos:

- McPhaden (2015) comments about recent El Nino predictions - the grand failure of the 2014 predictions, and the unexpected development of a strong El Nino in 2015.
- Guilyardi et al. (2016) reports the 4th Clivar Workshop on the evaluation of ENSO processes in climate models.
- Levine et al. (2016) discusses the question why El Ninos have stronger amplitude than La Ninas.

Further, Chen et al. (2017) and Latif et al. (2015) both refer to Hong et al. (2014) in discussing super El Ninos; Hong et al (2014) is the objective analysis that identified 1972, 1982, and 1997 as super El Ninos. *Since, such an analysis has already been undertaken, and is used in other studies, we do not agree with the reviewer that we need to develop a new identification scheme for super El Ninos.*

We feel that the reviewer's real concern is related to our not classifying 2015 as a super El Nino. In the original manuscript, we had devoted a figure and a paragraph of discussion to explain why 2015 cannot be considered a super El Nino - the reviewer has not given any explanation why this analysis is not acceptable; so, we are unable to understand the reasons for the reviewer's disagreement. Nevertheless, it is worthwhile to further clarify this issue; the steps that we have taken to address this issue are described at the beginning of this rebuttal.

3. How the Indian Ocean condition differed in these years? Why the IOD of 2006 and 2015 did not help the El Ninos of those years to culminate into super El Ninos? This was my concern earlier and I have not found any clear response from the authors. The Indian Ocean connection shown in Fig. 6 is a derivative of all the El Nino events. Please plot the evolution of IOD based on observed data in those classified super El Nino years and compare those with other events such as 2006, 2009, 2012 and 2015. The 2006 event is discussed using model simulations before showing any results from observation. I believe if one strong IOD can lead to a strong El Nino, it should be repeated in other strong IOD cases. If other factors are responsible for diminishing the role of IOD in some of the years, the presence of IOD in 1982 and 1997 could be coincidence unless we clearly separate the IOD influences from other incidental factors that could lead El Ninos to super El Ninos.

Since there are many different, unrelated questions in this paragraph, we use a list below to organize our answers.

- (a) The reviewer asks: "How the Indian Ocean condition differed in these years?"

We do not understand which are the years that the reviewer refers to in here; so we have to guess. We have two guesses, and hence two responses:

- i. Perhaps the reviewer meant to refer to 1972, 1982, and 1997.

In lines 104-105 we mention that positive IOD conditions prevailed during these years. We have also added a Supplementary Discussion, which states how we establish the state of the Indian Ocean during these years: here, we also mention the differences between 1972, 1982, and 1997; the first two are moderately strong IOD events, and the last one is a strong IOD event.

ii. Or, perhaps the reviewer meant the differences between 1997 and 2015.

We have already addressed this in the responses to the reviewer's previous questions (for example, see the discussion at the beginning of the rebuttal).

(b) In the next sentence, the reviewer asks: "Why the IOD of 2006 and 2015 did not help the El Ninos of those years to culminate into super El Ninos? This was my concern earlier and I have not found any clear response from the authors."

In the original review, the reviewer had commented "The discussions on why the 1994 and 2006 IODs could not lead to super El Ninos of those years are not convincing" - The reviewer's previous comments related to the 1994 and 2006 IODs; now, the reviewer mentions the years 2006 and 2015. Please note that we had devoted a paragraph, in our previous rebuttal, to explain why IOD alone do not lead to super El Ninos; however, since the reviewer did not detail why our response was not clear, we are at a loss to understand the reviewer's concerns.

Further, the question also does not make sense in the context of our results: this is because, our results clearly show that positive IOD and El Nino interaction is needed for super El Nino development; the year 2006 had a strong positive IOD, but no El Nino (lines 444-449); the year 2015 had a strong El Nino, but IOD conditions were non-existent over the Indian Ocean (see Supplementary Discussion and Supplementary Figure 1).

(c) Next, the reviewer states: "The Indian Ocean connection shown in Fig. 6 is a derivative of all the El Nino events."

First, we do not understand what a derivative of El Nino means. Secondly, Fig. 6 consists of two parts - the left half summarizes a partial regression analysis of observations, designed to isolate the role of IOD; on the other hand, the right half describes model simulations, designed to highlight the mechanisms whereby IOD affects Pacific winds. The first analysis statistically isolates the impact of IOD from El Nino - by design, it removes the El Nino impact, while clarifying the role of IOD. Frankly, we are at a loss to understand the reviewer's comment.

- (d) The reviewer continues: “Please plot the evolution of IOD based on observed data in those classified super El Nino years and compare those with other events such as 2006, 2009, 2012 and 2015. ”

The reviewer has not mentioned why this additional analysis is required; again, we are at a loss to understand the reviewer’s intent:

- Perhaps, does the reviewer doubt that IOD events occurred during 1972, 1982 and 1997?

The occurrences of the positive IOD events during 1972, 1982, and 1997 is well reported in the available literature (Saji et al., 1999; Saji and Yamagata, 2003a,b; Yamagata et al., 2004, Hameed, 2017). Thus, the need for such a figure is not very clear to us. Nevertheless, in in the revised version, we have added a supplementary discussion that describes how IOD events were detected. This scheme is based on previous work by Saji and Yamagata (2003a). While, it uses DMI as one criteria for identifying IOD, it also makes sure that there is coupled evolution of anomalies of SST and surface winds, and that SST anomalies are of opposite sign during the identified IOD events.

Also, since the state of the Indian Ocean during the strong 2015 El Nino is not widely known, and its comparison with that during 1997 serves to further highlight the arguments made in our paper, we have added a new figure (see Supplementary Figure 1). Further, we note that our scheme does not detect IOD events in 2009 and 2015; the evolution of the 2006 IOD is already discussed in Fig. 7.

- (e) The reviewer states incorrectly that “The 2006 event is discussed using model simulations before showing any results from observation.”

This is not correct: observational analysis is first presented (Figures 5,7); the model simulations for 2006 are first discussed in Fig. 8. Further, we are not aware of any conventions that stipulate that a model simulation or prediction should be preceded by observational analysis.

- (f) The reviewer’s concluding remarks: “I believe if one strong IOD can lead to a strong El

Nino, it should be repeated in other strong IOD cases. If other factors are responsible for diminishing the role of IOD in some of the years, the presence of IOD in 1982 and 1997 could be coincidence unless we clearly separate the IOD influences from other incidental factors that could lead El Ninos to super El Ninos.”

*Regrettably, the reviewer is incorrectly rephrasing our conclusions: **we did not claim that a strong IOD can lead to a strong El Nino; what we did claim was that a strong IOD can create conditions that favour a moderate, transient warming of the tropical Pacific. We also concluded that interaction between positive IOD and El Nino is needed for super El Ninos to develop - all the super El Ninos so far (1972, 1982, and 1997) satisfy these conditions.***

The nature of SST anomaly variation during the strong El Nino of 2015 further support our model - viz., super El Ninos can only develop when IOD co-occurs with El Nino. The confusion regarding the 2015 El Nino is related to its strong SST anomaly amplitude in the west and central Pacific; however, this is weak in the far-eastern Pacific. In other words, the strong eastern-ocean intensification that is a characteristic feature of super El Ninos is absent for the 2015 El Nino. The weakness of its SSH and wind anomalies compared to the 1997 event were presented in Fig. 1. Supplementary Figure 2, further, shows that the OLR anomalies during 2015 are one of the weakest in the available observational record. These disparities between the amplitude of SST anomaly and that of other ocean-atmospheric anomalies during 2015 call for a different explanation of the strength of its SST anomalies - recent studies, and our own assessment (Supplementary Fig. 3) points to the role of a longer time scale (decadal or climate change time scale) warming trend. We have added a paragraph in the discussion section to highlight this issue.

Finally, we take the opportunity to thank the reviewer for the critical comments. We feel that the manuscript has benefitted by addressing these, to the extent allowed by the scope of our work, and we hope that our revisions persuade you to accept our submission.

Response to Reviewer #2

Recommend: minor revision

The authors have made substantial changes in response to the reviewers' comments, thus improving their article considerably.

Their definition of a 'super El Nino' is now made clear, and is acceptable. Results using a simple tropical ocean model have been added, to good effect. The result is an interesting paper, worthy of publication with minor revision.

We thank the reviewer for the well-thought-out, constructive comments during the reviews, and sincerely appreciate the time and energy that the reviewer took out for these. Your reviews have helped us to significantly enhance the scientific aspects of this work and improve the clarity of the discussions. We are very grateful for this.

Please note that the line numbers, mentioned below, refer to the version of the manuscript where all changes are tracked; in the tracked version, deleted items are marked by a horizontal line passing through the center of the item; items added in response to Reviewer# 2 comments are in bold and have an yellow background.

Specific comments:

1. It is important to distinguish between actual and anomaly values. The notation itself (SSH, OLR etc) does not make this clear, so "anomaly" should be specified throughout the text wherever appropriate.

Thank you for pointing this out. We have revised the text to specify "anomaly" wherever appropriate.

2. Introduction line 60: wetter southern California and drier NW America is the favoured (but not guaranteed) response to El Nino events in general, not just super events. Is it known why 2015 was different?

We agree with you. The role of ENSO on precipitation over NW America is quite complex. While ENSO has a dominant influence in the climate variations of the tropics, this may not hold true at and polewards of the extratropics. Here, other competing factors have impacts equalling or exceeding

ENSO's influence. For example, Enfield et al¹ shows, for the case of North America, how the Atlantic Meridional Oscillation can modulate ENSO's influence. Cohen² suggests that the winter of 2015/16 over North America may have been highly influenced by conditions over the Arctic that led to a weakening of the polar vortex. Another possibility, that we intend to explore in a future work, is the influence of super El Nino events themselves in skewing the picture of how ENSO influences various regions worldwide.

3. Introduction line 78: it would be interesting (but not essential) to say something about the predictability of very strong/super El Nino events.

Thank you for the suggestion. We have added a paragraph in "Discussion" to address this issue (lines 533-539).

4. Introduction line 92: define/describe negative and positive phases

We have briefly described the negative and positive phases of IOD (line 105-107). In addition, taking into account the comments of Reviewer#3, we simplified the description of IOD, and reworked the entire paragraph for better readability.

5. Figure 2: the colour scale is rather misleading, as at a glance it suggests positive and negative values as in other figures. The message would be clearer with e.g. shades of a single color. The caption would be better as "seasonal cycle of the standard deviation of interannual SST and OLR anomalies."

Thank you for the kind suggestion. We have changed the color scheme for Fig. 2 to one ranging from Yellow through Orange to Brown (adapted from Cynthia Brewer's website colorbrewer2.org). We hope the misleading aspect has been eliminated. The caption for Fig. 2 was also changed as suggested.

6. Results line 104: a bit of explanation about how the convective anomalies arise would be useful (e.g. growth of small anomalies by Bjerknes feedback)

Thank you for the suggestion. We have reworked this paragraph to describe the processes involved in more detail (lines 128-142).

7. line 106: explain here that easterly anomalies favour local equatorial surface divergence and upwelling, which offsets the downwelling effect of Kelvin waves arriving from the west. Worth noting that in Fig. 3 the equatorial easterly anomalies are largely confined east of 90W until the event is well developed near the end of the year.

We have added this explanation (lines 140-141).

8. Paragraph 2: how (and when) was the model started? What are the initial conditions? Relatedly, what causes the prominent equatorial thermocline anomaly in May? Point out that the SST anomalies are not interactive and do not affect the ocean.

In response to this we have added the following. We clarify (line 159) that monthly mean wind stress anomalies were used to force the model. In lines 157-158, we have noted that the SST is not interactive. In lines 159-160, we have noted that the model was initiated in January from a state of rest.

Regarding the prominent thermocline anomaly in May - The composite El Nino westerly wind anomalies extend deep into the eastern Pacific during boreal spring. This may be related to the seasonal cycle of SST in the equatorial Pacific. The zonal gradient of climatological SST is weak around boreal spring, with a warm SST of 26.5 C found over much of the eastern Pacific. On this relatively warm SST background, it is easy to generate convective anomalies and wind anomalies over the eastern Pacific. Also, note the high standard deviation of interannual SST anomalies (Fig. 3) at the eastern Pacific around this season. Similarly note that the maximum eastward extent of OLR standard deviation around boreal spring.

9. Impact of IOD line 207: explain here what is meant by 'control for the effect of ENSO'. The caption to Fig. 6 has a lot of associated detail that belongs better in the main text.

We changed the phrase 'control for the effect of ENSO' to 'isolate the relative impact of ENSO'. We hope that this makes the meaning of the sentence more clear without introducing additional technical details within this paragraph.

Thank you for the observation on the detailed information in the figure captions. This was done for

two reasons. First, we tried not to break the flow of the discussion by introducing auxiliary details. Second, there was a restriction on the number of words allowed in the main text.

10. Line 210: point out here the key features of Fig 6abc: ie in Fig 6b substantial east Indian reduced cloudiness interpreted as less convective heating which is associated with westerlies in west equatorial Pacific; in Fig 6c the east Indian anomaly is weak and west Indian enhanced cloudiness (more heating) is associated with easterly equatorial winds that extend to the west Pacific.

Thank you for this suggestion. We have done this (paragraph starting on line 315).

11. Discussion line 338: I am not sure there is a case for an inherent coupled mode; rather ENSO and IOD are interacting phenomena.

We have removed this sentence.

12. Supplement - Fig 1 the warming trend is also prominent in Nino3.4

We have revised the caption of *new Supplementary Fig. 3* to indicate this.

13. Supplement - Fig 2: is the convective threshold relevant to the east Pacific, where surface temperature gradients are also influential in convection location?

In our view, while surface temperature gradients are important here, the absolute value of sea surface temperature should also be considered.

14. Supplement - Fig 9: bottom panels should be labelled e,f,g

We have corrected the Figure labels as pointed out.

15. last line of abstract: “tropical Pacific dynamics” rather than “ENSO dynamics”

We have implemented this correction.

16. introduction 21: “... made El Nino a term familiar to all people ”

Thank you for this. This sounds much better (line 21).

17. line 23: “strong” rather than “astonishing”

Thank you. We have changed this as suggested (line 24).

18. **line 28: “unusually strong zonal wind stress anomalies ...”**

We have revised this (line 29).

19. **line 52: “... SST is normally relatively cold ...”**

We have revised this (line 65).

20. **line 102: ... westerly wind anomalies (and likewise elsewhere)**

We have carefully checked the manuscript and have implemented this correction.

21. **results line 103: ... central equatorial Pacific ... deepen the equatorial thermocline ...**

We have rewritten the entire paragraph and implemented this along with your other suggestions (lines 128-142).

22. **line 128: ... the atmospheric Kelvin wave response ..**

Thank you. We have corrected this (line 139).

23. **line 129: ... the northeast tropical Pacific ...**

We have corrected this (line 189).

24. **line 133: ... strong eastern equatorial Pacific ...**

Thank you. We recomposed the paragraph related to this correction (lines 195-201).

25. **impact of IOD line 223: ... mid-troposphere and zero at the surface**

This was inserted (line 350).

26. **Line 232 .. three dimensional background flow ..**

This was inserted (line 370).

27. **Fig 4: the arrows have units of pseudostress, please explain**

Thank you for noticing and pointing this out. In the revised version of the figure, we changed the quantity to stress, and indicated the proper units (Fig. 4, Supplementary Fig. 5).

Once again, thank you for the constructive comments and suggestions that have further strengthened our manuscript. We are glad to notice that you have already recommended acceptance after minor revisions.

Response to Reviewer #3

This paper is a resubmission of a previously-submitted manuscript concerning the dynamics of super (very large amplitude, eastern-ocean intensified) El Ninos. It is much improved scientifically over the previous version. Here are a list of the improvements. 1) The atmospheric model is now supplemented by an ocean model. 2) The externally prescribed, background state of the atmospheric model, $\psi_b(x,y,z,t)$, is now fixed to the climatological annual cycle. 3) The physics of both models are described in sufficient detail (although I have concerns about both of them; see below). 4) A solution for a super El Nino is now presented, and the contributions of both Pacific and IOD winds explicitly shown. As noted by the authors, new ideas in this paper are: i) the limiting impact on ordinary El Ninos of westerly winds in the far-eastern Pacific; ii) a more-extended analysis of the 2006 IOD event than done previously; and iii) the importance of contemporaneous IOD events in generating super El Ninos. My reservation about these results is whether they are in fact completely new, as there has been so much published on the IOD/ENSO interactions (see, for example, Comment 16). At least to me, the above ideas are new and noteworthy, and they are now presented much more clearly than in the previous manuscript. So, I believe that with a bit more improvement the paper will be publishable. There are several scientific issues, however, that the authors will need to address. In particular, I wonder if the authors have adequately reviewed previous literature (Comment 16), and note that the descriptions of the oceanic and atmospheric models are unclear (Comments 1729). The paper also has many editorial problems that should be eliminated. I point out many of them, but by no means all. Editorial issues include: use of past tense when present tense is better; lack of spaces between words; paragraphs that are too long or should be combined. In some cases, I lost the logical flow of the text, because unrelated (weakly related) text was included. A list of specific comments follows. The list is lengthy, but many of the comments are very easy to respond to. I would like to see the authors' response to all of them, especially the scientific ones.

First of all, we sincerely appreciate the time and energy that you took out to go over the manuscript

with such care and to provide your thoughtful and at the same time critical comments. Your comments have significantly improved all aspects of the manuscript, and we are very grateful for this.

Please note that the line numbers, mentioned below, refer to the version of the manuscript where all changes are tracked; in the tracked version, deleted items are marked by a horizontal line passing through the center of the item; items added in response to Reviewer# 3 comments are in italics and have an yellow background.

1. abstract, line 7: It is dangerous (and egotistical) to use the phrase for the first time, since you may (are likely) to be wrong. In the present case, you discuss the impact of easterly winds in far-eastern ocean associated with ENSO. Such winds have been noted for decades, and I would not be at all surprised if someone has commented on their possible impact on ENSO. I would drop all such statements from the manuscript (there are a couple of other instances)

Thank you for the comment. We have removed the phrase “for the first time” and equivalent phrases from the text.

2. line 21: Perhaps general populace is better than regular people.

We have changed it to “a term familiar to all people”.

3. lines 27-29: The sense of the impact on thermocline depth is wrong. I guess on line 28 you mean thermocline depth anomalies across the Pacific and strong zonal-wind-stress anomalies in the western and central, equatorial Pacific. On line 27, start the sentence with Further, they were all also characterized .

Thank you for the several comments.

- We have added “anomalies” after “thermocline depth” and “zonal wind stress” (lines 28-29).
- Line 27 of the tracked manuscript now starts with: “Further, they were all also characterized”.
- We are not sure that we should hyphenate “zonal wind stress”, since it is common usage and therefore is unlikely to be misread.

4. lines 37 and 44: I, for one, would use "First" rather than "Firstly," as the former is more common usage. To provided a clue to your readers that you are changing the topic, replace "The second" with "Second(ly), there is a."

Thank you for the comments. We have incorporated these (line 47 and line 56).

5. lines 45-48: It would be good to refer your readers ahead to a figure where you illustrate the properties noted here.

Thank you. We have done this (lines 57-58).

6. line 55: The logic of this sentence is not clear. It is better if you replace "mentioned here" with "mentioned, but that was not the case for the 2015 event."

We appreciate the help with clarifying the logic of this sentence; we have incorporated this (lines 67-68).

7. lines 62-66: Does the sentence of this paragraph fit the logical flow of the text. I find it distracting. You could drop it, and write: "... measure of extremity. Given the other differences noted above, we do not consider the 2015 event to be a super El Nino. Our proposed model suggests that super El Ninos occur only when there are extremely strong SST variations in the far-eastern Pacific".

Thank you for the kind suggestion. We have dropped the distracting sentences, and rephrased the statement regarding the 2015 El Nino, as suggested by the referee (lines 80-83).

8. use of "eastern" vs. "far-eastern": In many cases, I think you really mean "far-eastern," rather than "eastern". Please check all usages.

Thank you. We have checked all usages of "eastern", and changed it to "far-eastern" wherever it was appropriate.

9. lines 69-70: Is this sentence really useful, or just distracting.

This sentence was deleted.

10. Perhaps don't start a new paragraph here. Perhaps replace "Many" with "Several."

We have merged the paragraphs (see line 91), and replaced “Many” with “Several” (line 91).

11. line 79: Replace ”will” with ”would.”

We have replaced the word as suggested (line 93).

12. lines 80-81: Replace with ”The event did not materialize despite the supposedly favorable conditions” being present. Clearly much ..” That is, drop the phrase about climate forecasts, which seems unnecessary and distracting.

We have revised the sentence as suggested (lines 94-96).

13. lines 86-87: Don’t start a new paragraph here. Don’t the two paragraphs discuss the same topic?

Thank you for the suggestion. We have merged the two paragraphs (lines 97-113).

14. lines 87-94; discussion of IOD: I realized that one of the reviewers asked you to define an IOD event. I, however, found this discussion very distracting. A simple fix is to put it in parentheses. Also the sentence on lines 92-94 seems like too much information to me, and I think you should drop it. Can you put the IOD discussion into a footnote.

Following on the reviewer’s suggestion, we have rephrased and considerably shortened the definition (of IOD). Further, the lines where we defined IOD is put within parantheses. See lines 105-107.

15. line 95: Replace with ”show that the atmospheric Kelvin wave.”

We have checked similar usages, and added the modifier “atmospheric” as needed. Thank you for commenting about this.

16. Introduction: Have adequately summarized preceding work in the Introduction (and elsewhere). For example, the following are two papers that you did not reference, which seem to me to be relevant: Annamalai et al. (2004; J. Clim., 302-319) and Annamalai et al. (2010; J. Clim., 3922-3952). I bring these two papers to your attention in particular because, as I recall, they conclude that the Pacific response to IOD events is much weaker than in your solutions.

Thank you for the comments. The correct reference, we think, is Annamalai et al (2005), and not Annamalai et al (2004). We have referred to (line 259), and further added a discussion (lines 378-384) about, Annamalai et al's (2005)s work.

As you noted, Annamalai et al (2005) concluded that the Pacific response to IOD events is insignificant; this contradicts our results, although both studies have used the same atmospheric model. The reason for the discrepancy lies in the incorrect specification of IOD's atmospheric heating pattern (their Figure 9b) in Annamalai et al (2005)'s LBM experiments: they assumed that IOD's heating pattern has the form of a dipole; however, this is incorrect. IOD's convective anomalies, from which the horizontal structure of the heating pattern can be inferred, is not a dipole for most of its lifetime: convective anomalies are initiated, and subsequently strengthen, over the eastern Indian Ocean; these generate the surface wind anomalies over the tropical Pacific. Eventually, as warm SST anomalies, driven by oceanic Rossby waves, develop over the central Indian Ocean, western Indian Ocean begins to develop enhanced convective anomalies. At some point around IOD's peak phase, a dipole convective anomaly exists; this leads to IOD's impact vanishing over the tropical Pacific (for example, see Figure 7 c of our manuscript, around the end of November 2006).

However, since Annamalai et al (2010)'s study was based on the wrong conclusions of Annamalai et al (2005), and further did not provide any new insights on IOD-ENSO connectivity, we have not referred to Annamalai et al (2010) in the revised version.

17. lines 364-366: Before proceeding with results, I first comment on the models used. Replace with "smoothed by a 91-day running-mean filter". Replace with "An intensity threshold of 4 m/s is used to discriminate WWB events, but for simplicity we did not apply a threshold in event duration."

Thank you for the comments. We have revised the two sentences as suggested by the reviewer (line 561 and lines 563-564).

18. The ocean model: Please reword the first two paragraphs in "The ocean model" section. Combine the two paragraphs. On lines 370-373, replace with: "It consists of a linear, 1.5layer (shallow-water) model, which simulates the first-baroclinic-mode response of the ocean, as well as a linear, empirical SST equation. The 1.5-layer model consists of ..."

Thank you for the comments. We have revised the paragraphs as suggested (lines 567-569).

19. lines 374-376: Replace with “The model equations are” and add commas after all the equations (they should be regarded as part of the sentence). On line 376, replace with ”the wind-stress forcings are τ_x and τ_y , $f = \beta y$ is the Coriolis parameter in the equatorial-plane approximation, $g' = g\rho^{-1}\delta\rho$ is the reduced-gravity coefficient, and .”

Thank you. The suggested changes were incorporated (lines 573-575).

20. line 383: Why is the grid spacing so large? Surely, you could use a resolution of 2550 km. If you use $\Delta x = 2^\circ$, $\Delta y = 1^\circ$, then I think you need to tell your readers why you think you can get away with such large values.

Thank you for asking us to clarify this. As implied by the reviewer, this resolution may be too coarse for equatorial-ocean modeling in general, and hence there is a need to clarify this to the reader.

Here, we specifically focused on low-frequency ENSO scale variations - a meridional resolution of 1° is sufficiently high to capture the equatorial Kelvin and Rossby waves which are important in El Nino dynamics, and do not introduce any noticeable distortions due to effects of spatial finite-differencing; further, zonal scales related to El Nino variability are much larger - a zonal resolution of 2° is more than sufficient. Following the reviewer’s suggestion, we have added lines 617-622 to clarify this issue.

21. lines 383-386: It is not logical to discuss equation (5) here. So, move this paragraph to the end of subsection after line 398.

Thank you for pointing this out. We have moved the said paragraph to the end of the subsection (lines 614-617).

22. equation (5): Does (5) make physical sense? Such a representation for T is common in analytic ENSO models, but I have never seen it used in a numerical model. I can understand the impact of h on T in the eastern Pacific where the thermocline is shallow, and h impacts the upwelling of subsurface water. I can also understand that easterly winds ($\tau_x < 0$) could impact T by Ekman divergence

and upwelling, but westerly wind ($\tau_x > 0$) should have no effect of this sort. I have no idea how τ_y can impact T. If you set $\gamma = 0$, do your results change? So, I think you need a better justification of (5). The fact that α , β , and γ depend on x is good. For example, I suppose that $\alpha(x)$ is set so that it is appreciable on in the eastern Pacific. Shouldnt those coefficients also be functions of y ? For example, the α -term should only apply in the Pacific cold tongue.

Thank you for the comments. We are sorry that our description of equation (5) was a bit sketchy, especially with regard to the second term on the right hand side. The second term models the impact of zonal advection; specifically the advection of mean zonal temperature gradients by anomalous oceanic zonal currents. This is affected by surface currents, which is not the same as the layer-averaged zonal current (u) simulated by the model. Previous studies, for example, Boulanger (2001) show that oceanic zonal surface velocities are poorly represented by a 1.5-layer shallow-water model, even if an Ekman layer is added. Further, there is some observational evidence (e.g. Burgers and van Oldenburgh, 2003) that the relation between surface wind and observed zonal surface velocities is slightly more accurate than that between the observed zonal surface velocity and the velocity of the 1.5-layer linear model. Therefore in this model, the impact of zonal advection is modeled by a term linearly related to the zonal wind stress τ_x , instead of u .

We have significantly enhanced the discussion of equation (5) to make this clear (lines 593-613).

Following the reviewer's suggestion, we also ran an experiment where we set $\beta = 0$ (the reviewer mentioned changing γ , but we think the reviewer meant β , since this term was the focus of the reviewer comment). Supplementary Figure 19a shows the impact of switching off this term: it leads to poor simulation of SST anomalies at the edge of the Pacific SST warm pool.

As noted by the reviewer, the coefficients of equation (5) only depend on longitude; ideally they should also vary as a function of latitude. However, for simplicity such a dependence was not incorporated, considering that the focus of this investigation is confined to SST variations close to the equator; close to the equator (between 5°S and 5°N), the correlation between observed and simulated SST anomalies are reasonably high (>0.7), given the simplicity of the model's dynamics and its SST equation.

23. lines 404-406: The most important aspect of the LBM is its linearity, so I would mention that first. Replace with: “The LBM is derived by linearizing the AGCM about a specified background state, $\psi_b(x,y,z,t)$, eliminating water vapor, and dropping the model’s radiation scheme. The diabatic heating that forces the atmosphere is then represented by an externally prescribed function, $Q(x,y,z,t)$.”

Thank you for the kind suggestion; they were incorporated as suggested (lines 628-631).

24. lines 412-413: Replace with: “The background state $\psi_b(x,y,z,t)$ is set to the seasonal climatology of the NCEP reanalysis data.”

Thank you. We have incorporated the suggestion (lines 631-632)

25. lines 414-442: Please carefully go over the entire discussion in these three paragraphs to check their correctness and to improve their clarity and logical flow.

Thank you for carefully checking these statements, and for the kind suggestion. We have reorganized the discussion by stating the general features of the LBM in the first paragraph (lines 624-640), followed by the details of the experimental runs in the second (lines 647-658). Further, we have removed equation (6) and accompanying discussion in response to the reviewer comments 26-29.

26. line 427 and elsewhere; April: Do not hardwire your discussion to the month of April. Let ψ_0 be your initial state variable, rather than $\psi(\text{April})$, and then you can refer to later times as ψ_{n-1} and ψ_n , and so on.

Thank you for the suggestion. This and other related discussions were removed in response to reviewer comments 26-29.

27. method of integrating the LBM: Rather than integrating the LBM forward in time continuously, the authors split the time domain into a series of steps. The interval between steps is not clear from what is written. Is it 10 days, one month, or some other interval? Please state the interval in the text. But, why run the model in this stepy manner? I guess because their LBM code is set up that way. If so, why didnt you just modify the code to run continuously?

Let t_n indicate a particular time step. Then, at that step the LBM is run out to equilibrium with $Q(x, y, z, t_n)$ and $\psi_b(x, y, z, t_n)$, and (I guess) ψ_n is this equilibrium value. This process is okay because the Rayleigh friction and Newtonian cooling time scales are so small, that is, $\nu^{-1} = \kappa^{-1} = 1$ day. As a result, the LBM rapidly comes into equilibrium in a time scale only somewhat longer than 1 day. Since the forcing function, Q and ψ_b , vary on a much longer time scale of the order of a month (30 days), the LBM response is essentially in quasi-equilibrium.

Thank you for the comments and the kind explanation. This and related discussions were removed in response to reviewer comments 26-29. Please also see our response to reviewer comment 29 below.

28. initial conditions at step t_n : The authors assume that they need to start the integration for ψ_n with a non-zero initial condition, which (I guess) they take to be ψ_{n-1} . But, since the system is in quasi-equilibrium (see previous comment), no initial condition is needed. In only a few days, the system adjusts to equilibrium with the forcing functions Q and ψ_b , and completely “forgets” the initial condition.

Thank you for the comments and the insight. This and related discussions were removed in response to reviewer comments 26-29. Please also see our response to reviewer comment 29 below.

29. equation (6): Equation (6) appears to be a method for interpolating between time steps. It is impossible to tell, however, because the equation *makes no physical sense at all* since its left- and right-hand sides have different units (units of ψ_b vs. units of $\psi_b Q$). So, I have no idea how to interpret or fix it. On the other hand, because the LBM solution is essentially in quasi-equilibrium with the slowly-varying forcing functions, I don't think any version of (6) is actually needed. It should be dropped. In summary, the authors simply need to calculate n without any initial condition ($\psi_{n-1} = 0$ will do). Calculating in this simple way should reasonably approximate the solution to a continuous integration. Given that (6) is not clear, I can't tell whether the LBM solutions reported in the text are sensible. This issue must be addressed.

Thank you for the comments. We apologize for the confusing equation (6) and the incomplete discussion related to it. There were some errors in the manuscript sent out for review (equation 6

was wrongly typeset, and some lines discussing the rationale for that equation were not included).

Let me explain.

The original intent, in introducing this equation, was our concern about the propagation of initial conditions (we now understand that our arguments were incorrect). Equation (6) was meant to be typeset as:

$$\Delta\psi = \Delta\psi_b \times \frac{\partial\psi}{\partial\psi_b} + \Delta Q \times \frac{\partial\psi}{\partial Q} \quad (1)$$

This followed from treating $\psi(\psi_b, Q)$ as a function of the two independent variables, ψ_b and Q , expanding into its Taylor series about ψ_b and Q , and retaining only first order terms.

In the runs described in the previous version of the manuscript, the first term on the r.h.s was estimated by integrating the LBM with Q^{n-1} as the thermal forcing and $\Delta\psi_b = \psi_b^n - \psi_b^{n-1}$ as the specified background state. Similarly the second term was estimated by running the LBM with $\Delta Q = Q^n - Q^{n-1}$ as the thermal forcing and ψ_b^n as the specified background state.

In our view, this method is all right, as it is - however, it does not serve the purpose for which it was intended. The solution ψ^n for the next state (month, in our case) obtained by adding ψ^{n-1} to to equation (1) is nearly similar to ψ_n obtained by integrating the LBM with $Q(x, y, x, t^n)$ as the thermal forcing and $\psi_b(x, y, z, t^n)$ as the background state; likely, the solutions obtained by using equation (1) are smoother, since higher-order terms were dropped. However, it is not at all relevant to the issue of the propagation of initial conditions. In light of this, we have dropped the method of LBM integration indicated by equation (6), and as recommended by the reviewer just used the LBM in the standard way; for example, similar to Annamalai et al (2005)'s use of the LBM.

We are very grateful that the reviewer went into great lengths to explain the errors in our logic, and for the generous and constructive suggestions.

30. lines 152-160: In this paragraph you state what you will be doing in the subsequent section. The paragraph should be moved to line 162, as a introduction to the section. Suppose you are a reader trying to reread Section "The impact of IOD on the equatorial Pacific." You would want to read the

paragraph on lines 152-160, but would not find it. Generally, for this reason it is bad practice to introduce the next section in a last paragraph to the preceding one.

Thank you for the suggestion. We have repositioned the paragraph as suggested by the reviewer (lines 223-231); further, we rephrased the entire paragraph to improve clarity.

31. lines 169-171, 185, and elsewhere; direct vs. indirect evidence: I do not understand the distinction between direct and indirect evidence. I guess you are arguing that the 2006 was a pure an IOD event (no EL Nino at all) as possible, and in that way your analysis is better (more direct) than previous ones. Really? Why not drop this confusing nomenclature, and just write something like: Here, to supplement previous work we analyze in detail a situation during 2006, a year with a prominent IOD event but very weak or no El Nino response.

Thank you for the comments, and suggestions. Following on the reviewer's advice, we have rephrased the sentence (line 281 - 283) as follows: "Here, supplementing previous work, we clarify IOD's impact on the equatorial Pacific through data analysis and numerical simulations; to demonstrate its significance, we also analyze in detail a situation during 2006 - a year with a prominent IOD event but very weak or no El Nino response"

32. line 185: Rewrite the text to avoid using "for the first time."

We have deleted this paragraph, and the preceding one, since we that thought that they seemed distracting.

33. line 192: Regarding the relative sizes of convective variations in the Indian and Pacific Oceans, is comparing their areal average values appropriate. After all, the area of the Pacific Ocean is much larger than that of the Indian Ocean.

Thank you for the comment. In light of the comment, we have added a sentence (lines 305-306), stating that the tropical Indian Ocean domain is only about 60% of the area of the tropical Pacific.

34. lines 197-198: Dont start a new paragraph here, as the following two sentences are related to the topic of the previous paragraph.

Thank you for the comment. We have, in the current revision, reworked the whole section, considerably, and with great care, taking into account the comments of Referees 2 and 3.

We consider different aspects of IOD convection in separate paragraphs - one (lines 298-306) discusses the amplitude of these anomalies, and the succeeding one (lines 315-330) discusses the changing structure of IOD's convective anomalies. The latter aspect was originally discussed in lines 234-237 of the previous manuscript.

35. line 201: Start a new paragraph with the sentence: “To demonstrate the IOD ...”; it does start a new topic. ”

Thank you for the comment. This content was integrated into the paragraph starting 298.

36. lines 203-204: Dont start a new paragraph here.

Thank you, and we agree.

37. line 204 and many places elsewhere: You wrote “(Fig. 6(a)-(c))” here, and there are similar expressions elsewhere. For one thing, there are too many parentheses. For another, “Fig.” should be plural. So, replace with “(Fig. 6a-6c) here and similarly elsewhere. ”

Thank you for the comment and the suggestion. We have implemented this suggestion.

38. line 209: You note here that the IOD winds are strong over the western Pacific. How strong relative to ENSO winds? Can you please compare them? My impression is that IOD winds are much weaker than ENSO winds.

Thank you for raising this question. We have added new Supplementary Figures 7 and 9 to discuss this; further, a statement regarding the strength of IOD vs El Nino winds over the far-western Pacific was added (lines 326-328).

39. lines 212-223; overview of the LBM: I expect that this discussion will change once the description of the LBM is fixed in the Methods section. In any case, try to shorten it. Move line 230, which is logically out of place where it is now located, to line 216, replacing nearby text with something

like: “It is linearized about a background state, $\psi_b(x, y, z, t)$, which we set to be the annual cycle from the NCEP reanalysis⁴⁵ climatology.”

Thank you for the suggestion. We have implemented the reviewer suggestions (lines 343-352).

40. lines 238-239: Lots of flowery words here, so much so that for me the meaning of the sentence is lost. Also, this first sentence is really a justification to study the 2006 events, so if you keep it, it belongs at the beginning of the section.

Thank you for the comment. We have dropped this line.

41. line 241: I have never seen the word Noteworthy used in an oceanographic paper before. A google search, however, confirms that it is in fact a real word, albeit one that is not commonly used. Perhaps replace it.

Thank you for the comments. The word was dropped. Further, the discussion in the paragraph starting on line 238 (in the previous version) was rephrased, and expanded for clarity (see the two new paragraphs starting on line 396 and 406 of the revised manuscript).

42. line 261: You tend to use emotional language in the text, which should be avoided in scientific writing. So, I would drop strongly from the sentence here. Also, on this line you wrote west Pacific, and there are similar usages elsewhere. I think you should replace such expressions with western Pacific. Alternately, but not as good, you might write West Pacific.

Thank you for the comments. We have reduced the usage of emotional language throughout the text, and have dropped “strongly” from that sentence (line 437). Also, we have replaced usages of “west Pacific” with “western Pacific”.

43. line 266: Again emotional language. I would drop overwhelming. Frankly, I don't feel overwhelmed by what you presented. I do feel, though, that you have made a strong and convincing argument. Leave the decision to be “convinced” or “overwhelmed” to your readers.

We agree with you, and have removed those words (line 444).

44. line 278-279: The complete sentence on these lines is not physically precise. Drop it.

Thank you for the suggestion. We have dropped this line.

45. lines 294-296: Why does the SSHA appear first at the eastern boundary? Your explanation only applies to SSTA. It must result from reflected Rossby waves, which increase the eastern-boundary response from that of the incoming Kelvin wave. You might note this property.

Thank you for the suggestion. We have implemented this (lines 471-480).

Once again, thank you for giving us the opportunity to strengthen our manuscript with your valuable comments and suggestions. We have worked hard to incorporate your feedback and hope that these revisions persuade you to accept our submission.

References

- [1] D. B. Enfield, A. M. Mestas-Nunez, and P. J. Trimble. “The Atlantic multidecadal oscillation and its relation to rainfall and river flows in the continental US”. In: *Geophysical Research Letters* 28 (2001), pp. 2077–2080.
- [2] J. Cohen. “Weather forecasting: El Nino dons winter disguise as La Nina”. In: *Nature* 533.7602 (2016), pp. 179–179.

Reviewers' comments:

Reviewer #1 (Remarks to the Author):

I am totally confused with this article. The results are inconclusive, the mechanisms proposed are incomplete, there are logical inconsistencies in the arguments. So, I am sorry, I cannot be very positive on this paper. I have provided my responses to authors comments and my concerns in the followings. I am not against the interaction between IOD and ENSO, but the study needs substantial improvements (by considering other factors such as the Pacific decadal variability) to clearly postulate a model for super El Nino.

Authors:

The classification of the 1972, 1982, and 1997 events as super El Ninos are based on the objective analyses by Hong et al. (2014), Takashi and DeWitte (2015), and Li et al. (2015); the first study is also referred to by other investigators who have discussed super El Ninos, viz., Chen et al. 2017 and Latif et al. 2015. Thus, the identification of 1972, 1982, and 1997 as super El Ninos have a strong, objective basis, and is not unique for this study.

Reviewer:

I suggest authors to carefully read these papers, particularly Chen et al. (2017). Figures in Chen et al. (2017) suggest 2015 to be a strong event comparable to 1982 and 1997 if not stronger than them.

Authors:

We have added a few lines of discussion to clarify this point (lines 30-38 in the tracked version of the manuscript).

Further, the following were added to enhance the comparison between 2015 and other El Nino events:

Supplementary Figure 2, which is new, compares El Ninos since 1979 using OLR anomalies. Supplementary Figure 1, also new, compares Indian Ocean response during the 1997 and 2015 El Ninos.

Reviewer:

IODs are not necessarily symmetric around the equator. This fact is acknowledged in the caption of your Fig. 7. So, it is not clear to me why the anomalies are averaged between 5S and 5N in the Supplementary Figure 1. Nonetheless, Sup. Figure 1g does show a strong SST gradient like a positive IOD in 2015 (even stronger than that of 1997). Why was the 2015 El Nino not affected by that?

Authors:

The reviewer also has asked about "the statistical significance of the composite of those three events shown in Fig. 2". Regrettably, the reviewer is mistaken about Fig. 2: this figure is a plot of the seasonal cycle of standard deviation of SST and OLR. Thus, we cannot address the question; it is not meaningful in the context of Fig. 2.

Reviewer:

That was a typo. I meant Fig. 4. It should not have been a problem for the authors to pick that figure number since that is the only composite plot in the main part of this manuscript.

What is the statistical significance of Fig. 4? Also, what is the statistical significance of IOD composite anomalies used to force the ocean model for the results shown in Supplementary Figs. 13 and 14? The forcing used could be miss-leading unless the forcing has a statistical significance to actually represent all the events. If there is no statistical significance, it is better to pick representative years rather than

use the misleading composite anomalies.

Authors:

Further, Chen et al. (2017) and Latif et al. (2015) both refer to Hong et al. (2014) in discussing super El Ninos; Hong et al (2014) is the objective analysis that identified 1972, 1982, and 1997 as super El Ninos. Since, such an analysis has already been undertaken, and is used in other studies, we do not agree with the reviewer that we need to develop a new identification scheme for super El Ninos.

Reviewer:

I suggest authors to carefully read these papers, particularly Chen et al. (2017). Figures in Chen et al. (2017) suggest 2015 to be a strong event comparable to 1982 and 1997 if not stronger than them.

Authors:

Regrettably, the reviewer is incorrectly rephrasing our conclusions: we did not claim that a strong IOD can lead to a strong El Nino; what we did claim was that a strong IOD can create conditions that favour a moderate, transient warming of the tropical Pacific. We also concluded that interaction between positive IOD and El Nino is needed for super El Ninos to develop - all the super El Ninos so far (1972, 1982, and 1997) satisfy these conditions.

Reviewer:

I did not say authors are claiming that all strong IODs are leading to strong El Ninos. My question was if one of the strong IOD events could lead to a strong El Nino event, why it does not happen for some other strong IOD events. I am not against the suggestion that IOD and ENSO interact and that could lead to stronger El Nino events. But the results presented here are not sufficient to prove the point while excluding any other possibilities.

This is what is written in the first paragraph of your manuscript: "Here, combining observational analysis and numerical simulations, we demonstrate that all major features of a super El Nino result from the interaction of an El Nino event with a positive Indian Ocean Dipole event. "

Which are the years for these interactions leading to super El Ninos? 1982 and 1997 (since not much analysis is provided for 1972 in your study)? Why there is no other strong events after the 1997 event in spite of having several positive IODs together with El Ninos. The discussion on just one event of 2006 is not very conclusive. The results are not robust to argue that there was no coupled feedback in tropical Pacific without a coupled model experiment. The authors did not discuss at all about 2012 when there were an El Nino and a positive IOD. What prevented that event to become a super El Nino? Is it not the decadal variability in the Pacific that constrained El Nino and super El Nino to evolve post 1997? I would also suggest to show the IOD and NIno indices together for the whole analyses period to see which others year could have (or not have) the possibility of such interactions.

You also said in the first paragraph: "We also demonstrate a self-limiting behaviour inherent in El Nino Southern Oscillation (ENSO) dynamics: this, a consequence of the atmospheric Kelvin wave response that develops to the east of ENSO's convective anomalies, dampens SST variability in the eastern Pacific and prevents super El Ninos from developing through tropical Pacific dynamics alone."

In this context, the discussions in the first paragraph of page 5 are relevant. In line 112 of that paragraph you said "However, this remote effect (on SST) is countered by easterly wind anomalies that are present to the east of the convective anomaly (Figures 1a,e and Figure 3); these, the atmospheric Kelvin wave response to the convection anomaly, favour local equatorial divergence and

upwelling, which offsets the downwelling effect of oceanic Kelvin waves arriving from the west.

Which easterly wind anomaly? The shown easterly wind stress anomalies are asymmetric around the equator and the peak anomalies are off-equatorial. Is it a Kelvin wave response? Additionally, how these anomalies and the self-limiting mechanism are destroyed or limited by an ensuing positive IOD? Which are the El Niño years when this limiting mechanism worked?

Authors:

The nature of SST anomaly variation during the strong El Niño of 2015 further support our model - viz., super El Niños can only develop when IOD co-occurs with El Niño. The confusion regarding the 2015 El Niño is related to its strong SST anomaly amplitude in the west and central Pacific; however, this is weak in the far-eastern Pacific. In other words, the strong eastern-ocean intensification that is a characteristic feature of super El Niños is absent for the 2015 El Niño. The weakness of its SSH and wind anomalies compared to the 1997 event were presented in Fig. 1.

Reviewer:

A comparison between two does not rule out the fact that one of them cannot be categorized as super El Niño unless we compare all the events in super El Niño category. Based on the figures shown in Chen et al. (2017), I see 2015 event has SST and WWE anomalies comparable to 1997 and 1982 events (please refer to Figs 1, 2 and 3 of Chen et al. 2017). And Chen et al did argue that the 2015 is a Super El Niño. So, please compare your results with them.

Authors:

Supplementary Figure 2, further, shows that the OLR anomalies during 2015 are one of the weakest in the available observational record. These disparities between the amplitude of SST anomaly and that of other ocean-atmospheric anomalies during 2015 call for a different explanation of the strength of its SST anomalies - recent studies, and our own assessment (Supplementary Fig. 3) points to the role of a longer time scale (decadal or climate change time scale)

Reviewer:

I don't see a reason why OLR anomalies should be used to compare El Niños defined by SST indices. Please show the amplitudes of Niño3 and Niño3.4 (instead of OLR anomalies above that region) to compare the strengths of El Niños. And please compare your results with that of Chen et al. 2017.

Reviewer #2 (Remarks to the Author):

Recommend: minor revision

The authors have made further substantial changes in response to the reviewers' comments, thus improving their article.

With further revision to take care of some details, and address one issue described below, this will be a paper of interest and worthy of publication.

Specific comments:

Over the course of several revisions, a lot of supplementary material has become mixed in with the main article without clear separation. Some rearrangement to make the distinction clearer would be

welcome. Perhaps much of the detail in 'methods' could be moved to 'supplementary' to allow some material into the main article, and perhaps some experiments could be moved to 'supplementary' in a self-contained way.

The issue is with the conclusion that the 2006 El Nino was forced by IOD dynamics, and that westerly wind bursts were not evident. In Fig. 7c there are westerly wind episodes in the west Pacific in July, August and particularly October 2006, each of which contributed to an ocean response that led to SST increases in the far eastern equatorial Pacific, and to central Pacific SST increase after the October event. I am surprised that none of these met the westerly burst criterion set by the authors.

The corresponding observations and simulations with the atmospheric model shown in Fig. 8a-d seem to have a different timescale, with some unspecified time averaging that emphasises longer timescales – this needs explanation. The evidence presented suggests that the atmospheric anomalies in the east Indian ocean region could have had a substantial influence on the western Pacific winds, but does not rule out a role for higher frequency wind-burst-type events.

Minor comments and suggestions (line numbers as in the tracked-changes version):

line 24: ... anomalies associated with ...

line 202: Next, in place of Then,

line 209: better to say 'Here the cold tongue cools ... while expanding ...'

line 283: .. prominent positive IOD ...

line 318: is marine cloudiness a good (or at least reasonable) indicator of deep convection?

Line 400 and following: it would be more reasonable to say that the transition to warm Nino3.4 was correctly predicted, but the sharp increase in Sep-Oct was not captured, and the sharp decline in Dec-Jan was underestimated.

Note that many forecast models contain representation of Indian Ocean processes, so it would be worth recommending analysis of forecast behaviour in that region in the discussion.

Line 463: further rather than once again

line 466: ... positive IOD events

line 515: delete 'It is only reasonable to assume that'

line 619: $2c/\beta$, and 470km seems on the high side

Supplement:

define DMI

Supp Fig 6: should this correspond to Fig 6d? Is this May-June-July forcing?

Supp Fig 7 caption: 'correlation' in place of 'relative strength'

Supp Fig 10: are these all 3-month averages (including observed)?

Reviewer #3 (Remarks to the Author):

SUMMARY: The authors responded to all the concerns in my second review favorably. I now have no scientific objections to publishing this paper. In reading over the manuscript, however, I still have editorial concerns (confusing text, somewhat awkward English, etc.). Below, I note a few examples. I have not gone over the paper in detail, carefully reading all the changes to the text, so I expect there are more. So, I request that the authors read over all the new text that they have written (as well as old text) several times. Be sure the text is in good English, and that it flows logically. After these editorial concerns are addressed, I believe the paper will be acceptable for publication. I do not need to see it again.

A brief list of specific editorial comments follows. There are more, which I leave up to the authors to find and eliminate.

SPECIFIC COMMENTS:

1. abstract, last 4 lines: Replace with: "...(ENSO) dynamics: the atmospheric Kelvin wave...convective anomalies dampens SST variability in the eastern Pacific,thereby preventing...".
2. line 36: There should be a comma after "analysis."
3. line 50: The hyphen should be a minus sign, that is, "--40 Wm⁻²."
4. line 56: Since you replace "Firstly" rather than use "First," you should replace "Secondly" with "Second."
5. line 60: There should be no space before "weak."
6. line 82: Does it make sense to add the phrase "As we shall see," at the beginning of the sentence? Maybe.
7. line 88: Isn't "nonlinearity" a single word that does not need a hyphen?
8. lines 106 and 107: Replace with "Indian Ocean, and this anomaly pattern."
9. line 107: Replace with "Second, we show."
10. line 108: Replace with "ENSO dynamics, transforming a developing."
11. lines 128--142: This new text is very confusing to me. The first line of the highlighted text states that: "This [weakening] manifests a self-limiting aspect of ENSO dynamics, the physical reason for which is as follows." But, the text that follows does more than talk about self-limitation, as it first discusses Bjerknes feedback. That is not logical, and as a result it took me several readings to figure out your intended meaning. You need to clarify this paragraph. Since it really discusses textit{two} ideas, you need to provide some clue to your readers that is what you are doing. One approach might be to split the paragraph into two separate ones. Another might be to introduce "First" and "Second" at the beginning of relevant sentences. Regardless, this text needs to be clarified.
12. line 154: Doesn't "ocean" need to be capitalized?

13. line 161: Replace with "to zero over the far-eastern Pacific (15°S -- 15°N , 120°W -- 80°W)."
14. lines 214 and 224: In each sentence you use "this" without a noun. It is not quite clear to what they refer. Perhaps replace the first with "this difficulty" and the second with "Their existence requires the presence of." There are other similar places in the text where the meaning of "this" is not clear.
15. lines 250--252: Shouldn't the past tense usage be present tense?
16. line 254: Replace "on this" with "on these properties."
17. line 458: I don't think you mean to start a paragraph here.
18. line 574: Don't start a paragraph here. Replace with "where τ_x and τ_y are the wind-stress forcings,...".
19. line 593: Don't start a paragraph here.
20. line 617: Replace with "America. The model resolution."
21. lines 630--632: Replace with: "radiation scheme. The background state $\psi_b(x,y,z,t)$ is set to the seasonal climatology of the NCEP reanalysis data. The diabatic heating that forces the atmosphere is represented by an externally prescribed function, $Q(x,y,z,t)$." Just too many semicolons. By the way, elsewhere you overuse semicolons. Perhaps split them off into separate sentences.

Summary of additional comments from referee #2:

This referee felt part of the disagreement arose from confusing definitions. It is clear the classification of El Nino events can be subjective, and that a large part of the community regarded the 2015/16 event as a super El Nino. Therefore, we feel this needs to be recognised in the manuscript, and abstract, and the various El Nino events need to be clarified in this context. One option referee #2 proposes is to classify 2015/16 as a super El Nino while using the term super East Pacific El Nino for other events, and explaining why 2015/16 was different to other East Pacific events.

At the same time the approach to define Indian Ocean Dipole (IOD) events is non-conventional and could lead to confusion for the reader. Based on the standard SST index, 2015 would be classified as a positive IOD event, and this should be recognised in the manuscript. Referee #2 agrees with referee #1 that you should use another atmospheric index for the relevant Indian Ocean region atmospheric state to make the important atmospheric component clear (using the term IOD anomalies to refer to these is confusing and should be avoided).

At the same time this referee also reiterated that other factors, such as westerly wind bursts need to be discussed and the manuscript should make it clear that the Indian Ocean Dipole is one of several processes and not the controlling factor. In this regard a possible title would be: "Indian Ocean processes that influence major El Nino events"

1 Response to Referees

1.1 Response to Referee #1

We thank the reviewer for the comments on the revised version of the manuscript.

Rev#1

I am totally confused with this article. The results are inconclusive, the mechanisms proposed are incomplete, there are logical inconsistencies in the arguments. So, I am sorry, I cannot be very positive on this paper. I have provided my responses to authors comments and my concerns in the followings. I am not against the interaction between IOD and ENSO, but the study needs substantial improvements (by considering other factors such as the Pacific decadal variability) to clearly postulate a model for super El Nino.

The reviewer's statement that "The results are inconclusive, and the mechanisms proposed are incomplete, there are logical inconsistencies in the arguments" would have been more useful, if the reason for each of these judgments (inconclusive, incomplete, inconsistent) was detailed.

In the revised version, we have considered Pacific decadal variations, as suggested by the reviewer. In Supplementary Fig. 1, we show interannual and decadal Nino3 SST anomalies from 1955 to 2010; Supplementary Fig. 18 shows estimated decadal variations in globally averaged and tropical Pacific averaged SST anomalies. Supplementary Fig. 20 shows interannual Nino3 SST anomalies from two different SST datasets. Supplementary Fig. 22 shows interannual DMI series from 1958 to 2015.

Rev#1

Authors: The classification of the 1972, 1982, and 1997 events as super El Ninos are based on the objective analyses by Hong et al. (2014), Takashi and DeWitte (2015), and Li et al. (2015); the first study is also referred to by other investigators who have discussed super El Ninos, viz., Chen et al. 2017 and Latif et al. 2015. Thus, the identification of 1972, 1982, and 1997 as super El Ninos have a strong, objective basis, and is not unique for this study.

Reviewer: I suggest authors to carefully read these papers, particularly Chen et al. (2017). Figures in Chen et al. (2017) suggest 2015 to be a strong event comparable to 1982 and 1997 if not stronger than them.

We now reproduce Fig. 1 of Hong et al. (2014), whose objective analysis revealed the unique nature of the 1972, 1982, and 1997 El Nino events. In their Fig. 1, they show that interannual Nino3 anomalies

for these events were 1 standard deviation above other El Ninos, in the 1950–2010 period they analyzed. Our Supplementary Fig. 1 reproduce the results of Hong et al. (2014).

We have also extended their analysis upto 2015 in Supplementary Fig. 20, and show that the magnitude of the 2015 El Nino is one standard deviation lower than the super El Ninos (1972, 1982, 1997).

Rev#1

Authors: We have added a few lines of discussion to clarify this point (lines 30-38 in the tracked version of the manuscript). Further, the following were added to enhance the comparison between 2015 and other El Nino events: Supplementary Figure 2, which is new, compares El Ninos since 1979 using OLR anomalies. Supplementary Figure 1, also new, compares Indian Ocean response during the 1997 and 2015 El Ninos.

Reviewer: IODs are not necessarily symmetric around the equator. This fact is acknowledged in the caption of your Fig. 7. So, it is not clear to me why the anomalies are averaged between 5S and 5N in the Supplementary Figure 1. Nonetheless, Sup. Figure 1g does show a strong SST gradient like a positive IOD in 2015 (even stronger than that of 1997). Why was the 2015 El Nino not affected by that?

We have revised the figure mentioned by the reviewer, and an average between 10S and Eq is taken for SST, SSH and OLR anomalies in the revised Supplementary Fig. 21.

Reg. the question, why the 2015 El Nino was not affected by the strong SST gradient seen in Sup. Fig. 1g, we have the following explanation.

IODs are characterized by cool SST anomalies in the eastern Indian Ocean during most of their lifetime [Fig. 2 of Saji et al (1999)]. Warm SST anomalies develop later in the central and western Indian Ocean (Saji and Yamagata, 2003). The DMI that we (Saji et al., 1999) had designed as an index for IOD was intended to capture this zonally varying SST anomaly structure during IOD.

Note that in our paper (Saji et al. 1999), where we originally defined the DMI, we had removed decadal variations from the monthly SST anomalies. In general, many studies do not remove decadal variations, and confuse strong SST gradients in some years as an IOD event. To clarify the issues associated with IOD detection from SST data, we published an analysis in 2003 (Saji and Yamagata, 2003; Journal of Climate), wherein the characteristics of Indian Ocean SST were discussed in the context of IOD detection. Here, we show that decadal variations, if not removed, can create elevated DMI values that are not necessarily associated with IOD. Further, we show that a basin-wide Indian Ocean SST

anomaly generated by ENSO can also give rise to spurious DMI values. *Please see our upcoming article in Oxford Research Encyclopedia (Hameed, 2017; in press; recently I use Hameed as my last name, not Saji) that discuss these issues in more detail.*

To summarize, IOD is characterized by cool SST anomalies over the eastern tropical Indian Ocean – zonal SST gradients need not necessarily imply that an IOD event is taking place. Suppl Fig. 21 (in current version of manuscript) show the actual conditions of the Indian Ocean, and by doing so demonstrates that an IOD event was not present during 2015. However, this may be unnecessarily confusing, since the DMI for 2015 would show a positive excursion, as pointed out by the reviewer. To clarify this, we have added Supplementary Fig. 22 which shows interannual DMI time series, using both monthly and seasonally averaged anomalies. IOD events peak in boreal fall, and a large DMI value during SON is a reliable indicator of IOD in that year. These confirm that an IOD event was not present during 2015: the SON value of DMI is one of the lowest among all years, although monthly DMI is elevated from November of 2014 and peaks in July 2015. After July 2015, DMI plunges sharply and falls to zero in October – the reviewer seems very knowledgeable about IOD, and it should be obvious that this behaviour of DMI is not representative of an IOD event.

Rev1

Authors: The reviewer also has asked about the statistical significance of the composite of those three events shown in Fig. 2. Regrettably, the reviewer is mistaken about Fig. 2: this figure is a plot of the seasonal cycle of standard deviation of SST and OLR. Thus, we cannot address the question; it is not meaningful in the context of Fig. 2.

Reviewer: That was a typo. I meant Fig. 4. It should not have been a problem for the authors to pick that figure number since that is the only composite plot in the main part of this manuscript.

We concede that it could have been a mistake, but for the fact that the reviewer had specifically asked about “a composite of those three events” – Fig. 4 (of the previous manuscript) composited four El Nino events.

Rev1

What is the statistical significance of Fig. 4? Also, what is the statistical significance of IOD composite anomalies used to force the ocean model for the results shown in Supplementary Figs. 13 and 14? The forcing used could be miss-leading unless the forcing has a statistical significance to actually represent all the events. If there is no statistical significance, it is better to pick representative years rather than use

the misleading composite anomalies.

The purpose of showing statistical significance is to highlight the robustness of a feature under consideration. In the case of ENSO, countless studies have already examined its characteristics. The El Nino events cited are well-known and accepted (http://origin.cpc.ncep.noaa.gov/products/analysis_monitoring/ensostuff/ONI_v5.php). Anyway, Fig. 2 (in current version of manuscript), already demonstrated the statistically robust nature of the zonal wind anomaly features (esp. easterly anomalies in the eastern Pacific) associated with El Nino that we discussed in our work through a correlation analysis.

For the IOD, the feature of relevance would be the westerly anomalies that it forces over the Pacific. That this feature is significant can be seen from the data presented in Fig. 5 and Supplementary Figs. 6–8 (numbers as in revised version). Also, the statistical robustness of these features were discussed in the observational analysis of Saji and Yamagata (2003b), about a decade and a half ago.

Rev1

Authors: Further, Chen et al. (2017) and Latif et al. (2015) both refer to Hong et al. (2014) in discussing super El Ninos; Hong et al (2014) is the objective analysis that identified 1972, 1982, and 1997 as super El Ninos. Since, such an analysis has already been undertaken, and is used in other studies, we do not agree with the reviewer that we need to develop a new identification scheme for super El Ninos.

Reviewer: I suggest authors to carefully read these papers, particularly Chen et al. (2017). Figures in Chen et al. (2017) suggest 2015 to be a strong event comparable to 1982 and 1997 if not stronger than them.

Since we discuss our response to the same comment from the reviewer earlier in this rebuttal, the same response is not reproduced here.

Rev1

Authors: Regrettably, the reviewer is incorrectly rephrasing our conclusions: we did not claim that a strong IOD can lead to a strong El Nino; what we did claim was that a strong IOD can create conditions that favour a moderate, transient warming of the tropical Pacific. We also concluded that interaction between positive IOD and El Nino is needed for super El Ninos to develop - all the super El Ninos so far (1972, 1982, and 1997) satisfy these conditions.

Reviewer: I did not say authors are claiming that all strong IODs are leading to strong El Ninos. My question was if one of the strong IOD events could lead to a strong El Nino event, why it does not happen

for some other strong IOD events. I am not against the suggestion that IOD and ENSO interact and that could lead to stronger El Nino events. But the results presented here are not sufficient to prove the point while excluding any other possibilities.

This is what is written in the first paragraph of your manuscript: Here, combining observational analysis and numerical simulations, we demonstrate that all major features of a super El Nino result from the interaction of an El Nino event with a positive Indian Ocean Dipole event.

Which are the years for these interactions leading to super El Ninos? 1982 and 1997 (since not much analysis is provided for 1972 in your study)? Why there is no other strong events after the 1997 event in spite of having several positive IODs together with El Ninos. The discussion on just one event of 2006 is not very conclusive. The results are not robust to argue that there was no coupled feedback in tropical Pacific without a coupled model experiment. The authors did not discuss at all about 2012 when there were an El Nino and a positive IOD. What prevented that event to become a super El Nino? Is it not the decadal variability in the Pacific that constrained El Nino and super El Nino to evolve post 1997? I would also suggest to show the IOD and Nino indices together for the whole analyses period to see which others year could have (or not have) the possibility of such interactions.

1. The years in which IOD-El Nino interactions lead to super El Ninos are 1972, 1982, and 1997. These are years when an El Nino was present along with a strong positive IOD event.
2. Our model explains why there were no super El Ninos after 1997, upto the present. The several El Ninos that occurred after 1997 – 2002, 2004, 2009, 2014, 2015 (Supplementary Fig. 20 of revised version; also ONI index from NOAA) – did not co-occur with IODs. The moderate 2010 IOD event (Supplementary Fig. 22 of revised version), co-occurred with a weak La Nina (http://origin.cpc.ncep.noaa.gov/products/analysis_monitoring/ensostuff/ONI_v5.php).
3. The Pacific OLR anomalies shown in Supplementary Fig. 17 (of revised version) are very useful in diagnosing ENSO variations, as they represent a key part of the atmospheric response. There are no El Ninos visible from these OLR anomalies than 2002, 2004, 2009 and 2015. The Oceanic Nino Index (ONI, http://origin.cpc.ncep.noaa.gov/products/analysis_monitoring/

ensostuff/ONI_v5.php) suggest the same.

4. There was no IOD in 2012 (Supplementary Fig. 22) ; there was also no El Nino in 2012 (Supplementary Figs. 20, ONI index from NOAA mentioned above).
5. Nino3 and DMI time series are shown for the entire period of analysis in the revised version (Supp Figs. 1, 20 and 22)

Rev1

You also said in the first paragraph: We also demonstrate a self-limiting behaviour inherent in El Nino Southern Oscillation (ENSO) dynamics: this, a consequence of the atmospheric Kelvin wave response that develops to the east of ENSOs convective anomalies, dampens SST variability in the eastern Pacific and prevents super El Ninos from developing through tropical Pacific dynamics alone.

In this context, the discussions in the first paragraph of page 5 are relevant. In line 112 of that paragraph you said However, this remote effect (on SST) is countered by easterly wind anomalies that are present to the east of the convective anomaly (Figures 1a,e and Figure 3); these, the atmospheric Kelvin wave response to the convection anomaly, favour local equatorial divergence and upwelling, which offsets the downwelling effect of oceanic Kelvin waves arriving from the west.

Which easterly wind anomaly? The shown easterly wind stress anomalies are asymmetric around the equator and the peak anomalies are off-equatorial. Is it a Kelvin wave response? Additionally, how these anomalies and the self-limiting mechanism are destroyed or limited by an ensuing positive IOD? Which are the El Nino years when this limiting mechanism worked?

1. The easterly wind anomalies in the far-eastern Pacific are asymmetric because they are also modulated by atmospheric circulations forced by El Nino's convective anomalies over the ITCZ. These induce a Rossby wave circulation that modifies the symmetric response associated with the Kelvin wave. Additionally, in the real atmosphere the three-dimensionally varying background flow also distorts the structure of the equatorial waves.

The above mechanisms were already discussed in previous versions of the manuscript. In the revised version, we have tried to make the discussion more clearer (paragraph starting on line 133).

2. The easterly wind anomalies damp the oceanic response, as discussed using the ocean experiments. The teleconnection associated with a positive IOD induces westerly wind anomalies over the western equatorial Pacific. Although these anomalies are moderate, they persist during the time when self-limitation is strongest. The impact of these IOD winds are discussed in the paragraph starting line 299 of the revised manuscript. (see also Yamagata, 1985 for a discussion of the opposing effects, on the ocean, by the atmospheric Rossby and Kelvin waves associated with the convective response).
3. Self-limitation is a general feature of all El Ninos. This may be inferred from Fig. 1 of the revised manuscript, which shows standard deviation of SST and OLR anomalies during the period 1982 to 2015 (excluding the super El Nino events). For example, self-limitation is clear during 2015, as discussed in the manuscript.

Rev1

Authors: The nature of SST anomaly variation during the strong El Nino of 2015 further support our model - viz., super El Ninos can only develop when IOD co-occurs with El Nino. The confusion regarding the 2015 El Nino is related to its strong SST anomaly amplitude in the west and central Pacific; however, this is weak in the far-eastern Pacific. In other words, the strong eastern-ocean intensification that is a characteristic feature of super El Ninos is absent for the 2015 El Nino. The weakness of its SSH and wind anomalies compared to the 1997 event were presented in Fig. 1.

Reviewer: A comparison between two does not rule out the fact that one of them cannot be categorized as super El Nino unless we compare all the events in super El Nino category. Based on the figures shown in Chen et al. (2017), I see 2015 event has SST and WWE anomalies comparable to 1997 and 1982 events (please refer to Figs 1, 2 and 3 of Chen et al. 2017). And Chen et al did argue that the 2015 is a Super El Nino. So, please compare your results with them.

We have added Supp. Fig. 20 showing interannual Nino3 SST anomalies – these demonstrate that the amplitude of interannual SST anomalies associated with the 2015 El Nino is one standard deviation lower than those during the super El Nino years (1972, 1982, and 1997). Thus a large part of the extremity of the 2015 El Nino's SST anomalies is due to decadal variations, in agreement with a recent study (Park et al. 2017) – these authors suggest that the extreme SST anomalies during 2015 may be attributed to

anthropogenic climate change.

Rev1

Authors: Supplementary Figure 2, further, shows that the OLR anomalies during 2015 are one of the weakest in the available observational record. These disparities between the amplitude of SST anomaly and that of other ocean-atmospheric anomalies during 2015 call for a different explanation of the strength of its SST anomalies - recent studies, and our own assessment (Supplementary Fig. 3) points to the role of a longer time scale (decadal or climate change time scale)

Reviewer: I dont see a reason why OLR anomalies should be used to compare El Ninos defined by SST indices. Please show the amplitudes of Nino3 and Nino3.4 (instead of OLR anomalies above that region) to compare the strengths of El Ninos. And please compare your results with that of Chen et al. 2017.

We disagree with the reviewer's assertion that OLR anomalies are not appropriate for comparing El Ninos. For most scientists in the post-Bjerknes period, El Nino is a coupled phenomenon—it is characterized not only by SST anomalies, but also by strong coupled ocean-atmosphere variability. OLR anomalies are one of the important metrics that directly relate to coupled ocean-atmospheric processes taking place during ENSO.

As we have demonstrated here, for the 2006 and 2015 cases, considering the amplitude of the SST anomaly alone can be misleading. In our modern era, a variety of observations are available to monitor coupled aspects of El Nino. In the past this has not been possible. There is no reason, not to comprehensively evaluate an El Nino event using various metrics, and to see if the relationships between the metrics validate the presence of El Nino.

We take this opportunity to express our gratitude to the reviewer for their evaluation of our manuscript over the various revisions, and for taking the time to provide comments. We are also thankful for the suggestion to consider decadal anomalies, and appreciate the opportunity to answer the various questions put forward by the reviewer.

1.2 Response to Referee #2

Recommend: minor revision

The authors have made further substantial changes in response to the reviewers' comments, thus

improving their article.

With further revision to take care of some details, and address one issue described below, this will be a paper of interest and worthy of publication.

We thank the reviewer for the positive comments and for taking the time to provide further suggestions. Point-by-point responses to the reviewer's comments are given below.

Rev 2

Specific comments:

1. Over the course of several revisions, a lot of supplementary material has become mixed in with the main article without clear separation. Some rearrangement to make the distinction clearer would be welcome. Perhaps much of the detail in 'methods' could be moved to 'supplementary' to allow some material into the main article, and perhaps some experiments could be moved to 'supplementary' in a self-contained way.

We agree with the reviewer that some rearrangement is needed to improve the text. To this effect, we have substantially edited the text so that the language and logical flow of the text was improved. We also moved a substantial amount of information from figure captions into the main text. This has made the text more self-contained. We also trimmed the material to make room for these descriptions—we feel that the text has improved in clarity and logical flow as a result of this. We also have made substantial efforts to improve the captions in both the main and supplementary figures. Further, figure labels were edited to improve clarity.

Further, we agree that some part of the 'Methods' section could be moved out. Following on this, we have included only relevant material about the oceanic and atmospheric models that are needed to interpret the experiments. More detailed information about the models were moved into a 'Supplementary Methods' section.

Rev 2

2. The issue is with the conclusion that the 2006 El Nino was forced by IOD dynamics, and that westerly wind bursts were not evident. In Fig. 7c there are westerly wind episodes in the west Pacific in July, August and particularly October 2006, each of which contributed to an ocean response that led to SST increases in the far eastern equatorial Pacific, and to central Pacific SST increase after the October event. I am surprised that none of these met the westerly burst criterion set by the authors. The corresponding observations and simulations with the atmospheric model shown in Fig. 8a-d seem

to have a different timescale, with some unspecified time averaging that emphasises longer timescales—this needs explanation. The evidence presented suggests that the atmospheric anomalies in the east Indian ocean region could have had a substantial influence on the western Pacific winds, but does not rule out a role for higher frequency wind-burst-type events.

- Yes, we agree that more clarification on this issue should be provided in the manuscript. We have added a supplementary figure 9 to address the issue. The noted features in Fig. 7c may be interpreted as modulations of the interannual component by intraseasonal variations—the latter increase the interannual westerly anomalies during their westerly phase, and reduce them during their easterly phase. This leads to the visible westerly wind episodes. Throughout the development phase of IOD, the westerly and easterly phases of intraseasonal variations contribute equally—the amplitude of each phase amounts to about 18% of the amplitude of interannual wind anomalies. Thus the intraseasonal anomalies have no net effect on the ocean, although they do introduce the visible undulations in the wind and sea surface height anomalies. The strongest eastward propagating sea-level anomalies are triggered in October, when the interannual winds are at their peak, and intraseasonal variations are significantly low. *Note that in the revised version, we represent the strength of the intraseasonal variations in comparison to the interannual wind anomalies.*
- The definition of westerly wind bursts include a threshold for speed (4.0 ms^{-1}) which is not exceeded by daily wind anomalies during the period analysed. Further, Fig. 2 of Chen et al. (2017), which shows another westerly wind burst index for January to July, suggests that westerly wind burst activity was the weakest in 2006 in their period of analysis from 1979 to 2015. Another measure of westerly wind burst intensity is shown in Fig 5 of Chen et al. (2015) for Jan–Feb, Mar–Apr, and May–Jun—this index also shows that 2006 along with 2009 had the weakest westerly wind burst intensity for the period 1982 to 2014.
- Fig. 7 was presented using daily anomalies—this was done to emphasize the relative lack of high-frequency activity compared to interannual activity. We hope that the supplementary fig. 9 will clarify the relative strength of interannual anomalies. The simulations are based on monthly anomalies. We had smoothed it in longitude to emphasize scales longer than 5° . In the revised

version, we have removed the spatial filtering. We also had used a 3-month running mean to smooth anomalies in time—in the revised version, we mention this at the end of the figure captions.

Rev 2

3. Minor comments and suggestions (line numbers as in the tracked-changes version):

line 24:anomalies associated with ...

line 202: Next, in place of Then,

line 209: better to say 'Here the cold tongue cools ... while expanding ...'

line 283: .. prominent positive IOD ...

Thank you for the suggestions above.

Rev 2

line 318: is marine cloudiness a good (or at least reasonable) indicator of deep convection? Thank you for asking us to clarify this, as the question may be raised by some of the readers. We have added a note (lines 196–198 of untracked version of manuscript) as follows:

“ The Indian Ocean region considered is part of the tropical warm pool and features deep convection throughout the year (Meenu et al 2010); over here, marine cloudiness may reasonably represent deep convection. ”

Rev 2

Line 400 and following: it would be more reasonable to say that the transition to warm Nino3.4 was correctly predicted, but the sharp increase in Sep-Oct was not captured, and the sharp decline in Dec-Jan was underestimated. Note that many forecast models contain representation of Indian Ocean processes, so it would be worth recommending analysis of forecast behaviour in that region in the discussion.

Thank you for the comments. Yes, it would be reasonable to say so. However, El Nino conditions are forecast only when certain thresholds are met—for example, Nino3.4 exceeding 0.5° for 3 consecutive months. This may be the reason that an El Nino was not forecasted by the operational agencies (the comments on the failure of forecasts were paraphrased from McPhaden (2008)).

We have added a line towards the end of the discussion *to recommend analysis of forecast behaviour in the western Pacific* as follows (lines 381–383 of untracked manuscript):

To evaluate these processes, detailed analyses of forecast behaviour in the western Pacific during IOD years must be undertaken. The 2006 IOD event may provide a benchmark for assessing how equatorial teleconnections associated with an IOD event are resolved in such

models.

Rev 2

Line 463: further rather than once again

line 466: ... positive IOD events

line 515: delete 'It is only reasonable to assume that'

Thank you for these suggestions.

Rev 2

line 619: $2c/\beta$, and 470km seems on the high side Some authors use $2c/\beta$ (Clarke, 2008: An introduction to the dynamics of El Nino & the Southern Oscillation, Academic Press). In our experience from observational data, the decay scale of Kelvin waves in sea surface height anomalies seem to be about 5° in latitude. However, since most people use c/β , we have changed it so, and mention the decay scale as 330 km.

Supplement: define DMI Done

Supp Fig 6: should this correspond to Fig 6d? Is this May-June-July forcing? Thank you for the question. This should correspond to Fig. 5e (in revised version). The Supp Fig 6 in previous version is now Supp. Fig. 5.

Rev 2

Supp Fig 7 caption: 'correlation' in place of 'relative strength' Done.

Supp Fig 10: are these all 3-month averages (including observed)? Yes, the forecasts were provided as 3-month averages, and observed anomalies were smoothed with a 3-month running mean.

We take this opportunity to express our gratitude to the reviewer for their thorough evaluation of our manuscript over the various revisions, and for taking the time and energy to provide detailed and constructive comments. Your advices enabled us to significantly improve the data and arguments presented, and the overall presentational aspects of the manuscript.

1.3 Response to Referee #3

SUMMARY: The authors responded to all the concerns in my second review favorably. I now have no scientific objections to publishing this paper. In reading over the manuscript, however, I still have editorial concerns (confusing text, somewhat awkward English, etc.). Below, I note a few examples. I have not gone over the paper in detail, carefully reading all the changes to the text, so I expect there are more. So, I request that the authors read over all the new text that they have written (as well as old

text) several times. Be sure the text is in good English, and that it flows logically. After these editorial concerns are addressed, I believe the paper will be acceptable for publication. I do not need to see it again.

A brief list of specific editorial comments follows. There are more, which I leave up to the authors to find and eliminate.

We are glad to note the positive comments from the reviewer, and thank the reviewer for taking the time to provide further suggestions. Following on your advice, we have edited the manuscript to improve the clarity, the English, and the logical flow of the text. We also consulted and took advice from a colleague, who is a native English speaker, so that editorial concerns are adequately addressed. Point-by-point responses to the reviewer's comments are given below.

SPECIFIC COMMENTS:

Rev 3

1. abstract, last 4 lines: Replace with: "...(ENSO) dynamics: the atmospheric Kelvin wave...convective anomalies dampens SST variability in the eastern Pacific,thereby preventing...".

Done.

2. line 36: There should be a comma after "analysis." Thank you for pointing this out.
3. line 50: The hyphen should be a minus sign, that is, " -40 W m^{-2} ." Thank you. However, we moved the discussion of the 2015 event to the last section (Discussion), and revised it thoroughly. During this process, this line was removed.

Rev 3

4. line 56: Since you replace "Firstly" rather than use "First," you should replace "Secondly" with "Second." As mentioned in the response to comment #4, this line no longer appears in the text, after the language edits.
5. line 60: There should be no space before "weak." As mentioned in the response to comment #4, this line no longer appears in the text, after the language edits.

Rev 3

6. line 82: Does it make sense to add the phrase "As we shall see," at the beginning of the sentence? **Maybe.** Yes, it does. However, as mentioned in the response to comment #4, this line no longer appears in the text, after the language edits.

7. line 88: Isn't "nonlinearity" a single word that does not need a hyphen? Corrected.
8. lines 106 and 107: Replace with "Indian Ocean, and this anomaly pattern." Thank you. Corrected.
9. line 107: Replace with "Second, we show." Thank you. After the language edits, the line referred to no longer appears in the text,
10. line 108: Replace with "ENSO dynamics, transforming a developing." Thank you. After the language edits, the line referred to no longer appears in the text,
11. lines 128–142: This new text is very confusing to me. The first line of the highlighted text states that: "This [weakening] manifests a self-limiting aspect of ENSO dynamics, the physical reason for which is as follows." But, the text that follows does more than talk about self-limitation, as it first discusses Bjerknes feedback. That is not logical, and as a result it took me several readings to figure out your intended meaning. You need to clarify this paragraph. Since it really discusses two ideas, you need to provide some clue to your readers that is what you are doing. One approach might be to split the paragraph into two separate ones. Another might be to introduce "First" and "Second" at the beginning of relevant sentences. Regardless, this text needs to be clarified.

Rev 3

We have modified the text to improve the clarity of the discussion (lines 95–107 of untracked manuscript).

12. line 154: Doesn't "ocean" need to be capitalized? Yes. Corrected.
13. line 161: Replace with "to zero over the far-eastern Pacific (15°S–15°N, 120°W–80°W)." Thank you for this suggestion. We have corrected the sentence as suggested.
14. lines 214 and 224: In each sentence you use "this" without a noun. It is not quite clear to what they refer. Perhaps replace the first with "this difficulty" and the second with "Their existence requires the presence of." There are other similar places in the text where the meaning of "this" is not clear.

Rev 3

Thank you for pointing this out. We have examined the manuscript thoroughly and removed such mistakes.

15. **lines 250–252: Shouldn't the past tense usage be present tense?** We have changed the past tense usage to present tense.
16. **line 254: Replace "on this" with "on these properties."** Thank you. We have corrected this.
17. **line 458: I don't think you mean to start a paragraph here.** Thank you for the comment.
18. **line 574: Don't start a paragraph here. Replace with "where τ_x and τ_y are the wind-stress forcings,...".** Corrected.
19. **line 593: Don't start a paragraph here.** Corrected.
20. **line 617: Replace with "America. The model resolution."** Replaced.
21. **lines 630–632: Replace with: "radiation scheme. The background state $\psi_b(x,y,z,t)$ is set to the seasonal climatology of the NCEP reanalysis data. The diabatic heating that forces the atmosphere is represented by an externally prescribed function, $Q(x,y,z,t)$." Just too many semicolons. By the way, elsewhere you overuse semicolons. Perhaps split them off into separate sentences.** Thank you. We have reduced the use of semicolons elsewhere too.

Rev 3

Rev 3

We take this opportunity to express our gratitude to the reviewer for their thorough evaluation of our manuscript over the various revisions, and for taking the time and energy to provide detailed and constructive comments. Your advices enabled us to significantly improve the data and arguments presented, and the overall presentational aspects of the manuscript.

1.4 Response to comments from Referee #2 regarding points made by referee #1

We sincerely thank the referee for taking the time and energy to provide advice on our disagreements with referee #1. These comments were extremely useful to us and helped us understand the issues with the unresolved questions. In particular, we feel humbled by the referee's assessment, and regret that we were over-assertive in describing our results. In the revised version, we hope that you will find that we have balanced caution with confidence as appropriate.

In the following, we respond to the comments from Referee #2 regarding points made by referee #1. The line numbers below refer to that in the untracked version of the revised manuscript.

Rev2::1

Definition of super El Nino:

Classification of El Nino events is somewhat subjective, and there are no 'official' definitions. Based just on the strength of central-eastern equatorial Pacific sea surface temperature (SST) the 2015/16 event should be regarded as a super El Nino ranking alongside previous very strong events, and I believe most climate scientists would be happy with that point of view. Based on the more limited extent of the large SST warming in 2015/16 (not reaching right up to South America), and on the accompanying atmospheric anomalies, the 2015/16 event is also recognised as somewhat different to the other very strong events.

Thank you for the comments. As the referee points out, based just on the strength of monthly SST anomalies the 2015/16 event may be regarded as a super El Nino: super strong in terms of the SST anomaly amplitude.

It may however be noted that SST changes are but one aspect of El Nino. Scientists, forecasters, and policymakers care about El Nino because of its significant societal impacts. Many of these impacts are due to changes in weather patterns locally within the tropical Pacific or its immediate vicinity. Outside the tropical Pacific, weather patterns are perturbed by atmospheric teleconnections forced by Pacific convective anomalies associated with El Nino. Further, El Nino is a coupled phenomenon—it is characterized not only by SST anomalies, but also by strongly coupled ocean-atmosphere variability. Therefore, in addition to the spatial organization of SST anomaly, we had pointed out to the unexpectedly weaker convective anomaly during 2015 to argue that 2015/16 may not be as super as it seems to be. *However, we concede that our arguments are somewhat qualitative and were not backed up by an objective SST anomaly index for "super El Nino" in the previous versions.*

Rev2::1

Is the definition used by Hameed et al justified? Their statement in the introduction 'we do not consider the 2015 to be a super El Nino' is a rather particular viewpoint. While they explain the reason, given that others do regard 2015/16 as a super El Nino I think it would be preferable if they modified their viewpoint and definition. If they regarded 2015/16 as a super El Nino, but used the term 'super EP El Nino' for the other events, that would be non-controversial. (EP meaning 'East Pacific', a terminology already in use in classifying El Nino events that occur more towards South America.) They could then

propose reasons why 2015/16 was not a super EP El Nino.

We are sorry that our statement was over-assertive, but was not backed by an objective index. In the revised version, we have adopted a simple index from Hong et al. (2014) to address the question about “how to identify a super El Nino”. In this metric, which measures the amplitude of *interannual* SST anomaly over the equatorial eastern Pacific (5S–5N;150W–90W), super El Ninos are differentiated from the rest by their extreme anomaly amplitude—the 1972, 1982, and 1997 El Ninos are separated by 1 standard deviation from the rest of the El Ninos in the post World War II SST record.

We regret that we did not include an SST based index for super El Ninos from the beginning. This could have eliminated much of the confusion about our rather unique view point on the 2015 El Nino. Here, we would like to explain why we did not include an SST based super El Nino index in earlier versions. Hong et al (2014)’s index is based on interannual SST anomalies. To define interannual SST anomalies, decadal variations should be removed from monthly SST anomalies. In principle, it is challenging to remove decadal anomalies from the end-points of a time-series, and the 2015 event occurs towards the end of the current record. (*Lanczos filters are applied as weighted running mean averages, and a portion of the data at the end-points have to discarded*). This limitation prevented us from attempting such a filtering in the previous versions. However, with data available up to Dec 2017, it is possible to design a 49-weight filter that can remove decadal signals from 2015 as well. **In SuppFig. 20, discussed in line 357, we show the same super El Nino index, but with data shown upto Nov 2015. This analysis demonstrates that a large fraction of the extreme SST anomalies during 2015 are due to decadal anomalies, as we had originally suspected.** *A caveat with the low number of weights that we have used in SuppFigs. 18,20,22 is that some interannual frequencies may leak into the decadal signal (in principle, SuppFig. 19). However, visual examination of the estimated decadal anomalies shows that the decadal trend from 2012 is a monotonically increasing trend, which clearly has no such leaked interannual signals. Therefore, we are reasonably confident that our estimate of decadal anomalies during 2015 is robust. Because of this caveat, however, we have moved the discussion of the 2015 event into the last section (Discussion). The caveat is also discussed in lines 369–373.*

Hong (lead author of Hong et al., 2014) in her PhD thesis (available online at <https://link.springer.com/book/10.1007/978-981-10-0527-5>), considered not only the problem of

identifying super El Ninos, but also synthesized valuable information about the structure of super El Ninos from insitu observations as well as oceanic and atmospheric reanalysis products. In a sense, the study of super El Ninos is just in a preliminary phase. *We hope that the numerical experiments, along with the paradigms suggested in our work, will compliment the observationally based characteristics of super El Ninos, noted by Hong, to simulate further thinking and research in the field.*

Rev2::1

The main reason they propose is that there was not an Indian Ocean Dipole (IOD) event in 2015. Here the approach used by the authors is unnecessarily confusing. Using the standard IOD index, based on differences between observed eastern and western Indian Ocean SST anomalies, there was a positive event in the latter half of 2015. However, taking into account other factors such as actual SST and atmospheric anomalies, the authors do not consider 2015 as a positive IOD year, as explained in their supplementary material. This is confusing for the reader: it would be much better if the authors accept that 2015 was an IOD year (based on the SST index), but define and use another (atmospheric) index for the relevant Indian Ocean region atmospheric state. (With regard to influence on the equatorial Pacific, what counts is the atmospheric state in the Indian Ocean region [albeit usually closely connected with the oceanic state], and from the observational evidence presented in the article this was relatively quiet in the 2015 El Nino episode.)

In our defence, we would like to point out that we could have claimed that 2015 was a positive IOD year, and that due to its strong SST anomalies 2015 was also a super El Nino year. Based on these claims, we could have argued that all available observations fit with the proposed model. We feel that a large part of referee #1's disagreement with us is due to our argument that 2015 is not an IOD year.

Just as the referee noted in relation to El Nino, classification of IOD is also subjective. Our work (Saji et al. 1999) was the first to define an index for IOD. However, in that work, we had taken due care to remove decadal SST variations before identifying an IOD event. The rationale for removing decadal SST variations is discussed in Saji and Yamagata (2003), and in our recent IOD review forthcoming in the Oxford Research Encyclopedia (<http://climatescience.oxfordre.com/page/forthcoming/>).

However, many authors do not take due care in identifying IOD events, and make claims about IOD events that may not be correct. In the Pacific Ocean, SST variations are dominated by ENSO, but SST

in the Indian Ocean is somewhat equally influenced by various factors ranging from a global-warming trend and decadal anomalies to thermodynamically forced SST variations induced by ENSO (Wallace et al., 1998). Unless due care is taken, it is easy to misdiagnose IOD events. Saji and Yamagata (2003) also suggested a variety of process based metrics, to be used along with the standard DMI index, for IOD identification.

Figure 1

Figure 1 shows the DMI from raw monthly anomalies (bars) from January to December of 2015. The datsource is the HadSST3 monthly SST from MetOffice(U.K.). IOD is characterized by a rapid peaking of DMI in boreal fall. However, in Fig. 1 above, DMI is strongly elevated in March and from June–August. From August, DMI rapidly declines and is even negative in October. This behaviour of DMI is not characteristic of IOD (Saji et al. 1999). The solid blue line is the raw monthly anomaly smoothed with a 3-month running mean, and the dashed line is bias-corrected DMI (see Methods section of our manuscript; also Saji and Yamagata, 2003). During boreal fall of 2015, the value of DMI (Supplementary Fig. 22) is about 0.2 standard deviations (cannot be distinguished from climatology). Consistent with this, the actual observed ocean-atmosphere anomalies (Supplementary Fig. 21) also do not suggest a positive IOD in 2015.

The authors make use of Indian Ocean region atmosphere anomalies in some of their models. Somewhat confusingly they call these IOD anomalies: this is terminology which could be avoided. (I think this arises as they use composites based on the IOD index in years when there are also substantial and systematic atmospheric anomalies; but that is not always the case as evident in 2015.)

We agree that this is very confusing. We had used this term in many figure labels. In the revised version, **figure labels were changed in Figs. 8,9 and SuppFigs 12,13,15 following the referee's suggestion.** We use the term "IOD-induced winds" or the "simulated impact of IOD on the Pacific" to avoid confusion.

Rev2::1

Have the authors done enough to show how ENSO/IOD interactions lead to super El Ninos? Effectively they have used relatively simple ocean and atmosphere models and some rather selective composite or idealised forcing terms to demonstrate some plausible processes. Overall the results are suggestive rather than conclusive, but should provoke further investigation.

1. We agree with the overall assessment that the results are suggestive rather than conclusive.
2. The composites may appear selective because they are limited to the post-1979 satellite era. Since OLR data was used to force the atmospheric simulations, satellite era data was chosen for consistency with observed composites. *We will be expanding on these results in a work under preparation, where we compare, in further detail, the differences of super El Ninos from regular El Ninos. Here, we use historical data from 1940's onward.* The El Nino composite includes all well-known strong El Ninos, except for 2009 and 2015. We did not include 2009, because it had a Modoki-like structure, and we did not want the results to be biased because of this El Nino's spatial structure. We did not include 2015, since we wanted to compare its features with the composites and simulations that we presented. For IOD, we chose strong post-1979 IOD years consistent with our other works (Saji and Yamagata, 2003; Hameed 2017, Oxford Research Encyclopedia of Climate Science).
3. The forcing terms are idealized, but their spatial structure is derived from observed OLR anomalies, and therefore a certain degree of realism is built into the experiments presented.

Rev2::1

Is this article an important advance? The authors are over-assertive in describing their results, and the result is not as clear as they suggest. In my opinion: worth publishing as an interesting idea (with revised content), but not as a major advance.

We are humbled by the assessment, and regret that we were over-assertive in describing our results. We also thank the reviewer for commending our work for publication in Nature Communications.

Rev2::1

The authors should include discussion of other factors (westerly wind bursts in particular) that also influence large El Nino events, and take the point of view that Indian Ocean influence is one among several processes rather than a controlling factor. Reviewer 1 makes frequent mention of Chen et al (2017), as do the authors in their responses. This is a paper that principally discusses the role of westerly wind bursts in the 2015 event. (To make matters complicated, there are two relevant Chen et al. papers by different Chens: Chen L. et al Scientific Reports 2017, and Chen D. et al Nature Geoscience 2015 - also about westerly bursts and El Nino.) The authors do not mention either in their article, and they should mention both.

In the revised manuscript, we have enhanced the discussion of westerly wind bursts, and also mentioned Chen et al. (2015) and Chen et al. (2017). We also address factors related to decadal variations in the revised version. Our analysis suggests that SST anomalies during the 2015 El Nino were likely enhanced by decadal anomalies.

Rev2::1

In the light of this, I would recommend that the authors (in addition to other recommendations already made):

Rev2::1

- the title of the article to e.g. Indian Ocean processes that influence major El Nino events'

We thank the referee for the suggestion to change the title to "Indian Ocean processes that influence major El Nino events". However, the article not only discusses the impact of IOD, but also about ENSO's *self-limiting dynamics*. This is discussed through Pacific Ocean processes. Further, the original title implies that it is only one suggested model (A model for super El Ninos)—we feel that the distinction should be clear to most readers. Since our model, albeit its simplicity, discusses both Pacific and Indian Ocean processes of relevance to the issue, we hope that we can keep the title as it is.

Rev2::1

- the 2015 event explicitly in the abstract

Done; The last sentence of the abstract now reads:

Our model explains the features of the 1972, 1982, and 1997 El Ninos; the SST anomalies during the 2015 El Nino, however, were likely enhanced by strong decadal variability.

Rev2::1

- that 2015 was a super El Nino, and describe it as different from other such events e.g. not an EP-type super El Nino.

1. In lines 334–335 of revised manuscript, we state:

In the rest of the article, we discuss **the 2015 El Nino which also appears to be a super El Nino due to its extremely strong SST anomalies.**

2. We then proceed to discuss its difference from other super El Ninos by examining its structure and coupled variations, and also using the interannual Nino3 SST index (cf. Fig. 1 of Hong et al 2014).

(a) With the decadal analysis, we are able to show that peak interannual Nino3 SST anomaly amplitude for 2015 is 1 standard deviation lower than for the 1972, 1982, and 1997 El Ninos : The latter events are well-separated (by 1 sigma) from other El Ninos in the reasonably well-sampled post World War II observational record.

(b) There is some uncertainty in the decadal analysis towards the end of the time series, and this is discussed in the main text (lines 369–373).

Rev2::1

- that 2015 was a positive IOD year (based on the standard index), but classify Indian Ocean atmospheric anomalies. (Along the lines that reviewer 1 suggests, it would be helpful to provide a figure with the IOD index, plus an index (based on OLR?) for the atmospheric anomalies that would make the important atmospheric component clear.) Avoid expressions such as 'IOD wind anomalies' .

1. We have **added a statement that DMI calculated from monthly SST anomalies would**

show a weak positive IOD, and thereafter discuss that DMI should be calculated appropriately for IOD identification (**para starting line 359**).

2. We have **added SuppFig. 20 to show the IOD index** (DMI) from 1958 to 2015.
3. We have also addressed the referee's caution about using the term IOD anomalies. We agree that this is very confusing. We had used this term in many figure labels. In the revised version, **figure labels were changed in Figs. 8,9 and SuppFigs 12,13,15 following the referee's suggestion**. We use the term "IOD-induced winds" or the "simulated impact of IOD on the Pacific" to avoid confusion.

Rev2::1

- **more discussion of other factors (westerly wind bursts in particular) that also influence large El Nino events, and take the point of view that Indian Ocean influence is one among several processes rather than a controlling factor.**

1. **New SuppFigs. 2, 9**, first discussed in **lines 74 and 269** respectively, were added to enhance discussion of westerly wind bursts. These also address concerns by referee#2 on the role of high-frequency windbursts during 2006. **Relatedly, we have referenced both Chen et al. (2017) and Chen et al. (2015), as suggested.**
2. We have also discussed decadal anomalies as a factor for the extreme SST anomalies during the 2015 El Nino.

Rev2::1

- **the point of view that the processes demonstrated using relatively simple models are suggestive rather than definitive, meriting further investigation.**

To address this concern, we have:

1. **toned down our language**, as advised by the referee. For example, the line reading "..., we **demonstrate** that all major features of a super El Nino result from ..." was changed to "..., we **suggest** that **eastern-ocean intensified super El Ninos** result from ...".
2. explicitly mentioned the caveats associated with our study in the discussion as quoted below:

So far as we can tell, the mechanisms elucidated here are consistent with available observations. **However, they are based on simple models and hence should be regarded as preliminary until verified against more complex models.** We hope that, since these mechanisms involve only the basic elements of equatorial dynamics, such future investigations will uphold the validity of our model. Further, due to data limitations, **a less optimal Lanczos filter was used to remove decadal variations from recent data (Supplementary Fig. 19), which may leak in a fraction of interannual anomalies into the estimated decadal signal.** However, the observed decadal trend, post-2011, does not show the existence of any interannual variations, suggesting that our estimate of the decadal signal during 2015 is robust.

We hope that you will find that we have addressed all the concerns that you have raised, sincerely and to the best of our abilities, and that these changes will persuade you to accept our manuscript for publication.

REVIEWERS' COMMENTS:

Reviewer #2 (Remarks to the Author):

Recommend: accept subject to minor presentational revision

The authors have made further substantial changes in response to the reviewers' comments. The material has been re-organised effectively, and this version reads well. Further details of the various filtering, compositing and averaging procedures are provided, making the results more transparent and comprehensive. (There is a large amount of supplementary material, but the extra detail is useful.) Due attention to the description/classification of the major 2015 El Nino event has been made. The description of the work is now less assertive, as is appropriate.

A few presentational suggestions (referring to the unmarked version):

Abstract: last sentence: 'the large 1972, 1982 and 1997 El Ninos; the large SST anomalies ...'

line 33: '... a large eastward expansion ...'

line 48: 'McPhaden later also called the ...', and delete the sentence 'However, it was Hong ...'

line 71 '... some El Nino events are eastern-ocean damped (i.e. not eastern-Pacific intensified). However, very strong ...'

line 103: '... easterly wind anomalies in the atmospheric Kelvin wave...'

line 227: '.. to clarify that atmospheric Kelvin waves...'

line 235: '... they inappropriately prescribed a zonal...'

line 240: Could add a section heading here, e.g. 'IOD influence in 2006'

line 322: 'The strongly asymmetric anomalies have a relatively weak symmetric component and thus relatively weak IOD-induced Pacific wind anomalies.'

line 331: typo: rapidly

line 348: 'Nevertheless, the SST anomalies observed ... similar amplitude to those during ...'

line 360: ' ... with a positive IOD event...'

line 379: 'The failed predictions on average ...'

Supplement:

caption Fig 1: 'Filtered amplitude of interannual ...'

Fig. 2: I am not sure what the probabilities mean here. Is it events/month divided by all events in the year? Perhaps better just to provide the count of the number of events.

Fig. 9: How are the daily wind anomalies calculated? Daily values minus what reference?

Fig. 10: 'Equatorial evolution of ...'

'The onset of the 1994 IOD event started earlier, ...'

Fig. 16: Omit 'significant' in the caption, or include the test for significance.

Fig. 20: 'Filtered interannual ...'

Fig, 22: 'Filtered monthly ...'

Fig. 23: Rather than 'The failed El Nino prediction of 2006', put for example 'Nino3.4 forecasts in 2006, which did not capture the onset of anomalies above the 0.5 degree threshold indicated.'

1 Response to Referee# 2

Recommend: accept subject to minor presentational revision

The authors have made further substantial changes in response to the reviewers' comments. The material has been re-organised effectively, and this version reads well. Further details of the various filtering, compositing and averaging procedures are provided, making the results more transparent and comprehensive. (There is a large amount of supplementary material, but the extra detail is useful.) Due attention to the description/classification of the major 2015 El Nino event has been made. The description of the work is now less assertive, as is appropriate.

We thank the referee for their positive assessment and recommendation and the additional suggestions.

1. A few presentational suggestions (referring to the unmarked version):

The point-by-point response to their suggestions are listed below. Line numbers as in the tracked version.

- (a) Abstract: last sentence: 'the large 1972, 1982 and 1997 El Ninos; the large SST anomalies ...'

Thank you. This was added. Please see lines 15–16.

- (b) line 33: '... a large eastward expansion ...'

Done. Please see line 61.

- (c) line 48: ‘McPhaden later also called the ...’ , and delete the sentence ‘However, it was Hong ...’

The sentence was deleted as suggested. In addition, we also deleted the entire paragraph referred to by the referee, and integrated it with the sentence beginning on line 54, taking into account editorial comments.

- (d) line 71 ‘... some El Nino events are eastern-ocean damped (i.e. not eastern-Pacific intensified). However, very strong ...’

Done. Please see line 115.

- (e) line 103: ‘... easterly wind anomalies in the atmospheric Kelvin wave...’

Thank you. We have incorporated this. Please see line 170.

- (f) line 227: ‘. to clarify that atmospheric Kelvin waves...’

Done. Please see line 343.

- (g) line 235: ‘... they inappropriately prescribed a zonal...’

Done. Please see line 352.

- (h) line 240: Could add a section heading here, e.g. ‘IOD influence in 2006’

Thank you for the kind suggestion. We have added a section heading on line 359.

- (i) line 322: ‘The strongly asymmetric anomalies have a relatively weak symmetric component and thus relatively weak IOD-induced Pacific wind anomalies.’

Thank you for the kind suggestion. Please see that we have incorporated this in lines 483–484.

- (j) line 331: typo: rapidly

Changed (line 496). Thank you.

- (k) line 348: ‘Nevertheless, the SST anomalies observed ... similar amplitude to those during ...’

Done. Please see lines 516–517.

- (l) line 360: ‘ ... with a positive IOD event...’

Done. Please see line 534.

- (m) line 379: ‘The failed predictions on average ...’

Thank you for the suggestion, which was incorporated (line 560).

2. Supplement:

- (a) caption Fig 1: ‘Filtered amplitude of interannual ...’

Thank you. This was incorporated.

- (b) Fig. 2: I am not sure what the probabilities mean here. Is it events/month divided by all events in the year? Perhaps better just to provide the count of the number of events.

We apologize for not explaining this properly. Our description was also incorrect. We did not take the area-average of equatorial wind anomalies over the west and central Pacific, but only did take latitudinal averages. We understand the confusion caused. The events were counted across spatial and temporal points. In the revised version, we expanded the discussion in the caption of Supplementary Fig. 2 as follows:

The frequency of occurrence of westerly wind anomalies (data source: NCEP reanalyses) equal to or exceeding 0.17 Nm^{-2} during each month of 1997 is plotted in the figure, and was calculated as follows. We first distributed daily, equatorially averaged (2.5°S – 2.5°N) wind stress anomalies at each of 38 grid points from the western Pacific (120°E) to the central Pacific (190°E) across 25 equally spaced bins. The number of data points where westerly wind anomalies equalled or exceeded 0.17 Nm^{-2} was divided by the total data points in each month, and multiplied by 100 to obtain the probability of occurrence shown on the y-axis. The number of data points where the mentioned threshold was reached is as follows: 1 in January, April–July, September, and November–December; 2 in February and October; 3 in August; 20 in March.

- (c) Fig. 9: How are the daily wind anomalies calculated? Daily values minus what reference?

The reference climatology for daily wind anomalies is a smoothed daily climatology constructed from the mean and first 3 harmonics of the raw (or unsmoothed) daily climatology. The Methods section was updated to address this ambiguity (see lines 577–582).

- (d) Fig. 10: ‘Equatorial evolution of ...’ ‘The onset of the 1994 IOD event started earlier, ...’

Thank you. This was corrected.

- (e) Fig. 16: Omit ‘significant’ in the caption, or include the test for significance.

Thank you for the suggestion. We have omitted ‘significant’ in the caption.

- (f) Fig. 20: ‘Filtered interannual ...’

Thank you. We have added the word ‘filtered’ at the beginning of the caption.

- (g) Fig. 22: ‘Filtered monthly ...’

Done. Thank you.

- (h) Fig. 23: Rather than ‘The failed El Nino prediction of 2006’, put for example ‘Nino3.4 forecasts in 2006, which did not capture the onset of anomalies above the 0.5 degree threshold indicated.’

Thank you for the suggestion. We have changed the title of the said figure caption to ‘Nino3.4 forecasts in 2006’. Further, we

added the following at the end of the figure caption.

-Note that although warm Nino3.4 SST anomalies were correctly predicted, on average the forecasted anomalies were below the 0.5 degree threshold for El Nino. The forecasts also did not capture the peak value of the 2006 event. Further, they failed to predict the rapid termination of the event.

Finally, we would like to express our sincere gratitude for the time and energy that you took out to go over the various versions of the manuscript with such care and to provide your thoughtful and at the same time critical comments.